# Meta-analysis of 22,710 human microbiome metagenomes defines an oral-to-gut microbial enrichment score and associations with host health and disease

Large public datasets of the human microbiome now exist but combining them for large-scale analysis is difficult due to a lack of standardization. We developed curatedMetagenomicData (cMD) 3, a uniformly processed collection of over 22,000 human microbiome samples with manually curated metadata from 94 studies and 42 countries. This large and diverse resource allows for meta-analysis of the links between microbes and human health. Through meta-analysis, we identified hundreds of microbial species and thousands of microbial functions significantly associated with a person's sex, age, body mass index, and disease status, and catalog these as references. We developed an "oral enrichment score" (OES) based on the relative abundance of bacteria typically found in the oral cavity and not in the gut. Higher OES in the gut is a consistent feature in individuals with disease, suggesting that the relative abundance of oral bacteria in the gut is a simple and quantifiable signal of altered microbiome health. These analyses identify modest but widely shared patterns in human microbiomes, serving as a reproducible and readily updatable reference.

Microbiome associations with basic host characteristics such as age, sex, or body mass index (BMI) and different pathologies are important aspects of the study of the human microbiome in normal and altered physiology, but due to their complexity and modest strength of association relative to individual variability, great uncertainty remains in their characterization. Sex, for example, may modulate the gut microbiome via endogenous hormones[1–3], mediating susceptibility to several diseases[4]. The relationship between aging and the human microbiome has been extensively studied; adjustment for age, for example, can improve the identification of gut microbiome associations with disease[5]. Consistent age-associated microbiome changes, however, remain difficult to define. One of the most interesting and consistent findings has been a progressive increase in the gut microbial diversity of longevous populations[6,7], although this can be attributed to an increased presence of pathobionts and a possible enrichment of

oral microbes[7,8] or selection bias[9,10]. Meta-analyses that have characterized gut microbiome variation relative to BMI[11–13] have been limited in sample size and/or could not account for potential confounders due to a lack of curated metadata. Finally, gut microbiome shifts related to multiple diseases have been studied[14], providing promising patterns of microbial species that are altered in multiple etiologies and describing broad, potentially systemic aspects of the altered host's physiology[15–17].

Oral microbiome species can, in some cases, transit across the barrier of the stomach and reach the gut, where they integrate into the gut ecology[18]. Enrichment in oral bacterial species in the gut has been associated with colorectal cancer (CRC)[19–21], atherosclerotic cardiovascular disease (ACVD)[22], and inflammatory bowel disease[23]. Schmidt et al.[18] observed evidence of oral to gut transition of microbial strains in both healthy and diseased subjects. These findings motivate a

✉e-mail: nicola.segata@unitn.it; levi.waldron@sph.cuny.edu

quantitative definition of the extent of oral to gut microbial enrichment and a systematic investigation of its potential role across a range of diseases.

Here, we present version 3 of curatedMetagenomicData[24] (cMD 3), an expansion and refactoring of the original resource, providing 94 shotgun metagenomic datasets with manually-curated metadata from 42 countries and 6 continents. This version provides 22,710 samples (3.6 times larger than version 1, including 3.5 times more studies) with updated taxonomic and functional potential dedicated tools, with expanded manually curated metadata on more than 100 different individual-participant characteristics. cMD 3 provides a higher degree of manual curation than alternatives[25,26], allowing adjustment for some potential confounding factors in meta-analysis. Additionally, cMD3 is freely available via ExperimentHub and example vignettes are included in the resource, making quick usability a key advantage. We interrogate cMD 3 to characterize microbial signatures of sex, age, BMI, and fifteen pathologies, and to provide an updated survey of cross-study machine learning prediction accuracy. We then define a numeric "oral enrichment score" (OES) to enable simple quantification of the relative abundance of oral-associated species in the stool microbiome and show that OES is associated with age and multiple diseases. This work improves the current knowledge of host phenotype-microbiome links and provides tools and microbial signatures to support future epidemiological studies of the human microbiome.

## Results

### A resource of 22,710 manually curated and uniformly processed metagenomes

curatedMetagenomicData (cMD) version 3 is a data package publicly and freely available in R/Bioconductor, with a command-line interface providing uniformly processed, quality-controlled shotgun metagenomic data and manually curated metadata. Manual curation was performed by a panel of 17 curators supported by machine-based validation. We used the bioBakery 3 pipeline[27] to generate quantitative taxonomic abundances (MetaPhlAn3) and functional potential estimates (HUMAnN3, see "Methods").

The cMD 3 includes 22,588 samples from 93 different human microbiome datasets and 42 countries (Fig. 1a, Supplementary Data 1) plus one more study (HeQ_2017, $n = 122$[28]) that was included in this paper. Manual curation was employed to standardize attributes including body site, country, age category, general lifestyle, sequencing information, and health/disease-related information. Key covariates in microbiome analysis are available for most participants, including age ($n = 21,213$), sex ($n = 19,751$, males = 9773, females = 9978), BMI ($n = 12,826$), and recent or current antibiotic usage or other therapies ($n = 28,099$, Fig. 1b).

Although the sample set analyzed is dominated by stool specimens ($n = 21,152$, 93%) - reflecting the current focus in human microbiome studies - it also includes 857 samples from multiple oral cavity locations, 504 skin samples, 96 vaginal samples, 93 nasal cavity samples, and 8 breast milk samples. 7246 samples have specific geographical information, linking them to 144 cities or villages (Supplementary Data 1, sheet 2), and 4270 subjects are grouped within 1609 households. A total of 2599 participants were sampled at more than one time point (10,328 samples, median of 3 samples per participant, range: 2–57; median time between collections = 180 days). Fifty-one datasets include participants with a health status considered a disease or a deviation from health, representing 142 distinct host conditions. The 3 diseases with the most cases and control samples are CRC ($n = 1650$ from 11 studies), inflammatory bowel disease (IBD, $n = 3278$ from 8 studies), and type-2 diabetes (T2D, $n = 3439$ from 11 studies, Supplementary Data 1, sheet 2). The data and metadata included in pre-release versions of cMD 3 have been already used to investigate specific microbiome components in relation to their prevalence in different populations[29,30], geographical distribution[31–33], association with host phenotypes (e.g., age or disease[12,19,34,35]) and their inferred ecological relationships with other members of the microbiome[36–38].

### Meta-analysis to identify sex-associated microbial features

Sex differences in the human microbiome may arise through differences in genetics, nutrition, or response to environmental exposures such as drugs, endogenous hormones, or disease[1,3,7,39–42]. Different configurations of the gut microbiome associated with sex have been reported[2,4,5,13,43], but these studies have been limited in population diversity and standardized metadata. We queried cMD 3 for stool metagenomes from cross-sectional studies (using only the first time point in time-series datasets), including healthy (as defined by not being diagnosed in the original study by a specific disease) adult (>16 years) participants, with within-dataset balance of sex (>25% for the least represented sex) and a minimum of 40 samples per sex type. We obtained a total of 5505 samples (2216 males, 3288 females) from 13 countries spanning Asia, Africa, Europe, and North America. This represents the largest available resource for investigating sex-linked microbiome features in the healthy adult gut.

Alpha diversity (Shannon entropy) was significantly increased in females with the standardized mean difference (SMD) of −0.16 (95% CI [−0.21, −0.11], $P < 4.6 \times 10^{-9}$) relative to males, as previously reported[40]. Sex contributed significantly to beta diversity (FDR = 0.1) in 15 and 7 datasets out of 18 by PERMutational ANalysis Of VAriance (PERMANOVA) and ANalysis Of SIMilarities (ANOSIM), respectively. We then searched for individual microbial features differing by sex-based SMDs adjusted by age, BMI, and sequencing depth. In total, we found comparably more differentially abundant species and genera in men than women (Fig. 2a, b, tot. significant: 30 and 23 species, and 33 and 16 genera, FDR = 0.01, Supplementary Data 2). The meta-analysis approach proved superior to the single-cohort analysis approach as none of the sex-specific associations found in the meta-analysis were identified at FDR = 0.2 in more than one-third of the single, under-powered cohorts, even though the direction of association is broadly conserved (Supplementary Fig. 1). Likewise, we did not identify significant heterogeneity (avg. $I^2 = 16\%$, $Q_{gen} > 0.1$) in all the significant species.

The two strongest species associations with females were *Akkermansia muciniphila* (SMD = −0.27, 95% CI [−0.35, −0.19], FDR = 0.0001), in line with the meta-analysis by ref. 8, and *Intestinimonas butyriciproducens* (SMD = −0.22, 95% CI [−0.27, −0.16], FDR = 0.0001). The higher abundance of *A. muciniphila* in adult females in 18 studies reiterates the necessity of accounting for sex as a potential confounding variable in microbiome studies. The two strongest associations with males were *Phascolarctobacterium succinatutens* and *Prevotella copri*, (SMD = 0.21, 95% CI [0.1, 0.32], and 0.21, 95% CI [0.14, 0.28], FDR = 0.0001), the latter consistent with Zhang et al.[7]. Interestingly, all top four associations with sex can be linked with differences in dietary habits and physiological response to food: *P. succinatutens* was previously observed to increase in males after a weight-loss intervention[44], and *P. copri* has been previously associated with non-Westernized lifestyles[29], and fiber-rich and Mediterranean diet-related nutritional habits[45,46]. Conversely, *A. muciniphila* and *I. butyriciproducens* are important butyrate producers[47]. *A. muciniphila* has been shown to play a role in decreasing the risk of metabolic syndrome[48]. Our findings are potentially explainable by the longer transit time previously observed in females[49], confirmed in a recent investigation[50], and they were partially mirrored in the functional potential analysis, where for example the *L-lysine fermentation to acetate and butyrate* was increased in females (SMD = −0.11, 95% CI [−0.18, −0.05], Q = 0.009, Supplementary Fig. 2), and maltose-6'-phosphate glucosidase was enriched in males (SMD = 0.17, 95% CI [0.12, 0.22], FDR = 0.0001, Supplementary Fig. 2a, Supplementary Data 2).

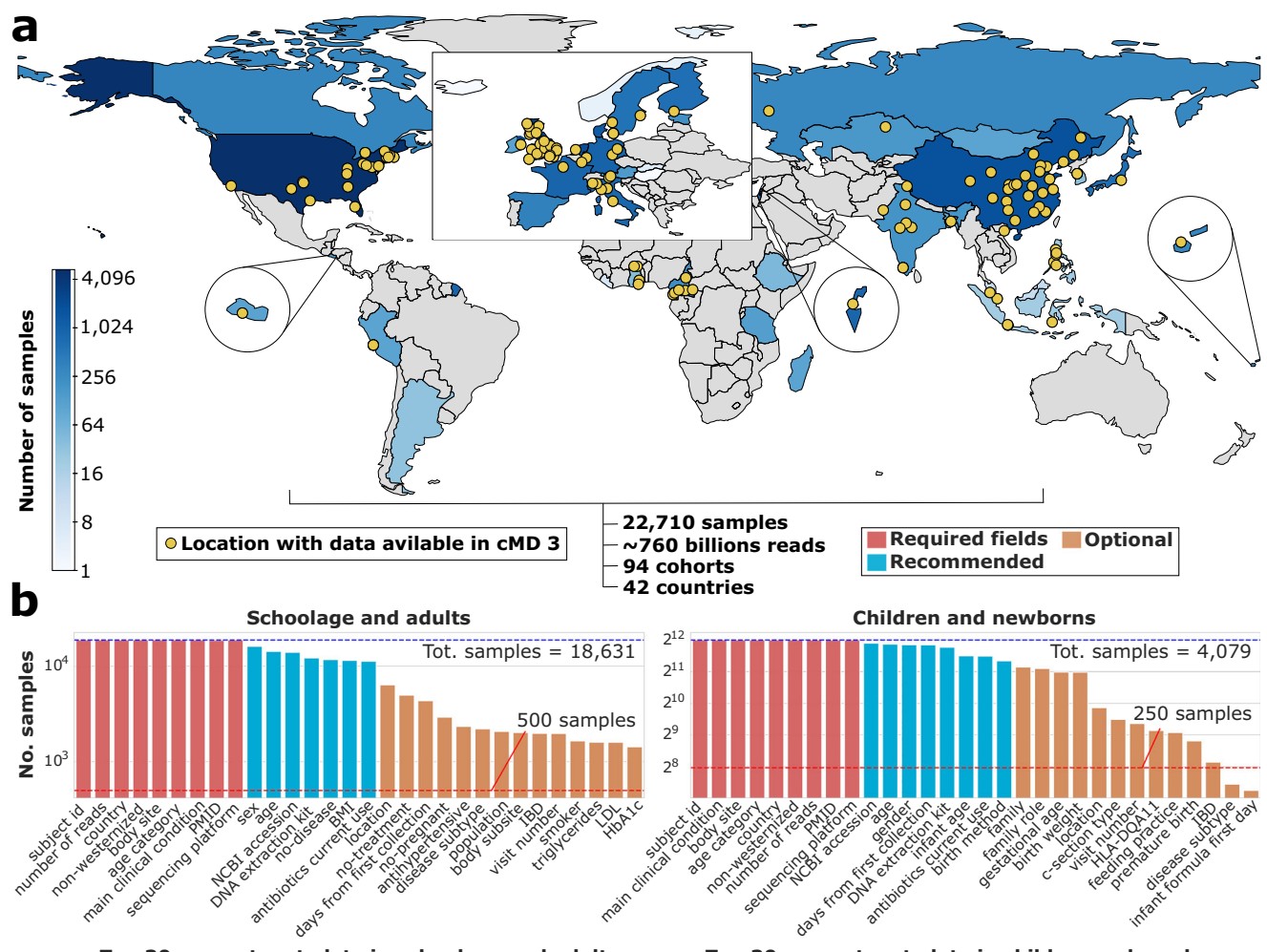

**Fig. 1 | curatedMetagemomicData (cMD) 3 is an open-source software package providing uniformly processed metagenomic data and manually curated metadata, available in R/Bioconductor and via the command line. a** cMD 3 distributes more than 22K metagenomes derived from 94 cohorts and 90 publications (42 countries and ~160 more detailed locations). cMD 3 metagenomic data are profiled with the MetaPhlAn 3 and HUMAnN 3 tools from the bioBakery3 suite. cMD 3 contains standardized, manually curated, and machine-validated sample-

level metadata on host's characteristics and specimen's technical attributes. **b** Barplot showing the top 30 most complete metadata attributes among the samples from individuals above 12 years (left) and below (right), where the red lines mark 500 and 250 samples, respectively. Blue lines mark the total number of samples in each category. Metadata are colored according to whether they are required (red), recommended (blue), or optional (orange).

We additionally applied machine learning to assess the strength and reproducibility of microbiome-sex links. We predicted host sex from raw relative abundances of 30 taxa with the highest SMD between sexes. Our prediction model used a Random Forest algorithm[51] in a leave-one-dataset-out (LODO) assessment[19,52] (Fig. 2c) and considered age, BMI, and sequencing depth as baseline features. Using only the age and BMI of the individuals showed moderately predictive results (Fig. 2c, mean Area Under the Receiver Operating Characteristic (AUROC) = 0.58), while using only microbial species was substantially more predictive (mean AUROC = 0.68, Fig. 2c). The addition of age, BMI, and sequencing depth together with the microbial species provided little improvement over the species-only model (mean AUROC = 0.69), highlighting consistent cross-study microbiome-sex associations. Similar accuracy in sex prediction was achieved using profiles of KOs (mean AUROC 0.66 with and without age, BMI, and sequencing depth, Fig. 2c). The sex-attributable differences in the gut microbiome functional potential were particularly visible in two datasets (JieZ_2017 and QinJ_2012) suggesting, in these two datasets, a higher carriage of those statistically discriminant features which showed the strongest meta-analysis effect sizes (Supplementary Figure 3b).

## Meta-analysis to identify age and BMI-associated microbial features

Age and BMI are relevant to epidemiological analyses and the gut microbiome[10,53]. We analyzed adult, non-disease-associated stool samples from cMD 3 by partial correlation meta-analyses of BMI (adjusting for age, sex, and sequencing depth, $n = 6361$ from 32 datasets with BMI interquartile range (IQR) ≥3) and of age (adjusting for BMI, sex, and sequencing depth, $n = 4723$ from 18 datasets with an interquartile ≥15 years). Shannon diversity was positively associated with age (meta-analysis correlation coefficient = 0.09, 95% CI [0.06, 0.12], $P = 1.6 \times 10^{-10}$), in line with previous observations[7,54], and negatively associated with BMI (meta-analysis correlation = −0.08, 95% CI [−0.12, −0.05], $P = 7 \times 10^{-5}$) as previously reported[55]. Sixty-seven species and 24 genera were significantly correlated with age (FDR = 0.01, Fig. 3a, Supplementary Data 3), while 150 species and 45 genera were significantly correlated with BMI (Fig. 3b, Supplementary Data 4). Taxa showing the strongest correlations with age were the negatively associated *Bacteroides* species "OM05_12" (meta-analysis correlation = −0.11, 95% CI [−0.16, −0.07], Q < 0.0001) and the positively correlated *Bifidobacterium dentium* (meta-analysis correlation = 0.12,

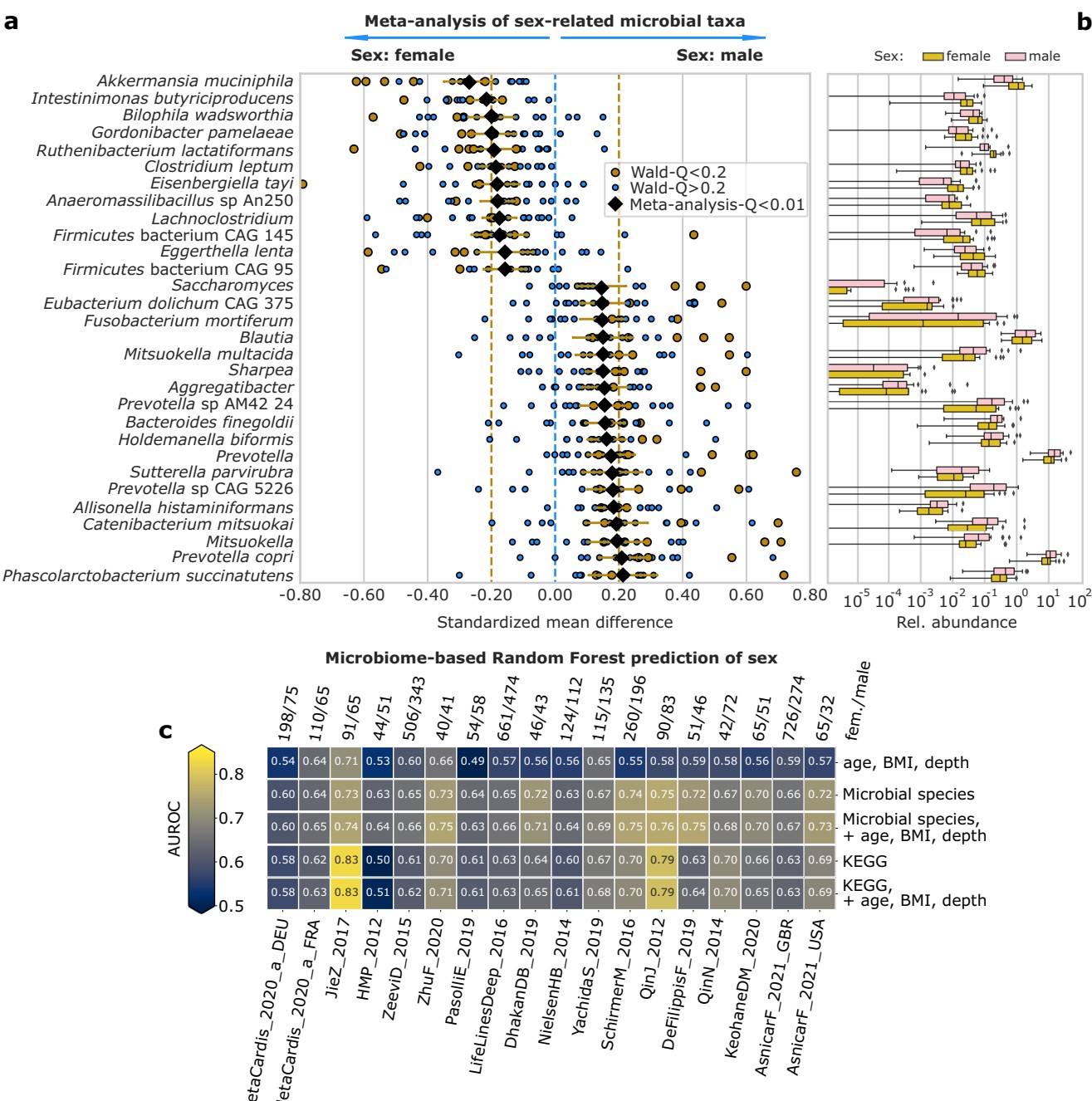

**Fig. 2 | Meta-analysis of the gut microbiome from 5505 individuals (3288 females and 2216 males) across 18 datasets, revealing sex-associated microbial differences in healthy adults based on stool samples. a** The 30 microbial species and genera with the highest standardized mean difference (SMD) meta-analysis coefficient (FDR = 0.01) between sexes. Effect sizes are calculated as SMDs from a linear model controlling for age, BMI, and sequencing depth, applied to centered log-ratio transformed species relative abundances. Yellow: significant effect size (FDR = 0.2). Light blue: non-significant effect size. Black diamonds: SMD between male and female. Yellow horizontal lines indicate the 95% confidence intervals of the effect size from the meta-analysis. **b** Mean relative abundance distribution of

the 30 taxa in the 18 datasets, grouped by sex. The *y*-axis is in the log10 scale. Boxplots span the median, interquartile range and 1.5 times the interquartile range or the most extreme value. Values outside of this range are plotted as points. **c** Area Under the Receiver Operating Characteristic of a leave-one-dataset-out validation predicting sex using a Random Forest algorithm trained on various features: (i) age, BMI, sequencing depth of the samples; (ii) species relative-abundances, with and without (iii) age, BMI and sequencing depth as features; (iv) KEGG-level-collapsed UniRef90 gene families abundances with and without (v) age, BMI and sequencing depth as features. On top: the number of female and male participants in each study.

95% CI [0.8, 0.18], *Q* < 0.0001), consistent with one previous study[7]. The two strongest correlations with BMI were the negatively associated Firmicutes species "CAG 95" (meta-analysis correlation = −0.11, 95% CI [−0.16, −0.07], *Q* < 0.0001), and the positively correlated *Gemella haemolysans* (meta-analysis correlation = 0.11, 95% CI [0.06, 0.15] and 10, 95% CI [0.08, 0.13], *Q* = 0.0001). The increased presence

of several putative oral taxa, such as *Streptococcus gordonii* in older ages and *Streptococcus mitis* in higher BMI, also agrees with the previous studies[7,56]. We confirmed the decreased abundance of *A. muciniphila* at a lower BMI[43,57], when adjusted for sex and age. We could also replicate the previously reported positive association of the *Blautia* genus with an increased BMI[55]. We confirmed, at the functional level,

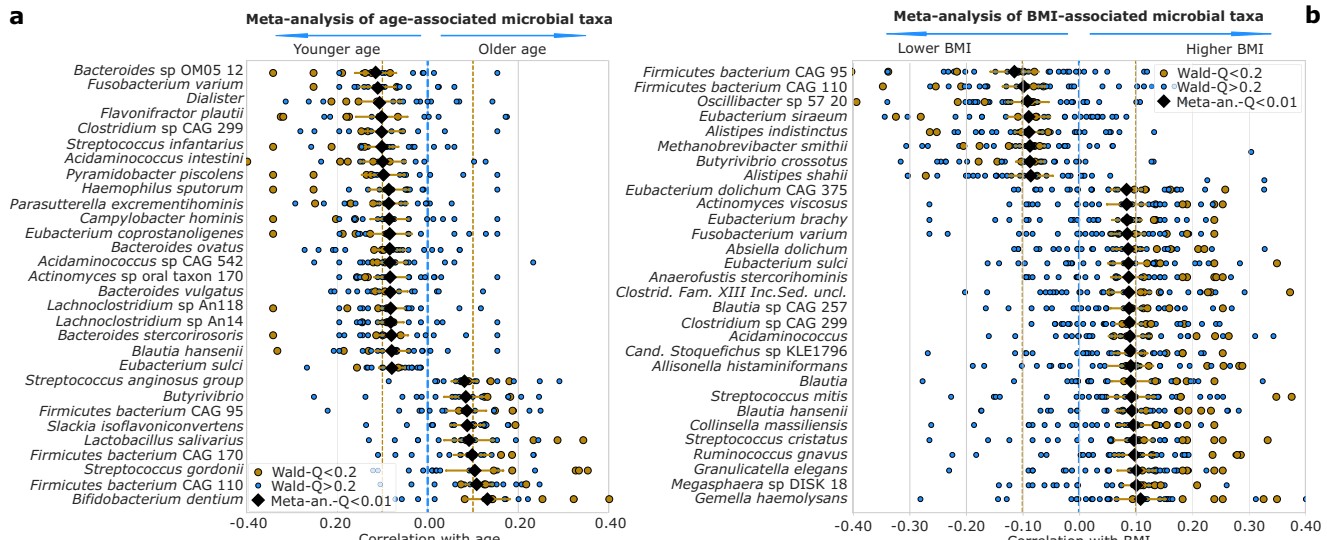

**Fig. 3 | Meta-analysis of correlation of age-related (n = 4723) and BMI-related (n = 6361) microbial species and genera. a** 30 microbial species and genera having the highest meta-analysis correlation coefficient with participant age (FDR = 0.01) with a prevalence of at least 1% in the cohort of adult control participants. Partial correlations are calculated by a linear model controlling for sex, BMI, and sequencing depth using centered log-ratio transformed species and genera relative abundances. Yellow circles: individually significant effect sizes (FDR = 0.2). Light blue circles: not individually significant. Black diamonds: meta-analysis correlation

coefficient computed as a random-effect meta-analysis on Fisher-Z transformed partial correlation. Meta-analysis coefficients and confidence intervals are then reverted back by inverse Fisher-Z transformation. Yellow horizontal lines represent 95% confidence intervals of the meta-analysis correlation. **b** 30 microbial species and genera have the highest meta-analysis correlation coefficient with BMI (FDR = 0.01), with a prevalence of at least 1%. Partial correlations are calculated as in (**a**) but controlling for sex, age, and sequencing depth.

associations between age and decrease of super pathways for the biosynthesis of vitamin B1 and B2 (meta-analysis correlation = −0.1, 95% CI [−0.13, −0.07] and −0.09 and 95% CI [−0.13, −0.06], FDR = 0.0001)[58] (Supplementary Figs. 4, 6; Supplementary Data 3). Despite the limitation of BMI as an indicator of obesity, at the functional level we also identified the *phosphatidylglycerol biosynthesis I* and *II*, previously linked with obesity[59], and the *glycogen biosynthesis I* (from ADP-D-Glucose) positively correlated with BMI (meta-analysis correlation 0.09 and 0.07, 95% CI [0.06, 0.11] and [0.04, 0.1], FDR = 0.0001), indicating a putative causative role of the microbiome in obesity (Supplementary Figs. 5, 7, and Supplementary Data 4).

## Meta-analysis identifies microbial taxa associated with multiple diseases

We performed a meta-analysis to investigate microbiome features potentially associated with health or disease. From cMD 3, we used the datasets collected from the stool and included at least 10 adults with specific diseases and 10 controls. These criteria resulted in 15 diseases (from 30 cohorts), which are: CRC, type 2 diabetes (T2D), Crohn's disease (CD), ulcerative colitis (UC), ACVD, coronary artery disease (CAD), cirrhosis, schizophrenia, Parkinson's disease (PD), asthma, migraine, soil-transmitted helminths (STH), CAD with an episode of heart failure (HF), myalgic encephalomyelitis or chronic fatigue syndrome (ME/CFS), and Behcet disease (BD). Biomarkers of "unhealthy" or dysbiotic microbiomes, which are associated with multiple diseases as opposed to being biomarkers of specific diseases, have been reported[15–17,60,61]. Here, we add evidence of such biomarkers at higher resolution. Duvallet et al.[15] analyzed 28 published case-control gut microbiome studies spanning 10 diseases to characterize disease-associated microbiome changes. However, they used only 16S rRNA sequencing data, limiting the resolution. We analyzed 4646 samples from 15 diseases, comprising 2300 cases and 2346 controls (see "Methods"). We adopted a hierarchical random-effects meta-analysis to synthesize effect sizes: at the first level for diseases studied in more than one dataset (CRC, T2D, CD, UC), and at the second level across 15 diseases (4 from meta-analyses, 11 from individual datasets).

We identified 34 microbial species associated with either health or disease (24 with health and 10 with disease, FDR = 0.01, Fig. 4a). Disease-associated species included *Streptococcus vestibularis*, *Actinomyces odontolyticus* (SMD = 0.35, 95% CI [0.18, 0.51], 0.34 [0.19, 0.49], Q = 0.001, 0.0005), and two species already linked to poor cardiometabolic health, as well as depression (*Clostridium innocuum* and *Clostridium bolteae*)[12,62] (SMD = 0.27 [0.14, 0.4], and 0.22 [0.1, 0.34], Q = 0.001, and 0.008) (Fig. 4a). The strongest association with controls was *Eubacterium ramulus* (SMD = −0.31, 95% CI [−0.46, −0.17], Q = 6.1 × 10⁻⁴), but 3 more *Eubacterium* species (*Eubacterium hallii*, CAG 38, and CAG 274), plus 4 *Roseburia* species were also among the top 20 control-associated (*R.* CAG 309, 182, 471, 303) (Fig. 4a). *A. muciniphila*, whose role in health and disease is debated[63], was not statistically associated with either one of the two groups (Q > 0.25). We performed a similar analysis not adjusting for sex. While the overall correlation between the two analyses was high (Spearman's rho = 0.93, P < 10⁻²⁰), the top 30 species differed: for example, they included the nosocomial pathogen *Streptococcus anginosus* as the top disease-associated species, suggesting a possible interaction with sex and the presence among the top 10 disease-associated species, of several CRC-related biomarkers such as *Peptostreptococcus stomatis* and *Gemella morbillorum*[19] (Supplementary Fig. 8). Our results identified *A. odontolyticus*, *Eubacterium*, and *Roseburia* as health-associated species, agreeing with a previous study that analyzed 2320 samples associated with 9 disease phenotypes[61].

We evaluated the reproducibility of our results using a multivariate method. This involved predicting health or disease states based on microbial species data. We did this for 15 distinct diseases, both individually and collectively. Our testing included cross-validation and LODO methods. Additionally, these tests were done with and without incorporating other patient-related metadata. We applied three different LODO methods: (1) used all datasets, except the test set, for training, (2) excluded the disease present in the test set from the training process, and (3) allowed the disease present in

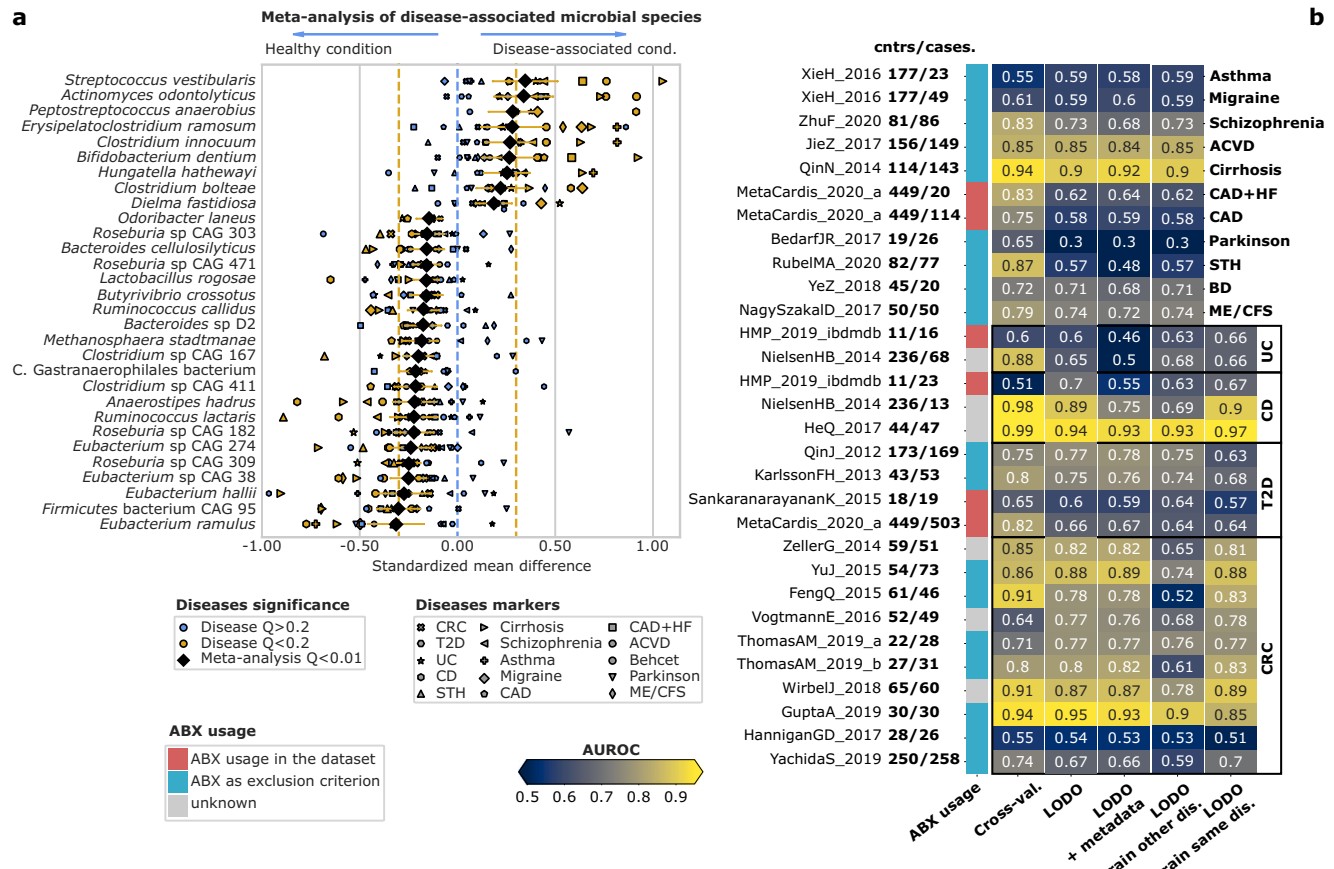

**Fig. 4 | Meta-analysis of 15 diseases and 30 cohorts reveals microbial markers of disease or health in 2346 controls and 2300 diseased patients. a** The 30 microbial species with the highest meta-analysis coefficient and FDR = 0.01 of disease-associated vs. control samples, with a prevalence of at least 1% in the cohort. Effect sizes are computed as Standardized Mean Differences (SMDs) from a linear model controlling for sex, age, BMI, sequencing depth, and usage of antibiotics in the type-2 diabetes datasets on centered log-ratio transformed species relative-abundances. Acronyms are colorectal cancer (CRC), Crohn's disease (CD), ulcerative colitis (UC), type-2 diabetes (T2D), atherosclerotic cardiovascular disease (ACVD), Behcet disease (BD), soil-transmitted helminths (STH), myalgic encephalomyelitis or chronic fatigue syndrome (ME/CFS), coronary artery disease (CAD), coronary artery disease with heart failure (CAD + HF), Parkinson's disease (PD). Effect sizes of CRC, CD, UC, & T2D are synthesized in a meta-analysis before the second meta-analysis so that these more frequently studied diseases do not dominate the results over diseases for which a single dataset is available (ACVD,

asthma, migraine, STH, cirrhosis, ME/CFS, schizophrenia, BD, CAD, CAD + HF, PD). Yellow shape: individually significant effect ($Q$ < 0.2). Light-blue shape: non-individually significant effect. Black diamonds: SMDs between cases and controls synthesized by meta-analysis. Yellow horizontal lines show 95% confidence intervals of the synthesized effect size. **b** heatmaps showing AUROCs of different Random Forest experiments on the binary discrimination "disease (generic) vs. healthy". Numbers next to study names report the number of cases and controls for each cohort. Cross validations are 10-fold repeated 10 times. Four different LODO AUROCs are shown: 1,2) models trained using all independent datasets (of the same and different diseases) as the test set, using either (1) microbial species relative abundance, or (2) relative abundance plus age, sex, BMI and depth of the sample as features; (3) models trained only on different diseases than the test set; (4) models trained only on the same disease as the test set. The cyan-red-gray bar indicates whether the samples possibly took antibiotics or this information was not available.

the test set to be included in the training. LODO AUROCs (case 1) were greater than 0.6 in 24/30 datasets (binomial $P$ = 0.001, Fig. 4b), and results in LODO validation were close to those of cross-validation (mean AUROC 0.78 versus 0.72 in cross-validation vs. LODO). Only the BedarfJR_2017[64] dataset clearly opposed the general tendency. In general, having multiple datasets relative to the same disease resulted in a higher AUROC on average mean AUROC of the diseases with multiple datasets of 0.76, vs. 0.69) when excluding the test set disease from the training (from case 1 to case 2). Switching from case 2 to case 3, hence allowing more instances of general "unhealthy microbiome" but no other instances of the same disease, resulted in an AUROC increase of 0.005, 0.1, and 0.1 for UC, CD, and CRC, respectively. On the other hand, it decreased by 0.06 in T2D (Fig. 4b). The findings of the LODO cross-disease and cross-dataset predictions (AUROC ~ 0.74) suggest the power used here is sufficient to accurately discriminate between healthy and diverse diseased phenotypes and begin to substantiate a more formal definition of the healthy human microbiome.

Eighty-nine MetaCyc[65] were associated (FDR = 0.01) with cases and 141 with controls (Supplementary Fig. 9 and Supplementary Data 5). Among the top control associated, queosine and preQ0 biosynthesis, linked to the 7-deazapurines biosynthesis potential[66] were enriched in multiple diseases including CD, CRC, cirrhosis, and other diseases from the cardiometabolic spectrum[67] (SMD = −0.32, 95% CI [−0.46, −0.18], and −0.32 [−0.49, −0.16], $Q$ = 7 × 10⁻⁷ and 1 × 10⁻⁵).

## A simple oral to gut microbial enrichment score as a quantitative measure of dysbiosis

Increased enrichment of oral microbes in the gut microbiome in response to physico-chemical alterations of the intestinal lumen in inflammation has been postulated in several diseases[19,20,22], with evidence of within-individual strain enrichment[18] However, a quantitative and conveniently calculable definition of the total accumulated amount of enrichment, based on microbiome taxonomic profiling, is lacking. We thus queried cMD3 for the oral microbiome samples (*tot. samples* = 857). We used these to

compute sets of typical oral species based on multiple prevalence thresholds. We queried cMD 3 for all the microbiomes from healthy adult populations of at least 100 individuals (*tot. studies* = 25, tot. samples = 6891) and all the microbiomes from adult, diseased individuals from populations of at least 5 individuals (tot. studies = 48, tot. samples = 3346). We used AUROC to measure the ability of the computed scores to distinguish between healthy and unhealthy populations (Fig. 5a, b). The proposed score is calculated as the summed relative abundance of oral species found in at least 1% (~9 samples) of the oral samples (*n* species = 305) in a gut microbiome sample (AUROC = 0.79, Mann–Whitney $P = 2 \times 10^{-5}$, Fig. 5b). To evaluate the ability of our score to distinguish between diseased and healthy individuals in real case-control settings, we applied it to the 30 cohorts of 15 heterogeneous diseases previously introduced (Fig. 4). OES was positively and significantly associated with cases in 9 of 10 CRC cohorts and 9 out of 20 non-CRC disease cohorts (Mann–Whitney FDR = 0.1, Fig. 5c). After adjustment for host sex, BMI, and age, it remained positively and significantly associated with disease in 7 of 10 individual CRC datasets and 8 of 20 non-CRC datasets (Wald FDR = 0.1, Fig. 5c): ACVD, CAD, CAD+HF, cirrhosis, two cohorts of CD, and one cohort of T2D. Ignoring statistical significance, the score was positively associated with cases in 29 or 30 studies ($P = 5.8 \times 10^{-8}$, binomial test) with a median AUROC for discrimination of cases vs controls of 0.65 (Fig. 5c, d). In log-linear regression meta-analysis, the OES was increased in diseases on average by threefold (rate ratio 95% CI [2.2, 4.4], meta-analysis $P = 2.6 \times 10^{-11}$, Fig. 5e, Supplementary Data 6). The score was moderately associated with age, adjusted for BMI and sex in non-disease-associated samples (rate ratio 1.008, 95% [1.002, 1.013], $P = 0.005$. The score was positively associated with age in 14 of 18 individual datasets.

Using an alternative definition of oral enrichment, calculated as alpha-diversity (Shannon entropy) of oral species, the number of positive associations was similar (26/30, binomial $P = 5.9 \times 10^{-5}$, Supplementary Fig. 10). This suggests that the simpler score computed as the sum of the relative abundances of the oral species is adequate.

## Effect of compositional data transformation

To assess the sensitivity of meta-analysis results to the choice of data transformation, we reanalyzed sex, age, and BMI using arcsine square-root transformed species relative abundances. The meta-analysis coefficients obtained from the two transformations were highly concordant (Spearman's $r = 0.9$ for sex, $r = 0.68$ for age, $r = 0.79$ for BMI; see Supplementary Fig. 11). However, the proportion of significant positive and negative associations (FDR = 0.2) between species and these variables differed substantially depending on the transformation applied. With CLR transformation, the percentage of positive meta-analysis coefficients were 67% for sex (male), 21% for age, and 76% for BMI. In contrast, the arcsine square-root transformation yielded 31% for sex (male), 82% for age, and 34% for BMI. The results for the CLR transformation are reported in the main text, while tables of meta-analysis results under both transformations are available in Supplementary Data 2 (sex), 3 (age), and 4 (BMI).

## Discussion

Metagenomic studies of the human microbiome are accumulating at an increasing pace, making meta-analysis a possibility for increasing numbers of health outcomes. Meta-analysis can help identify reproducible health-microbiome links shared by diverse populations and in the presence of heterogeneous experimental methods. When studies are conducted in different populations or settings, it's likely that the distribution of unmeasured confounding variables will differ across the different study populations. For example, confounding effects of diet are likely to differ across populations with different standard diets,

making meta-analysis a useful way to identify associations shared by different populations and not driven by diet. Similarly, while any single study may suffer bias from batch effects that are unbalanced between cases and controls, the same bias is unlikely to be shared by independent studies. Here we present cMD 3, an expanded database of uniformly processed shotgun metagenomic data from human hosts and curated metadata, that enables meta-analysis and comparison with published data across a wide range of health outcomes, particularly for fecal microbiomes.

We leveraged cMD 3 to identify associations between the fecal microbiome and BMI, age, sex, and shared by unrelated diseases. This integrative analysis expands on previous meta-analytical efforts focused on specific conditions[7,19,52,54,68–71], or considering a limited number of samples and conditions[15,17,18,61]. The effect sizes for individual taxa are generally small for age, BMI, and sex: less than 0.2 SMD, meaning for example, that the relative abundance of a taxon in one population exceeds the median of the other population in 58% of individuals instead of 50%[72]. However, with a large number of associated taxa or functional modules, it is still possible to construct reasonably accurate machine learning models, with mean AUROC in LODO independent validation in the range of 0.7. Associations with any disease state compared to healthy control participants were stronger, with the largest SMD at 0.35, mean AUROC in LODO of 0.7 across all diseases, above 0.80 for the most predictable diseases, and 0.80 across all diseases for the "oral enrichment" score. While meta-analysis can, in some situations, reduce the effects of confounding by combining populations with different distributions and effects of confounders, we cannot assess the causality of these associations beyond controlling for age, sex, and BMI. Instead, we present taxonomic and functional shifts in the gut microbiome that can be considered as replicable associations, across multiple studies in diverse locations, with age, sex, BMI, and disease. We noted that although meta-analysis coefficients were strongly correlated between CLR compositional transformation and variance-stabilizing arcsin square-root transformation, the direction of many associations differed depending on the transformation applied. Interpretation of results varies between absolute and relative abundance, but this analysis does not favor one transformation over the other.

Oral to gut transmission has been called a hallmark of disease[18], and the presence of oral-typical bacterial species in the gut has been reported to be associated with increased age[7]. In cross-sectional, stool-only studies it is not possible to directly measure oral to gut transmission, therefore, we propose a simple measure of total "oral enrichment" as the sum of relative abundances of oral-associated species (Supplementary Data 7) in the gut. Elevated levels of this score are associated with numerous adverse health outcomes when controlling for age, sex, and BMI: across a heterogeneous set of 15 diseases represented in cMD 3, the score is increased in the disease group in 29 of 30 studies ($P = 5.8 \times 10^{-8}$, binomial test, $n = 4646$, 3.2-fold increase in the diseased population). Among control samples it is associated with increased age ($n = 4723$, 1.04-fold increase per each year of age). These analyses confirm previous findings in a much larger and more diverse sample and provide an easily calculated score to further study oral microbiome enrichment in the gut in the life cycle and in disease. The association with both age and numerous diseases highlights the importance of age as a potential confounder and source of selection bias in observational microbiome studies.

Altogether, this study provides: (i) a large, manually-curated, harmonized database focusing on human stool samples but also including oral, nasal, skin, and urinary tract microbiomes, (ii) a set of cross-cohort host-microbiome associations that are less likely to be affected by confounding, lack of generalizability, or study artifacts than a single study of a more homogeneous population, and (iii) an easily calculated OES providing a quantitative measure of one type of

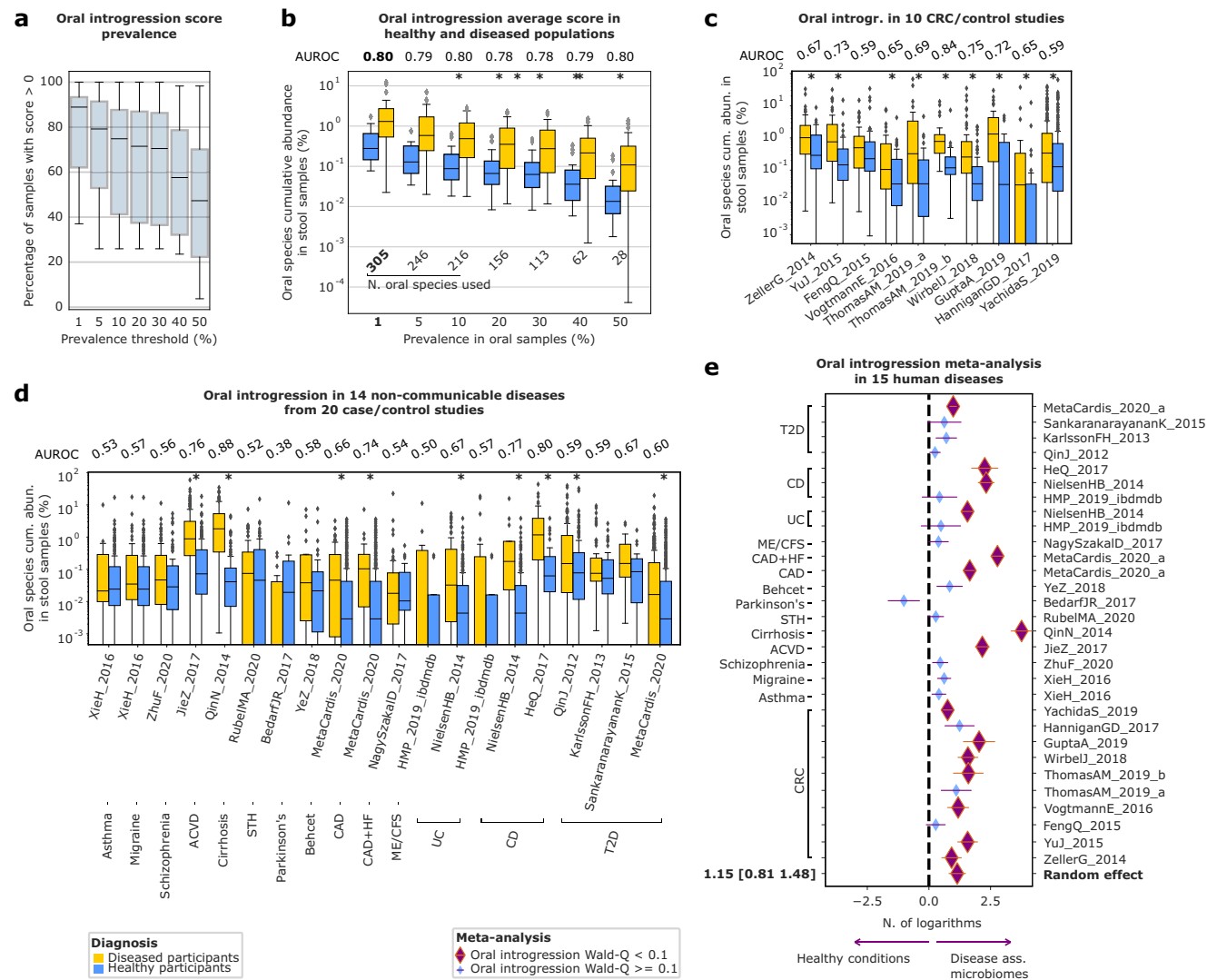

**Fig. 5 | Sum of relative abundances of typically oral taxa in the human gut microbiome is a potential indicator of disease. Per-study sample size is shown in Fig. 4. a** Boxplots showing the percentage of positively scoring individuals from a dataset of 6891 gut microbiomes from healthy, adult participants, under different definitions of the score based on progressive thresholds of prevalence to define an oral species. Boxplots span the median, interquartile range and 1.5 times the interquartile range or the most extreme value. Values outside of this range are plotted as points. **b** Distribution of the per-population mean score in 6891 gut microbiomes from adult, healthy individuals (25 studies), and 3632 gut microbiomes (48 studies) from adults who have received a specific diagnosis. Asterisks mark the between-distribution (disease vs healthy), two-sided Mann–Whitney test $P < 0.05$, exact $p$-values are reported in Supplementary Data 8. AUROC of the per-dataset average score predicting the diseased state is reported. Boxplots span the median, interquartile range and 1.5 times the interquartile range or the most extreme value. Values outside of this range are plotted as points. **c** log10 distributions of summed relative abundance of oral-cavity typical microbial species (defined using 1% of the oral samples as a threshold) in 10 cohorts of CRC patients (orange) and related controls (blue). Asterisks mark the between-distribution (disease vs healthy), two-sided Mann–Whitney test FDR < 0.1, $p$-values are reported in Supplementary Data 8. AUROCs of the oral enrichment score for

predicting CRC versus controls are presented. Boxplots span the median, interquartile range and 1.5 times the interquartile range or the most extreme value. Values outside of this range are plotted as points. **d** Boxplots showing the log10 distributions of oral-cavity typical microbial species summed relative abundance (defined using 1% of the oral samples as a threshold) in 14 diseases (20 cohorts), divided by disease (orange) and controls (blue). Asterisks mark the between-distribution (disease vs healthy), two-sided Mann–Whitney test FDR < 0.1, exact $p$-values are reported in Supplementary Data 8. AUROCs the oral enrichment score against disease versus healthy conditions are presented. Boxplots span the median, interquartile range and 1.5 times the interquartile range or the most extreme value. Values outside of this range are plotted as points. **e** Forest plot showing the meta-analysis of the association of disease and corresponding healthy controls of oral species summed relative abundance in 30 cohorts. Single datasets effect sizes are computed as mean difference (beta coefficient) extracted by a linear model controlling for sex, age, BMI, number of reads, and antibiotics usage when possible. A natural log of the score is used. Zeros are imputed using the minimum value in each dataset. Purple/gold: coefficient different from zero (Wald FDR = 0.01 in the single cohorts, meta-analysis $P < 0.05$); blue/purple: coefficient non-significantly different from zero.

gut microbiome dysbiosis that may be relevant to multiple adverse health outcomes.

This resource has limitations that reflect publicly available human microbiome data. Most data are sampled from stool, with other body sites much less represented. Africa, South America, Middle East countries, eastern Europe, southern Asia and Polynesia are under-

represented. We will continue the curation effort, which will be facilitated by community adoption of standards such as the recently introduced STORMS checklist[73]. Future versions of the database will improve its sample size, scope of body sites and health outcomes, geographical and racial/ethnic representation, and will provide greater numbers of metagenomic features from updated taxonomic profiling software.

## Methods

### Development of the cMD 3 package

We expanded the cMD package to version 3 by including human shotgun metagenomics data. Selection criteria for inclusion were sample size, availability of raw data in the Sequence Read Archive, completeness of essential sample metadata (e.g., age, BMI, sex, disease status), relevance to existing cMD 1 studies, user requests, and community interest. Versioning of cMD is aligned with bioBakery[27]. The datasets are named after the first author's surname, given name initials, and the publication year. The number of samples in each dataset may or may not match the number of samples declared in the original publication or present in NCBI, as a sample to be included in cMD 3 must have raw sequencing data for uniform processing and essential metadata. The list of the datasets in cMD 3 is available in Supplementary Data 1.

Metadata was manually curated from original literature, supplementary information, and other sources. Manually curated metadata was checked against a controlled vocabulary using an automatic grammar-checker. When new metadata is discovered - for example, a new publication makes different information explicit, our team updates datasets, ensuring cMD 3 provides the most complete and up-to-date metadata. The accessibility of per-sample metadata in original publications remains the main driver of dataset acquisition. Metadata heterogeneity and richness is judged based on the odds that the data will be employed in further studies. Datasets addressing high-interest topics (i.e., cancer therapies and non-Westernized communities) have been included regardless of their sample size, and users' requests have been prioritized.

### Metagenomic profiling

Raw whole shotgun sequencing data were processed by MetaPhlAn3 (CHOCOPhlAn version 201901) for compositional profiling and HUMAnN3 for functional profiling[27]. Samples without a MetaPhlAn3 compositional profile were excluded from the further analysis. The cMD 3 includes 22,710 samples, accounting for 2060 microbial species and annotated with over 10 million UniRef90 gene families and 651 MetaCyc microbiome-implied pathways.

### Microbiome measures

We separately analyzed five different measures of the stool microbiome: alpha diversity (Shannon entropy), species, genus, KOs, MetaCyc pathways. We removed any feature with less than 1% prevalence across the full cMD 3 dataset, and any genus represented by a single species (that genus being instead analyzed as the species). Alpha diversity was computed over species-level abundance profiles of each sample.

### Data transformation

Taxonomic, KO, and MetaCyc pathway relative abundance were centered log-ratio transformed[74,75] after adding a pseudo-zero ($1 \times 10^{-6}$, or $1 \times 10^{-10}$ for KOs)[76] for differential abundance analysis. The transformation was done using the *scikit-bio* package in Python (ver. 0.5.6).

### Inclusion and exclusion criteria

For sex, age, and BMI analyses, datasets *AsnicarF_2021* and *Meta-Cardis_2020_a* were divided into UK/US parts and Germany/France parts, respectively. Meta-cohorts for analysis of these attributes were selected based on the following common criteria: reported age greater than 16 years, data collected from stool sample, generally healthy or control in a case-control study, baseline or single time point, and reported BMI. Some additional, feature-specific selection criteria were applied:

1. For analysis of sex-associated microbiomes: at least 25% and 40 samples of each male and female, resulting in 5505 healthy samples from 21 studies.

2. For analysis of age-associated microbiomes: at least 40 samples and an IQR of age greater or equal to 15 years, resulting in 4723 samples from 18 datasets.

3. For analysis of BMI-associated microbiomes: at least 40 samples and an IQR greater than or equal to 3, resulting in 6361 samples from 32 datasets.

4. For analysis of disease-associated microbiomes: case-control studies with at least 10 cases and 10 controls after excluding individuals with pre-pathological conditions (glucose tolerance and colorectal adenoma), including only samples annotated for age, sex, and BMI. This resulted in 2300 cases and 2346 baseline controls from 30 cohorts spanning 15 diseases, including four cardiometabolic diseases.

Two studies (*HMP_2019_ibdmdb* and *NielsenHB_2014)* included samples for both Ulcerative Colitis (UC) and Crohn's Disease (CD). These diseases were analyzed separately, each with a common control group. Another study (*XieH_2016)* included samples for both asthma and migraine; these were also analyzed separately using the shared controls.

This filtering resulted in the following meta-cohorts:
Four cardiometabolic diseases (type 2 diabetes (T2D, $n = 1427$), ACVD ($n = 305$), CAD ($n = 563$), and CAD with heart failure (HF, $n = 469$))
One psychological pathology (schizophrenia, $n = 167$)
One gastrointestinal tract disease having a tumoral character (CRC, $n = 1300$)
Two gastrointestinal tract autoimmune diseases (Crohn's disease (CD, $n = 309$), ulcerative colitis (UC, $n = 346$),
One autoimmune non-gastrointestinal tract disease (asthma ($n = 200$))
One multisystem inflammatory disease (Behcet disease, BD, $n = 65$)
One liver disease (cirrhosis, $n = 237$)
One helmint disease (STH, $n = 159$)
A partially uncovered pathology[77] (myalgic encephalomyelitis or chronic fatigue syndrome (ME/CFS), $n = 100$)
A partially uncovered etiology disease that involves the brain, though not considered a nervous system disease (migraine, $n = 226$)
Parkinson's disease ($n = 45$).

### Per-dataset regression models

For each disease outcome, we fit ordinary least square linear models using the model formula:

*clr(feature abundance) ~ outcome + sequencing depth + sex + age + BMI*

Sequencing depth was included to control for the effect of pseudocount addition in clr transformation. Another single model was fit for analysis of sex, age, and BMI as outcomes, as above but without disease outcome.

### Standardized mean difference (SMD) for binary outcomes

We used SMD[78] as a scale-free measure of association that can be synthesized across multiple studies. SMD was calculated for each study from the relevant regression coefficient as:

$$SMD = \frac{t \times (n1 + n2)}{\sqrt{n + n2} \times \sqrt{n1 + n2 - 2}} \tag{1}$$

where $t$ is the $t$-score, and $n1$ and $n2$ are sample sizes for control and disease, respectively. The standard error (SE) of SMD was computed as:

$$SE = \sqrt{\frac{n1 + n2 - 1}{n1 + n2 - 3} \cdot \frac{4}{n1 + n2} \cdot \left(1 + \frac{SMD^2}{8}\right)} \tag{2}$$

## Correlation for continuous outcomes

Correlation coefficients were computed from regression tables as[78]:

$$r = \frac{\sqrt{t}}{(t^2 + n - 1)} \quad (3)$$

where $t$ is the $t$-value and $n$ is the number of samples in the control group. The correlation coefficients were Fisher-Z transformed for meta-analysis, then the inverse of the Fisher-Z function was applied to the synthesized estimate and its confidence intervals.

## Methods for all meta-analyses

We used random-effects models to synthesize SMD and partial correlation coefficients and estimated between-dataset heterogeneity using Paule and Mandel's and Cochran's Q-statistic to assess heterogeneity[79], using the Python packages *statsmodels*, *skbio*, and *scipy*.

The standard error for synthesized effect size (SEm) was calculated as:

$$SEm = \frac{1}{\sqrt{\sum_{i=1}^{k} W_i}} \quad (4)$$

where $k$ is the number of studies, and $W_i$ is the weight of the $i$th study, which is calculated as the inverse of the sum of the estimated variance of the effect size coefficient for that study plus an estimate of the between-study variance.

## Meta-analysis of diseases

Whereas microbiome associations with age, sex, and BMI were synthesized using standard meta-analysis methods described above, we employed a two-step method to identify biomarkers of disease vs. health that are not dominated by the most studied diseases. In step 1, we performed a meta-analysis of four multi-dataset diseases (10 CRC, 3 CD, 2 UC, 4 T2D) to generate a single estimate and standard error for each of these four diseases. In step 2, we performed a meta-analysis to synthesize these 4 coefficients and of 11 diseases represented by a single dataset, treating all diseases as a single outcome (i.e., "disease" vs control).

## Other statistical analyses

Beta diversity was evaluated on Euclidean distance pairwise matrices computed over the centered log-transformed relative abundances of the sex-analysis table (Aitchison distance). The significant difference in microbial composition between sexes based on the Aitchison distance was assessed by PERMANOVA and ANOSIM tests using 999 permutations and Benjamini–Yekuteli FDR of 0.1 (*scikit-bio* (ver. 0.5.6) and *statsmodel* (ver. 0.11.1) Python libraries).

## Machine learning analysis

We used the Random Forest Algorithm[80] as implemented in the MetAML Python software[60] to discriminate between males and females, and to predict disease status in patients. Consistent with previous work[27], we set the following parameters for all experiments: 10,000 trees, 1% of the features in each tree, minimum 5 samples per leaf, unlimited tree depth, and entropy as the node impurity criterion. We used raw relative abundances of microbial features without any transformations. Accuracy was evaluated using the AUROC curve, averaged over 10 repetitions.

*Sex prediction:* Using LODO cross-validation[19,52], we evaluated the model's performance on each dataset independently. We tested three feature sets: (1) Baseline: BMI, age, sequencing depth, (2) Microbial: species or KOs separately, and (3) Combined: Microbial features plus baseline.

*Disease prediction:* We applied both LODO and tenfold cross-validation on microbial species relative abundances. For LODO, we tested three training strategies: (1) Standard: all non-test datasets, (2) Disease-excluded: all datasets except those with the test set's disease, and (3) Disease-only: only datasets representing the same disease as the test set. Additionally, we applied the strategy 1 in two feature groups: (a) microbial species alone, and (b) microbial species plus sex, age, BMI, and sequencing depth.

## Definition of the oral enrichment score

We defined the OES using cMD 3 data from 857 oral cavity samples (body_site:oralcavity). We tested various thresholds of minimum prevalence in these samples to define oral-associated species: 1%, 5%, 10%, 20%, 30%, 40%, and 50%, generating correspondingly smaller signatures of oral-associated species. The OES was defined as the sum of the relative abundance of oral species in this signature. Alternatively, we examined using Shannon entropy calculated on these oral species as an alternative score.

We then queried cMD 3 for all gut microbiomes (body_site: stool) from individuals 12 years of age or greater (age_category being any of: adult, senior, schoolage) at baseline (days_from_first_collection: either 0 o NA). We created:

1. a meta-dataset of individuals specifically enrolled as controls (study_condition: control), resulting in 6891 samples from 25 datasets of at least $n = 100$.
2. a meta-dataset of individuals who have received a diagnosis for some disease (study_condition: not control), resulting in 3632 samples from 48 datasets of at least $n = 5$.

We calculated the OES for each threshold in both meta-datasets. We then compared the two distributions of scores using the Mann–Whitney test and the AUROC of the score's ability to discriminate between control and non-control groups. Additionally, we aimed to maximize the number of individuals in the overall set of 10,523 with a non-zero score. We selected the score at the 1% prevalence threshold (defined by 305 oral species) for the following reasons:

1. The distributions of rank of the score differ significantly between cases and controls (Mann–Whitney $P < 0.05$).
2. It had the highest AUROC value for discriminating cases vs controls (0.80).
3. More than 80% of individuals had a non-zero score.

## Evaluation of the oral enrichment score

We evaluated the OES in case-control settings by analyzing the preselected set of 15 diseases, 30 cohorts, and 4646 stool samples described above. We used three methods of evaluation:

1. Mann–Whitney tests in each dataset to test the null hypothesis of identical rank distribution of the score in case and control groups, adjusting for multiple hypothesis testing by the Benjamini–Yekuteli procedure (FDR = 0.1).
2. Binomial test of the null hypothesis that the score is equally probable in each dataset to have a higher mean in cases or controls.
3. Multiple linear regression of log-transformed score plus pseudo-count of 0.0001 against disease status, controlling for sex, age, BMI, number of reads, and antibiotic usage. Since microbiome studies tend to exclude individuals with antibiotic exposure, if there was no specific information on antibiotic usage, we assumed the sample has no antibiotic exposure. Only when antibiotic exposure history was provided, we adjusted our regression model against it. Model coefficients of each dataset were synthesized by meta-analysis as above.

## Software availability

The cMD 3 command-line interface is available at https://zenodo.org/records/17498288.

The scripts to reproduce the analyses presented in this paper are available as R vignettes and Python Jupyter notebooks at https://zenodo.org/records/17498251. For multi-dataset diseases, we replicated the meta-analysis in R, using the *compositions* package for clr transformation[81] and the *metafor*[82] package for meta-analysis.

Manually collected and curated metadata provided by the cMD 3 package are available at https://zenodo.org/records/17498348. Metadata fields with a description of the data type and allowed values are available at inst/extdata/template.csv in this repo.

Guidelines for metadata curators are available at https://github.com/waldronlab/curatedMetagenomicDataCuration/wiki.

## Data availability

The cMD 3 package is available through Bioconductor at https://doi.org/10.18129/B9.bioc.curatedMetagenomicData. An additional dataset, HeQ_2017, was utilized in our analysis but is not yet included in cMD 3. The raw dataset and pre-merged MetaPhlAn3 and HUMAnN3 tables are available on Zenodo (https://doi.org/10.5281/zenodo.17567900) along with tutorial scripts on how to integrate the data into the cMD3 framework.

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

## Acknowledgements

This work was funded by the National Cancer Institute of the National Institutes of Health (R01CA230551 to LW).

## Author contributions

Conceptualization and Methodology P.M., N.S., L.W. Data curation P.M., G.A., L.S., D.G., A.W., C.M., K.L., K.G.P., G.P., S.D.G.T., A.B., G.D.A., R.A.,

K.E., F.Z., V.G., M.K., A.P., I.L., S.E., M.V.C., S.T., F.A., H.J. Resources and Code P.M., G.A., M.R.P., L.G., S.D., V.C., A.B.M., F.B., A.M.T., M.Z., E.P. Formal analysis P.M., G.A. Funding acquisition L.W. Project administration L.W., S.O. Supervision L.W., C.H., N.S. Validation G.A. Writing – original draft P.M. Writing – review & editing G.A., P.M., L.W., S.O., N.S.

## Competing interests

The authors declare no competing interests.

## Additional information

Paolo Manghi [1,14,15], Giacomo Antonello[1,2,15], Lucas Schiffer [3,4], Davide Golzato[1], Andres Wokaty[2], Francesco Beghini [1], Chloe Mirzayi[2], Kaelyn Long[2], Kai Gravel-Pucillo[2], Gianmarco Piccinno [1], Samuel David Gamboa-Tuz [2], Arianna Bonetti [1], Giacomo D'Amato[1], Rimsha Azhar[2], Kelly Eckenrode[2], Fatima Zohra[2], Valentina Giunchiglia [1,5,6], Marisa Keller [1], Anna Pedrotti[1], Ilya Likhotkin[7], Shaimaa Elsafoury [2], Ludwig Geistlinger[2], Aitor Blanco-Miguez [1], Andrew Maltez Thomas [1], Moreno Zolfo [1], Marcel Ramos [2], Mireia Valles-Colomer [1], Sabrina Tamburini [8], Francesco Asnicar [1], Heidi E. Jones[2], Curtis Huttenhower [9,10], Vincent Carey [11], Sean Davis [12], Edoardo Pasolli [13], Sehyun Oh [2], Nicola Segata [1,8,16] ✉ & Levi Waldron [1,2,16] ✉

[1]Department CIBIO, University of Trento, Trento, Italy. [2]Graduate School of Public Health and Health Policy and Institute for Implementation Science in Population Health, City University of New York, New York, NY, USA. [3]Section of Computational Biomedicine, Boston University School of Medicine, Boston, MA, USA. [4]Graduate Program in Bioinformatics, Boston University, Boston, MA, USA. [5]Department of Biomedical Informatics, Harvard University, Boston, MA, USA. [6]Department of Brain Sciences, Imperial College, London, UK. [7]Institute of Pharmacy and Molecular Biotechnology (IPMB), University of Heidelberg, Heidelberg, Germany. [8]IEO, European Institute of Oncology, IRCCS, Milan, Italy. [9]Harvard T.H. Chan School of Public Health, Boston, MA, USA. [10]The Broad Institute of MIT and Harvard, Cambridge, MA, USA. [11]Channing Division of Network Medicine, Mass General Brigham, Harvard Medical School, Boston, MA, USA. [12]Department of Biostatistics and Informatics, University of Colorado Anschutz School of Medicine, Aurora, CO, USA. [13]Department of Agricultural Sciences, University of Naples, Naples, Italy. [14]Present address: Research and Innovation Center, Fondazione Edmund Mach, San Michele all'Adige, Italy. [15]These authors contributed equally: Paolo Manghi, Giacomo Antonello. [16]These authors jointly supervised this work: Nicola Segata, Levi Waldron. ✉e-mail: nicola.segata@unitn.it; levi.waldron@sph.cuny.edu

