## [Transparent Peer Review file · Nature Communications]

Meta-analysis of 22,710 human microbiome metagenomes defines an oral-to-gut microbial enrichment score and associations with host health and disease

Corresponding Author: Professor Levi Waldron

Version 0:

Reviewer comments:

Reviewer #1

(Remarks to the Author)

Summary:

Manghi et al. presented a meta-analysis of human microbiome and identified bacterial taxa associated with age, sex, BMI and diseases. They further developed a simple approach to calculate the percentage of oral-typical bacteria in the fecal samples. This number has been shown elevated in patients with a number of diseases.

Major comments:

This manuscript seems to update their previous human metagenomic database published in Nature Methods 2017. I was not aware of this data collection before but the compilation should be very helpful to the community. Although it is highly skewed towards stool samples, the data collection includes samples from other body sites such as oral cavity, skin, and vagina. I followed the instructions and was able to download count and metadata tables using the command line tool provided by the authors. Therefore, I am convinced that the metagenomic resources provided by the authors are accessible to broad microbiome researchers and useful for addressing their own questions.

In contrast to the useful resource which I like a lot, the scientific part of this paper is much less interesting. The whole microbiome field is full of association studies between bacterial taxa and host phenotypes and the variables studied here (sex, age, etc.) are not new. The authors mentioned that "cMD has increased statistical power", which is true, but I am not fully convinced that redoing statistical analysis using a larger but heterogenous number of cohorts will give us new insights. Are there such examples showing that statistical significance is only possible when using multiple cohorts together but not in each individual cohort? Since the manuscript neither validated nor generated a new hypothesis, I would think of these analyses mostly confirmational of previous findings.

One of the highlights is that the authors developed an oral introgression score based on the total relative abundance of a pre-defined list of species highly abundant in oral cavity. This is great but unfortunately, I do not see any validation of this approach. People have used a different approach by comparing 16S sequences to Human Oral Microbiome Database (Coker, O. O., Dai, Z., Nie, Y., Zhao, G., Cao, L., Nakatsu, G., Wu, W. K., Wong, S. H., Chen, Z., Sung, J. J. Y. & Yu, J. Mucosal microbiome dysbiosis in gastric carcinogenesis. *Gut* 67, 1024–1032 (2018).). I am wondering how much difference in oral bacterial percentage will be found between the two approaches. Also, some species like *Prevotella* spp. can colonize both gut and oral cavity. Since the approach has a species-level resolution, I am not sure if the authors can properly assign a habitat for those species solely based on the species identity. Do you think that sequence-level approach will work better? Finally, it is important to show that, using the new approach, the oral bacterial fraction in the feces of healthy volunteers is low (at most a few percent). This is a sanity check: if any of the bacterial species in the list colonize both gut and oral cavity, the total fraction of oral bacteria in feces will be inflated even in healthy people. Typically, it does not make too much sense to observe >10% of oral bacteria in healthy people.

Another important but easy-to-fix drawback is that the statistical association between oral introgression score and disease status did not take antibiotics as a confounding variable. Antibiotic is the most important variable that could lead to oral

bacterial enrichment in feces. The apparent associations with disease status may be caused by the use of antibiotics and it would be critical to know if disease remains associated with oral bacterial percentage even after excluding the effects of antibiotics.

Minor Comments:

1. Abstract: "...to address several ongoing challenges...". What are those challenges?
2. Third line of introduction: phisiology should be physiology.
3. Random forest algorithm: "The parameters were set in advance based on our previous experience with this algorithm." The parameters are best searched on a dataset-by-dataset basis. My experience is that, 1,000 trees is a bit low and it is better to set a tree depth to avoid overfitting. It is always a good practice to set a high number of trees such as 5,000 or 10,000 and uses grid search for a few key parameters as mentioned in the manuscript.

Overall, I am not impressed by the scientific story of this paper. It is technically sound in general but has some flaws in the analysis of oral bacteria in feces. From my opinion, this is best considered as a resource paper: all of the statistical analyses included in the manuscript confirmed previous findings and showcased how this wonderful dataset can be reused to identify clinical associations.

Reviewer #2

(Remarks to the Author)

The manuscript by Manghi and colleagues describes meta-analyses of human microbiome associations with sex, age, BMI, and disease states, and a cross-study meta-analysis of introgression of oral microbes into the gut. Notably, as part of this study, the authors have substantially updated the curatedMetagenomicData R package, greatly increasing the number of samples. The amount of work that goes into curating these data is laudable. The authors similarly present a considerable body of data resulting from their meta-analyses. However, several caveats need to be addressed for these findings to be helpful to the microbiome community and beyond.

Major issues

1. Throughout the manuscript, the authors extensively use the definition "meta-analysis". A key component of such an analysis is an accurate description of the search criteria and keywords for relevant publications, inclusion and exclusion criteria for studies, and a flow diagram of how many studies were lost in each step for a given reason. In the methods section, the authors do provide the filtration criteria for the various comparisons (age, sex, BMI), however I could not find the criteria for including or omitting a study from the whole database. It seems like many studies dissecting the microbiome in human disease are missing, including those included in previous microbiome meta-analyses (of UC, CD, T2D), and it is unclear why. This could potentially impact the conclusions drawn by the authors throughout the manuscript.
2. It is unclear from the map as currently presented in figure 1 whether each point is simply the location of the study's primary institute, or relates to the geography of the human populations analyzed in the study. The latter is far more relevant than the first, and the figure should be adjusted to reflect the geographic distribution of the human populations included in each study. Much of the text in this figure could be relegated to the main text (or is indeed a repetition of it already), and visualizations could be added for a better summary of the breadth of studies included in the package. With the dominance of populations in highly developed countries in microbiome datasets, it would be helpful to better understand the representation of diverse groups in this new package, for example see the plots in <https://pubmed.ncbi.nlm.nih.gov/35167588/>.
3. While the authors provide a considerable amount of data in this manuscript, interpreting the results in an insightful manner is limited by several factors detailed in the following points. Notably, each of the analyzed parameters (sex, age, BMI, some of the diseases including CRC and T2D) was previously included in at least one microbiome-focused meta-analysis. While some of these works are cited in the manuscript, there is little discussion of the current findings in context of previous meta-analyses. Most importantly, do the meta-analyses agree? are there new microbial signals that were not previously detected? Or some that may be attributed to noise or confounders? And which components of the authors' approach underlie these differences?
4. Functional analyses: The functional genes presented in the supplementary figures throughout the paper should be filtered or caveated - many of them make little sense in the context of the microbiome, are primarily eukaryotic, or represent universal bacterial proteins (see the number of universal ribosomal proteins in Figure S5 - how should such a result even be interpreted?). Further, uninterpretable pathway codes, e.g. "PWY-5136, PWY-5138, PWY-7288, PWY66-391" should not be listed in the main text.
5. Throughout the manuscript, the authors should better connect key results with biological interpretation. For example, they say "Many of the taxa (species and genera) identified as negatively correlated with BMI and positively with age are known for being short-chain fatty acids producers": but which SCFAs do they mean? How do they know which taxa are truly SCFA producers, a trait that varies dramatically between species and strains? There is another example of this: "Female-associated microbiomes contained increased potential for tryptophan-derived indole synthesis... in line with the increased abundance of *A. muciniphila* (Chappell et al., 2016)" But the reference provided does not clearly support this statement.
6. In all figures, it is crucial to visualize or list the n of each study, as the n will have a large impact on whether a result in a study is significant or strongly predicted in the ML analyses. Which studies were class balanced? How are class imbalances (particularly among diseases) handled by the authors?
7. As currently presented in the manuscript, the meta-analysis of sex differences in the microbiome is largely uninformative. These differences can be due to behavioral, social, and dietary differences between sexes as opposed to biological differences. It is unclear what the value of the work as presented therefore is. The authors note that their result indicates that it may be important to control for sex in a microbiome study of disease (they point out *Akkermansia*) - however this is surely

already well-established. This point could perhaps be better made in their disease analyses, where they could demonstrate that controlling for sex across studies significantly changes results and interpretation of the studies.

8. In figure 2, the predictive effect is strikingly stronger in two studies, regardless of which features are included in the analysis. The predictive effect in these two studies is stronger than in any other comparison. The authors should discuss the possible explanations for this.

9. Can the authors connect their results in BMI and age to existing literature? For example, what about the Bacteroides:Firmicutes ratio? Alpha diversity and BMI or age? Throughout the manuscript, there is a lot of presentation of results without biological interpretation of these results, or sufficient discussion of how these results relate to what is already known in the literature. Further, the limits of BMI as a health indicator should be noted.

10. What other metadata exists for each study beyond age, sex, and BMI? Is dietary information available for any? There should be a supplementary figure (or perhaps worked into Figure 1) describing the extent of the metadata that was compiled for these studies.

11. Summarizing some of the above points regarding age, sex, BMI – in my opinion, the interesting parts of this manuscript are the diseases meta-analysis, and the oral to gut introgression analysis. However, the reader is presented with a considerable amount of data with limited contribution to knowledge before reaching these parts. It will be helpful if rather than these three sections, the authors would focus on one, for which they have the most relevant studies, and sufficient metadata to address the aforementioned confounders, and discuss it in context of previous works, as suggested above.

12. The methodology used to interrogate the microbiome in this work were MetaPhlAn3 for bacterial taxonomy and HUMAnN for functional gene content. The basis of these methods are sound, but limited. The work would be substantially improved with the inclusion of different metagenomic methodology, particularly for the purposes of verifying important conclusions. Furthermore, it would be helpful to address some of the technical variation between studies and how it affected the results. For example, information regarding differences in sequencing platforms should be included in the metadata, and integrated into the analysis.

13. The disease analysis could potentially be the promising part of the manuscript, but it has several conceptual flaws. First, there are only 25 cohorts included in the figure, out of the original 90 included in the study. First, this should be clearly state in the main text. Furthermore, the authors should better describe in the main text the exclusion criteria that resulted in the loss of the majority of studies. Finally, considering that all the meta-analyses in this study focused on a subset of the original 90 studies, figure 1 is somewhat misleading. Notably, it is confusing as to why so many metagenomic studies of the analyzed diseases are missing.

14. Diseases represented by a single study should not be included in this figure, as in this case this is not really a “meta-analysis”. Inclusion of several conditions where the microbiome is almost certainly not causal (e.g., Soil-transmitted Helminths) seems unwarranted.

15. Most importantly, it makes little sense to analyze all diseases in this figure together, as their etiology is almost certainly different, and causal microbial associations almost certainly differ between them. The “cross-disease” signal demonstrated is almost certainly a confounding signal - for example, increased use of antibiotics, hospital-exposure, or dietary restriction among sick individuals. It is possible to see this confounding effect in the data, as *Streptococcus anginosus*, a nosocomial pathogen is the most significant ‘disease-associated’ species. Indeed, for these reasons the cross-disease signal may actually represent the least important species for microbiome causality for any individual disease. This analysis would have to be reworked, or at the least controlled for antibiotic-usage and hospitalization between patients.

16. Figure 5 / oral introgression: Why are p-values used here instead of q-values? The results should be corrected for the number of hypotheses tested (across the 25 studies in this case).

17. The lack of paired-sample analysis (e.g. analysis of oral and gut samples from the same patient) in this meta-analysis as done in Schmidt et al. 2019 limits interpretation. The authors say “but a quantitative, reproducibly calculable definition is lacking” - but Schmidt et al. 2019 presented a quantitative definition for oral introgression: are the authors able to describe how many paired samples they have across their datasets? Further, discussion of interpretation of oral introgression is lacking. Is this effect simply identifying a decrease in healthy gut anaerobes and therefore an increase in transient oral microbes passing through the gut?

Additional comments

Figure 2: The meta-analysis plots in this Figure and Figure 3-4 are extremely busy and difficult to interpret. Many points are overlaid upon each other, and the horizontal bars are almost impossible to see. These figures need to be altered for readability.

AUC should be written as AUROC throughout the text.

Reviewer #3

(Remarks to the Author)

The article “Meta-analysis of 29,533 human metagenomes defines an index of oral to gut microbial introgression and associations with age, sex, BMI, and diseases” by Manghi and colleagues reanalyzes existing microbiome data sets, and performs several supervised correlation analyses across the different individual studies comprised in the latest iteration of the curated Metagenomic Data set version 3.

The paper presents incremental new insights into the associations between age, sex, BMI and several disease conditions with bacterial populations in stool samples. Furthermore, the authors propose the sum of 130 bacterial species’ relative abundances as a novel microbiome health score.

I found the paper of modest interest. I largely believe that most analyses were conducted correctly, but some of the applied statistical analyses could benefit from a better justification and explanation. I found the microbiome health score unconvincing. Overall, novel biological insights are limited. The paper could benefit from editorial input regarding clarity, the many very long sentences, and sometimes off-putting formatting.

The term “microbial introgression” is used in the title and name of the microbiome health score. The term “introgression” is unfamiliar to me in a microbiome context, the provided references discussing oral microbiome taxa in the gut use “transmission”, which seems more appropriate, and the term introgression has different meanings elsewhere. Perhaps the authors can at least define what they mean by it, or better still, consider replacing with a more appropriate term: they mean that DNA of oral species are found in the stool.

I have a few specific comments:

L66: “increase” relative to what?

L70: the sentence is too long, and the “or to the beneficial role...” part is phrased confusingly in the context of the rest of the sentence.

Figure 2: How did you standardize? Did you z-score the CLR?

Figure 2: what are “cases” and “controls” in an analysis of “sex”?

Figure 2: what are raw abundances? The figure axis says log2, the caption log10

L657: “raw abundances were scaled to the 0 1 range” --- Did you scale within a taxon (i.e. min max scaling) or per sample? In the latter case, did you include a “other taxa” category as a sum of the excluded taxa, and if not, how do you justify the CLR transform? In the former case of “min max” scaling, why apply the CLR?

L189: where is the figure presenting these results? Besides significance, what is the meaning of the effect sizes of difference that seem very small between the sexes.

L222: Related to the above, given that the authors largely find male and female microbiomes to be very similar, would perhaps the opposite conclusion, that sex is not an important variable in microbiome studies, be justified? Can the authors discuss this better for future reference?

L277: What does “correlations were synthesized” mean?

L282: if diversity goes up with age, would therefore a negative association of many species with age be expected due to compositional effects?

L330: rephrase “a hallmark of good health” cannot be established with the current study

L341: the model setup should be described in more detail in the methods

L379: rephrase to “health associated microbiome”

Figure 5: please resolve colors in a figure legend

L 413: why write “cumulative” instead of “sum”?

L418: 3/10 and 5/15 are not very convincing numbers. Can the authors perhaps create a randomized control score for comparison? E.g. multiple times, select 130 other taxa with somewhat similar abundance profiles to the 130 oral species, and count in how many disease studies these control scores are significant predictors? This could help the authors convince the reader of the impact of their novel score and finding.

L419 and L431: What is the difference between the data from the 15 non CRC disease cohorts (L419) and 12 diseases (L432)?

It was not clear to me if this paper is the official release publication of cMD 3. Please clarify.

Version 1:

Reviewer comments:

Reviewer #1

(Remarks to the Author)

The authors have done an excellent job to address my concerns. I have no further questions.

(Remarks on code availability)

Reviewer #2

(Remarks to the Author)

The revised version of the manuscript addresses nearly all my comments, and I thank the authors for their additional work. It would be valuable if the authors could address these remaining comments.

1. Regarding my first comment requesting that the authors define the inclusion and exclusion criteria for datasets in this study. The authors now include additional details, but they are still somewhat vague. For example, the authors note that sample size was an inclusion criterion – but what was the cut-off? What was the minimal sample size for inclusion? For the criteria listed in this section, specific numbers should be given where possible.
2. The authors have addressed most of my comments related to figure 1. Minor comment, the X-axis title in figure 1B is unclear, consider “Top 30 metadata present in...”.
3. Regarding my comment (#8) on the strong impact stemming primarily from two studies, the authors say that this is likely due to having ‘highly diseased’ and ‘particularly healthy’ individuals in these cohorts, but is there any evidence for this in the metadata? Or are they hypothesizing this to be the case? Please clarify in the text.

(Remarks on code availability)

Reviewer #3

(Remarks to the Author)

The authors have made a commendable effort in compiling and meta-analyzing multiple datasets. However, while the execution has improved, it still offers only relatively few novel biological insights beyond what has already been shown in previous studies. A score calculated from 305 species identified from 857 oral metagenomes is far from straightforward.

Easy access to curated data is crucial, and regular updates are valuable. The current submission therefore adds incremental value over the authors’ own prior work and that of others. For example, gmrepo 2, published recently, also analyzed the majority of existing microbiome datasets in a unified manner across disease indications, and it provides an online interface for exploration (PMID: 34788838). It is surprising that the authors have not cited this recent work, nor its earlier version. Proper referencing is essential for situating this study within the existing body of literature, and I recommend that the authors address this throughout the manuscript (see other, specific examples below).

The term “microbial introgression” remains a poor choice. The term introgression elsewhere means transfer of genetic material from one species into the gene pool of other species, not the detection of sets of DNA sequences in stool. Furthermore, they cite three studies in their rebuttal to this critique. The authors cite the same references in both their rebuttal and the revised manuscript, but none of these sources use the term “introgression.” As such, they do not support the use of this term in the context of the manuscript.

The authors acknowledge that the enrichment of oral bacteria during dysbiosis has been studied in previous work, but limit their literature references to outdated sources. They do not seem to be aware of Franklin et al, who (arguably more appropriately) termed the detection of oral cavity microbes in stool “oral stool microbiome coalescence” (PMID: 35433514). Importantly, Chen et al. (2024) in Nature Microbiology studied oral bacterial enrichment in stool samples as a marker for dysbiosis using mouse models and through the re-analysis of paired human oral and stool samples across diseases. Why do we need this again, and why do the authors insist on using a term that is signaling incorrect biology? <https://www.nature.com/articles/s41564-024-01680-3>.

Regarding my technical comments from the first round, the authors explain that they standardized the features of the CLR-transformed data. In other words, they added pseudo counts, log-transformed geometric mean-weighted relative abundances, and then z-scored those log-transformed values. By this point, all zeros in the sequencing data have been converted into non-zero figures that are difficult to interpret. We have seen from the tumor microbiome field that dealing with low biomass, such as oral DNA in stool, can be particularly problematic. There are inherent risks in performing extensive data manipulations early on in the analytical process, especially without thorough robustness testing. The authors should consider testing their key findings with analyses that remain closer to the raw data and are robust to these various transformations. This would be satisfied with simple distribution plots of untransformed data (e.g. raw / unmodified relative read counts) that clearly support their biologically most meaningful statistical findings.

In reply to my question “if diversity goes up with age, would therefore a negative association of many species with age be expected due to compositional effects?”, the authors argue that “since we compute linear models after having performed Centered Log-ratio Transformation, we do not expect these results to be driven by compositionality, even in the presence of dominant species.” I remain unconvinced by this argument. While CLR and related techniques help address compositionality in certain contexts, they are not a cure-all for the inherent limitations of sum-constrained data. Consequently, an increase in diversity, which requires a more even distribution when the number of features is fixed, will likely result in a higher chance of negative correlations between CLR-transformed abundances and age.

Overall, while the paper offers limited novel biological insights, the well-curated data adds value, and the study could be a useful resource for the community. I recommend publication in a technical journal focused on providing such resources.

(Remarks on code availability)

Version 2:

Reviewer comments:

Reviewer #2

(Remarks to the Author)

Thank you for the additional work. All of my comments have been adequately addressed.

(Remarks on code availability)

Reviewer #3

(Remarks to the Author)

The authors have addressed my concerns about data transformations by applying an arcsine transformation and demonstrating high correlation of inferred coefficients compared to the CLR method (rebuttal fig 1), which is an appropriate step. Their analysis suggests consistency in the ranking of taxonomic associations.

However, while this provides some evidence of robustness, inspecting the correlation plot suggests a potential shift in the directionality of many taxonomic associations with age. It appears that there are now more positive correlations with age whereas the authors report more negative associations for CLR transformed data in the revised manuscript. The authors should clarify this, and consider reporting results from both analyses, with numbers of positive versus negative associations for both transformations. Fortunately, the strongest reported associations appear unaffected, which is reassuring. Adding this clarification would enhance the work by introducing appropriate caution regarding less stable associations implicitly, without requiring further analyses.

Lastly, regarding terminology, I understand where the authors are coming from. They observed an enrichment of reads from oral taxa in fecal DNA. This can arise from any number of dynamic processes, none of which can be elucidated by the current work. As such, coining a novel term with dynamic implication does not seem appropriate. The authors argue that: “what we are sure to observe is the invasion phase, not much about the coexistence of communities (or parts of them) in the gut.” Since the associations are from static sequencing data, and not derived from time series analyses or experiments, this is an overinterpretation. “Oral taxon enrichment score”, rather than any process related terms, such as coalescence or translocation, could be adequate for the present work.

(Remarks on code availability)

To the reviewers of “**Meta-analysis of 22,710 human metagenomes defines an index of oral to gut microbial introgression and associations with age, sex, BMI, and diseases**”

We thank the reviewers for their time and detailed feedback, and for everyone’s patience during the unusually long time that, for reasons I won’t get into, this revision took. We have responded to all points raised by the reviewers, updated the analysis to incorporate substantial expansion of curatedMetagenomicData, and edited throughout to improve the clarity of the manuscript. The most significant points raised by the reviewers were:

- Reviewer 2, question 1, who noted that some important datasets have been excluded
- Reviewer 2, question 4, who criticized the functional analysis for lack of clarity
- Reviewer 1, minor comment 1, who noted that the random forest algorithm have been carried out with a small number of estimator trees (1,000)
- Reviewer 3, who is concerned about the normalization and transformation for compositionality issues

We detail our revisions in response to this feedback below, briefly summarized as:

- Adding 2,177 samples from 4 studies (+10.6% growth in the cMD database)
- Re-doing all analyses with these additional studies (including 3 more diseases and 2 more cohorts of type-2 diabetes and Crohn’s disease) - both the meta-analyses and ML (increasing RF trees to 10,000)
- The disease and the oral-to-gut introgression meta-analyses are now adjusted by antibiotics usage when possible. These were previously adjusted only by sex, age, BMI, and number of reads.
- Rearranging the figures as suggested, including stricter FDR control ($FDR < 0.01$), adjustment for antibiotic usage when possible, and new sensitivity analysis and validation of the oral-to-gut introgression score
- Refactoring our oral to gut introgression score paragraph.

REVIEWER COMMENTS

Reviewer #1 (Remarks to the Author):

Summary:

Manghi et al. presented a meta-analysis of human microbiome and identified bacterial taxa associated with age, sex, BMI and diseases. They further developed a simple approach to calculate the percentage of oral-typical bacteria in the fecal samples. This number has been shown to be elevated in patients with a number of diseases.

Major comments:

REVIEWER #1 COMMENT 1

This manuscript seems to update their previous human metagenomic database published in Nature Methods 2017. I was not aware of this data collection before but the compilation should be very helpful to the community. Although it is highly skewed towards stool samples, the data collection includes samples from other body sites such as oral cavity, skin, and vagina. I followed the instructions and was able to download count and metadata tables using the command line tool provided by the authors. Therefore, I am convinced that the metagenomic resources provided by the authors are accessible to broad microbiome researchers and useful for addressing their own questions.

We are glad the reviewer thinks that our resource might be of public utility to the scientific community. We acknowledge that cMD 3 is composed primarily of stool samples, reflecting the outsized fraction of the human microbiome literature that studies of stool represents. However, as more datasets are becoming available especially for the oral, skin, and vaginal bodysites, we are committed to alleviating this bias towards stool samples.

REVIEWER #1 COMMENT 2

In contrast to the useful resource which I like a lot, the scientific part of this paper is much less interesting. The whole microbiome field is full of association studies between bacterial taxa and host phenotypes and the variables studied here (sex, age, etc.) are not new. The authors mentioned that "cMD has increased statistical power", which is true, but I am not fully convinced that redoing statistical analysis using a larger but heterogenous number of cohorts will give us new insights. Are there such examples showing that statistical significance is only possible when using multiple cohorts together but not in each individual cohort? Since the manuscript neither validated nor generated a new hypothesis, I would think of these analyses mostly confirmational of previous findings.

We thank the reviewer for the criticism. We believe that this analysis fills the need of a more comprehensive analysis across many published datasets, and that heterogeneity of the included cohorts is in this analysis a strength for allowing identification of associations preserved across diverse settings.

Observational studies of association are always subject to risk of selection bias and confounding, but meta-analysis from diverse populations can mitigate this bias if the distribution of confounding variables, or their effects on exposure or outcome, vary between the populations included in meta-analysis. The meta-analyses of taxonomic associations with sex, age, and BMI provided in this paper will provide an ongoing reference for these widely-studied associations in the largest, most diverse sample available to date.

The reviewer is correct to be concerned about heterogeneity, both technical and in populations, between the studies analyzed (we now include a quantitative assessment of heterogeneity in all the meta-analyses which is available in the supplementary material). Interestingly, heterogeneity between studies is not substantial for most analyses *except* when pooling different diseases, as would be expected. We are not surprised however by this observation, as it is consistent with other meta-analyses we have performed in high-dimensional data in different contexts after uniform reprocessing of raw data (Thomas et al. 2019; Waldron et al. 2014). Therefore, we believe these meta-analyses add to the literature of individual analyses of these variables by:

1. providing an analysis of heterogeneity of the literature,
2. providing more precise estimates of association by employing a larger and more diverse sample
3. providing estimates that can be continually updated as data are added to cMD

Following the suggestion of the reviewer, we performed a new analysis to demonstrate the value of our meta-analysis over single-cohort analysis. Specifically, we found that only a reduced set of single cohorts (always less than one third) reached significance (at a permissive FDR<0.2) for associations that in meta-analysis were significant at a stricter threshold (FDR<0.01). For example, our highlighted example *Akkermansia muciniphila* is associated with females in the meta-analysis at FDR<0.01 but only 5 single cohorts were able to state the same with statistical significance (FDR<0.2). This means that in the great majority of the single cohorts, *A. muciniphila* would not have been discovered as associated with sex. We note that in all single cohorts, *A. muciniphila* is higher on average in females, which is something again captured by the meta-analysis but not by single cohorts. Another example is the association between Firmicutes bacterium CAG 145, for which there is a single cohort in which it is significantly higher in males, but the meta-analysis and each of the other single cohorts points in the opposite association, meaning that relying on that single cohort would have been misleading. Consistent results like this are an example of how heterogeneity in studies can be a strength, as it is more difficult (although certainly not impossible) for such a consistent association to result from confounding in so many different populations, and in this example the one outlier study may provide clues as to why differences in that one cohort may impact the association between *Firmicutes* bacterium CAG 145 and sex. The comparison of meta-analysis against single studies for association with sex is now reported in Supplementary Figure 1:

Supplementary Figure 1. Meta-analysis of 5,505 individuals' metagenomes from 18 datasets (3,288 females, 2,217 males) shows variations in the composition of the sex-associated microbiome that are not easily detected from the analysis of individual datasets. The 50 microbial species with the highest Standardized Mean Differences meta-analysis coefficient ($Q < 0.01$) of male vs. female samples (computed as Standardized Mean Differences from a linear model controlling for age, BMI, and sequencing depth on centered log-ratio transformed species relative abundances) are reported. Red-blue heatmaps highlight the direction of the effect-size (blue: male,

red: female). Signs mark the FDR significance ($Q < 0.2$ for single datasets, $Q < 0.01$ for the meta-analysis). Plus is used for the positive effect-sizes; minus is used for the negative effect-sizes.

We have also added discussion on the value of large-scale integrative analyses at the end of the second paragraph of the sex analysis, which now reads:

“In total, we found comparably more differentially abundant species and genera in men than women (**Fig. 2a,b**, tot. significant: 30 and 23 species, and 33 and 16 genera, $FDR = 0.01$, **Suppl. Tab. 2**). The meta-analysis approach proved superior to the single-cohort analysis approach as none of the sex-specific associations found in the meta-analysis are identified at $FDR = 0.2$ in more than one-third of the single, under-powered cohorts, even though the direction of association is broadly conserved (**Suppl. Fig. 1**). Likewise, we did not identify significant heterogeneity (avg. $I^2 = 16\%$, $Q_{gen} > 0.1$ in all the significant species).”

Among species identified in meta-analysis as associated with age ($FDR < 0.01$), only one (*Flavonifractor plautii*) is identified even at a more permissive $FDR < 0.2$ in at least a third of the individual datasets, and for BMI, none. Therefore none of the associations with sex, age, or BMI identified by the meta-analysis at $FDR < 0.01$ was found in more than one third of the single cohorts analyzed even at $FDR < 0.2$, highlighting that these small but consistent associations require the large sample size available in this meta-analysis to be detected. We emphasize that these associations identified in meta-analysis *cannot* result from batch effects between different datasets, as the analysis is performed in each dataset independently, while the overall effect is obtained by synthesis of fold-changes across all datasets.

REVIEWER #1 COMMENT 3

One of the highlights is that the authors developed an oral introgression score based on the total relative abundance of a pre-defined list of species highly abundant in oral cavity. This is great but unfortunately, I do not see any validation of this approach. People have used a different approach by comparing 16S sequences to Human Oral Microbiome Database (Coker, O. O., Dai, Z., Nie, Y., Zhao, G., Cao, L., Nakatsu, G., Wu, W. K., Wong, S. H., Chen, Z., Sung, J. J. Y. & Yu, J. Mucosal microbiome dysbiosis in gastric carcinogenesis. *Gut* 67, 1024–1032 (2018).). I am wondering how much difference in oral bacterial percentage will be found between the two approaches. Also, some species like *Prevotella* spp. can colonize both gut and oral cavity. Since the approach has a species-level resolution, I am not sure if the authors can properly assign a habitat for those species solely based on the species identity. Do you think that sequence-level approach will work better? Finally, it is important to show that, using the new approach, the oral bacterial fraction in the feces of healthy volunteers is low (at most a few percent). This is a sanity check: if any of the bacterial species in the list colonize both gut and oral cavity, the total fraction of oral bacteria in feces will be inflated even in healthy people. Typically, it does not make too much sense to observe $>10\%$ of oral bacteria in healthy people.

We thank the reviewer for the positive feedback on the oral-to-gut introgression score. We have included several new analyses and a comparison with the Human Oral Microbiome Dataset (HOMD) as a validation:

Answer 1 on comment # 3: comparison to HOMD

The reviewer suggested that a similar approach was based on the Human Oral Microbiome Database (HOMD), and asked how much difference could be observed between the two. We used the 492 oral species identified by HOMD using shotgun metagenomics to implement an oral to gut introgression score comparable to our species lists at several different prevalence thresholds. Rebuttal Figure 1 shows cumulative abundance of oral species in stool samples (continue here describing Rebuttal figure 1.)

Rebuttal Figure 1. Validation of the proposed approach for oral to gut microbial species introgression - oral samples (n = 857) from curatedMetagenomicData 3 are retrieved and several lists of putatively oral species are computed according to progressive thresholds of prevalence in this (1%, 5%, 10%, 20%, 30%, 40%, 50%). The oral species from the Human Oral Microbiome Dataset (HOMD) are also retrieved and evaluated. Lists of oral species are evaluated in two sets of fully healthy (n cohorts = 25, n samples = 6,891) and fully unhealthy populations (n cohorts = 48, n samples = 3,632). AUROC of each score predicting the two groups is presented. Average number of logarithms of difference between cases and controls in a meta-analysis of 15 human diseases (30 studies) is also reported.

In this analysis we report the distribution of scores and AUROC for predicting disease status using each oral to gut introgression index (7 from our approach and the HOMD) in 6,981 control participants from 25 studies and 3,642 participants with various disease diagnoses from 48 studies.

The HOMD score reaches an AUROC of 0.74 for discriminating health vs. disease, while our best approach (threshold of minimum 1% prevalence in oral cavity) reaches an AUROC of 0.8. We consider this to be a notable improvement, but the reasonable accuracy also of the independently derived HOMD score, adds to our confidence in the general approach of defining an oral to gut introgression score.

We also note that the HOMD approach reaches 10% oral to gut introgression in healthy samples (the upper limit suggested by the reviewer). By our definition (see below response to Comment 2), oral to

gut introgression percentage is well below 1%, indicating a sensitive and conservative measure. This might be the result of a large number of species in the HOMD signature (n = 492), some of which can be considered typical gut bacteria (i.e. it includes *Escherichia coli*).

Answer 2 on comment 3: show that the oral bacterial fraction in the feces of healthy volunteers is low (at most a few percent).

We have added this analysis as Figure 5b (below). As expected by reviewer 2, our preferred oral to gut introgression score, using species present in 1% or more of healthy oral microbiomes, account for between 0.1-1% total relative abundance in healthy stool microbiomes. We note that some species are normal commensal both in the gut and in the oral cavity, and genera like *Prevotella*, which contains different species typical of the two body sites. To demonstrate this, we analyzed healthy participants in cMD3 who provided both oral and stool samples (n = 1,053), and selected all species present in at least any 25 oral samples and any 25 stool samples (~2.4%). By these criteria there are 49 species present in both the oral cavity and stool. We computed the median relative abundance in stool and oral cavity for these 49 species, and report below the fold change of body site with the largest median relative abundance compared to the smallest, noting that very few have comparable relative abundance (fold-change close to 1):

SPC	N.STOOL	N.ORAL	MED.STOOL	MED.ORAL	FOLD-CHANGE
s_Actinomyces_graevenitzii	33	389	0.01	0.93	125.2
s_Actinomyces_odontolyticus	124	540	0.01	0.32	59.52
s_Actinomyces_oris	34	537	0	0.44	109.81
s_Actinomyces_sp_HMSCo35Go2	64	429	0.01	0.21	37.21
s_Actinomyces_sp_HPAAo247	30	297	0	0.05	17.78
s_Actinomyces_sp_ICM47	84	424	0.01	0.99	168.28
s_Actinomyces_sp_S6_Spd3	35	305	0.01	0.15	27.56
s_Actinomyces_sp_oral_taxon_181	30	291	0.03	0.12	3.59
s_Bifidobacterium_dentium	43	32	0.01	0.03	5.88
s_Bifidobacterium_longum	252	28	0.09	0.01	8.1
s_Rothia_mucilaginosa	152	579	0.01	1.4	151.01
s_Cutibacterium_acnes	59	185	0.03	0.01	2.17
s_Atopobium_parvulum	32	406	0.02	0.08	4.47
s_Olsenella_profusa	86	28	0	0	1.04
s_Olsenella_scitoligenes	185	26	0.01	0	5.65

s__Bacteroides_ovatus	302	35	0.33	0.01	55.81
s__Bacteroides_stercoris	198	28	0.59	0.01	79.6
s__Bacteroides_uniformis	324	55	1.23	0	250.89
s__Bacteroides_vulgatus	358	65	1.89	0.01	225.37
s__Prevotella_melaninogenica	39	562	0.01	1.15	123.01
s__Alistipes_putredinis	232	35	2.74	0	808.58
s__Gemella_haemolysans	55	573	0.01	0.54	77.12
s__Gemella_sanguinis	84	572	0	0.51	135.01
s__Staphylococcus_epidermidis	64	27	0.23	0.16	1.39
s__Granulicatella_elegans	37	552	0	0.11	29.99
s__Streptococcus_anginosus_group	61	322	0.01	0.03	6.47
s__Streptococcus_australis	57	420	0	0.12	25.34
s__Streptococcus_gordonii	64	440	0.01	0.26	30.99
s__Streptococcus_infantis	93	568	0.01	1.18	218.17
s__Streptococcus_mitis	215	606	0.01	2.63	305.31
s__Streptococcus_oralis	101	589	0.01	2.2	404.08
s__Streptococcus_parasanguinis	293	529	0.02	1.29	56.89
s__Streptococcus_salivarius	292	440	0.06	0.71	11.2
s__Streptococcus_sanguinis	56	538	0.01	0.8	115.73
s__Streptococcus_sp_A12	54	352	0	0.04	15.49
s__Streptococcus_vestibularis	62	268	0.01	0.03	3.66
s__Eubacterium_sulci	62	324	0	279	64.29
s__Mogibacterium_diversum	31	345	0	0.18	57.86
s__Dialister_invisus	84	144	0.11	0.01	13.66
s__Veillonella_atypica	135	443	0.01	0.55	53.06
s__Veillonella_dispar	196	508	0.01	0.71	62.76
s__Veillonella_infantium	137	484	0.01	0.29	52.05

s_Veillonella_parvula	236	567	0.02	0.68	34.85
s_Veillonella_rogosae	41	305	0.01	0.05	5.38
s_Veillonella_sp_T11011_6	129	404	0.01	0.22	23.19
s_Neisseria_flavescens	41	554	0.02	1.77	92.53
s_Haemophilus_parainfluenzae	265	587	0.02	2.53	112.48
s_Haemophilus_sp_HMSC71H05	156	125	0.01	0.12	8.87
s_Fretibacterium_fastidiosum	130	198	0	0.03	23.22

Rebuttal table 1. 49 species present in both the body-sites considered of at least 25 individuals who provided paired gut/oral samples (N individuals = 1,053), reporting for each species the number of samples in which the species was detected in each body-site, the median abundance of each body-site and the higher and the lower median.

The body site with the greater relative abundance of any of these 49 species has a median fold-increase of 52 times relative to the body site of lower relative abundance, and very few have comparable relative abundance in both oral cavity and stool. Neither is the prevalence (presence/absence) of most species close to equally distributed, even for the 11 species with fold-change less than 10. Therefore while some species do inhabit both oral and gut sites, almost all can be described as predominantly of either one or the other.

These changes have been included in a new figure 5, which now includes two more panels (**a** and **b** in the new version), relatively the prevalence of oral species (estimated according different possible thresholds, panel **a**), and the score evaluated in healthy and diseased populations (panel **b**).

Figure 5. Cumulative relative abundance of typically oral taxa in the human gut microbiome is a potential indicator of disease - **a**) boxplots showing the percentage of positively scoring individuals from a dataset of 6,891 gut microbiomes from healthy, adult participants, under different definitions of the score based on progressive thresholds of prevalence to define an oral species. **b**) distribution of the per-population mean score in 6,891 gut microbiomes from adult, healthy individuals (25 studies), and 3,632 gut microbiomes (48 studies) from adults who have received a specific diagnosis. Asterisks mark the between-distribution Mann-Whitney $P < 0.05$. AUROC of the per-dataset average score predicting the diseased state is reported. **c**) log10 distributions of summed relative abundance of oral-cavity typical microbial species (defined using 1% of the oral samples as a threshold) in 10 cohorts of CRC patients (orange) and related controls (blue). Asterisks mark the Mann-Whitney $FDR = 0.1$. AUROCs of the introgression score for predicting CRC versus controls are presented. **d**) boxplots showing the log10 distributions of oral-cavity typical microbial species cumulative abundance (defined using 1% of the oral samples as a threshold) in 14 diseases (20 cohorts), divided by disease (orange) and controls (blue). Asterisks mark the Mann-Whitney $FDR = 0.1$. AUROCs the introgression score against disease versus healthy conditions are presented. **e**) Forest plot showing the meta-analysis of the association of disease and corresponding healthy controls of oral species cumulative abundance in 30 cohorts. Single datasets effect sizes are computed as mean difference (beta-coefficient) extracted by a linear model controlling for sex, age, BMI, number of reads, and antibiotics usage when possible. A natural log of the score is used. Zeros are imputed using the minimum value in each dataset. Studies n is reported in **Fig. 4**. Purple/gold: coefficient different from zero (Wald $FDR = 0.01$ in the single cohorts, meta-analysis $P < 0.05$); blue/purple: coefficient non-significantly different from zero.

REVIEWER #1 COMMENT 4:

Another important but easy-to-fix drawback is that the statistical association between oral introgression score and disease status did not take antibiotics as a confounding variable. Antibiotic is the most important variable that could lead to oral bacterial enrichment in feces. The apparent associations with disease status may be caused by the use of antibiotics and it would be critical to know if disease remains associated with oral bacterial percentage even after excluding the effects of antibiotics.

Most of the studies (18 out of 30) included in our host-disease meta-analysis excluded any participants who had taken antibiotics in the previous three or six months; we have added this information to Figure 4, to show that antibiotics do not appear to be among the strongest drivers of the observed signals. Among the 12 that did not exclude participants who recently received antibiotics, 6 provided antibiotics usage as a variable. In particular, the analysis relative to type-2 diabetes, coronary artery disease, and coronary artery disease + heart failure controlled for antibiotics in the linear modeling. The MetaCardis dataset, which was used to perform these analyses, was included only in the revised version of the manuscript. Still, adjusting by antibiotics does not affect the AUROC compared to the cohorts for which this aspect was ruled out by the exclusion criteria. The revised Figures 4 and 5 are shown below.

Figure 4. Meta-analysis of 15 diseases and 30 cohorts reveals microbial markers of disease or health in 2,346 controls and 2,300 diseased patients - a) the 30 microbial species with the highest meta-analysis coefficient and FDR = 0.01 of disease-associated vs. control samples, with a prevalence of at least 1% in the cohort. Effect sizes are computed as Standardized Mean Differences (SMDs) from a linear model controlling for sex, age, BMI, sequencing depth, and usage of antibiotics in the type-2 diabetes datasets on centered log-ratio

transformed species relative-abundances. Acronyms are colorectal cancer (CRC), Crohn's disease (CD), ulcerative colitis (UC), type-2 diabetes (T2D), atherosclerotic cardiovascular disease (ACVD), Behcet disease (BD), soil-transmitted helminths (STH), myalgic encephalomyelitis or chronic fatigue syndrome (ME/CFS), coronary artery disease (CAD), coronary artery disease with heart failure (CAD+HF), Parkinson's disease (PD). Effect sizes of CRC, CD, UC, & T2D are synthesized in a meta-analysis before the second meta-analysis so that these more frequently studied diseases do not dominate the results over diseases for which a single dataset is available (ACVD, asthma, migraine, STH, cirrhosis, ME/CFS, schizophrenia, BD, CAD, CAD+HF, PD). Yellow shape: individually significant effect ($Q < 0.2$). Light-blue shape: non-individually significant effect. Black diamonds: SMDs between cases and controls synthesized by meta-analysis. Yellow horizontal lines show 95% confidence intervals of the synthesized effect size. **b)** heatmaps showing AUROCs of different Random Forest experiments on the binary discrimination "disease (generic) vs. healthy". Numbers next to study names report the number of cases and controls for each cohort. Cross validations are 10-fold repeated 10 times. Four different "LODO" (Leave-one-dataset-out) AUROCs are shown: 1,2) models trained using all independent datasets (of the same and different diseases) as the test set, using either 1) microbial species relative abundance, or 2) relative abundance plus age, sex, BMI and depth of the sample as features; 3) models trained only on different diseases than the test set; 4) models trained only on the same disease as the test set. The cyan-red-gray bar indicates whether the samples possibly took antibiotics or this information was not available.

MINOR COMMENTS:

1. Abstract: "...to address several ongoing challenges...". What are those challenges?

We thank the reviewer for this suggestion. We have accordingly modified this sentence in the Abstract, which nows reads as:

"to address several ongoing challenges in human microbiome epidemiology. These include assessing the relationship of the human microbiome with basic and clinical host outcomes, which can be performed in cMD 3 with a greater sample size and diversity of populations than previously possible."

2. Third line of introduction: physiology should be physiology.

We have amended this.

3. Random forest algorithm: "The parameters were set in advance based on our previous experience with this algorithm." The parameters are best searched on a dataset-by-dataset basis. My experience is that, 1,000 trees is a bit low and it is better to set a tree depth to avoid overfitting. It is always a good practice to set a high number of trees such as 5,000 or 10,000 and uses grid search for a few key parameters as mentioned in the manuscript.

We agree and have re-performed all analyses using 10,000 trees, as was also used in the bioBakery 3 paper (Beghini et al. 2021). While hyperparameter tuning by internal cross validation is important for other approaches such as support vector machines, but for random forest this is much less relevant. As with all our other studies that employed random forests (Thomas et al. 2019; Lee et al. 2022; Asnicar, Berry, et al. 2021) we prefer to fix the hyperparameters (number of trees, maximum samples per leaf, maximum tree depth, impurity criterion) prior to all analyses.

Before:

"The parameters were set in advance based on our previous experience with this algorithm."

Now:

“Consistent with previous work (Beghini et al. 2021), we set the following parameters for all experiments: 10,000 trees, 1% of the features in each tree, minimum 5 samples per leaf, unlimited tree depth, and entropy as the node impurity criterion.”

Overall, I am not impressed by the scientific story of this paper. It is technically sound in general but has some flaws in the analysis of oral bacteria in feces. From my opinion, this is best considered as a resource paper: all of the statistical analyses included in the manuscript confirmed previous findings and showcased how this wonderful dataset can be reused to identify clinical associations.

We thank the reviewer for their positive perspective on the cMD3 resources. Our intention is for this paper to 1) serve as a resource paper describing the cMD 3 package, 2) introduce and analyze a simple oral to gut introgression score that can be easily applied in future studies, 3) provide a number of meta-analyses that serve both to demonstrate the utility of the database and to be a reference for commonly referred-to microbiome associations (sex, age, BMI, and several diseases), and 4) demonstrate use cases and provide code for how to analyze different types of variable/clinical outcomes for this and future updates of curatedMetagenomicData.

Reviewer #2 (Remarks to the Author):

The manuscript by Manghi and colleagues describes meta-analyses of human microbiome associations with sex, age, BMI, and disease states, and a cross-study meta-analysis of introgression of oral microbes into the gut. Notably, as part of this study, the authors have substantially updated the curatedMetagenomicData R package, greatly increasing the number of samples. The amount of work that goes into curating these data is laudable. The authors similarly present a considerable body of data resulting from their meta-analyses. However, several caveats need to be addressed for these findings to be helpful to the microbiome community and beyond.

MAJOR ISSUES

1. Throughout the manuscript, the authors extensively use the definition “meta-analysis”. A key component of such an analysis is an accurate description of the search criteria and keywords for relevant publications, inclusion and exclusion criteria for studies, and a flow diagram of how many studies were lost in each step for a given reason. In the methods section, the authors do provide the filtration criteria for the various comparisons (age, sex, BMI), however I could not find the criteria for including or omitting a study from the whole database. It seems like many studies dissecting the microbiome in human disease are missing, including those included in previous microbiome meta-analyses (of UC, CD, T2D), and it is unclear why. This could potentially impact the conclusions drawn by the authors throughout the manuscript.

We thank the reviewer for noting these limitations. We included the following section in the method to clarify the study selection criteria:

“We expanded the cMD package to version 3 by including human shotgun metagenomics data. Selection criteria for inclusion were sample size, availability of raw data in the Sequence Read Archive, completeness of essential sample metadata (e.g., age, BMI, sex, disease status), relevance to existing cMD 1 studies, user requests, and community interest.”

In this process of continuous growth across many different study conditions it isn't feasible to perform ongoing systematic review. However, we do not see plausible potential for reviewer bias in the inclusion/exclusion of studies, such as in a meta-analysis of a single treatment/exposure and outcome where the reviewer might have a preference (even unconscious) for the outcome of the meta-analysis, because we have no prior expectations for what microbiome associations will be identified in each dataset. We therefore refer to our methods as “meta-analysis” and not “systematic review and meta-analysis”, and acknowledge that some relevant datasets may be missing for no reason other than that we have not yet prioritized them for addition.

2. It is unclear from the map as currently presented in figure 1 whether each point is simply the location of the study's primary institute, or relates to the geography of the human populations analyzed in the study. The latter is far more relevant than the first, and the figure should be adjusted to reflect the geographic distribution of the human populations included in each study. Much of the text in this figure could be relegated to the main text (or is indeed a repetition of it already), and visualizations could be added for a better summary of the breadth of studies included in the package. With the dominance of populations in highly developed countries in microbiome datasets, it would be helpful to better understand the representation of diverse groups in this new package, for example see the plots in <https://pubmed.ncbi.nlm.nih.gov/35167588/>.

In response to this comment we have produced a new version of Figure 1 that:

- includes a heatmap-style color scheme representing the number of participants per nation in logarithmic scale
- eliminates the text boxes to increase the size of the map
- maintains the actual locations of data collection within those countries (in red). Some of these locations are cities, while some are villages and others are putative areas of habitation of some tribes. We hope that this clarifies the granularity of participant location in cMD3.

The new **Figure 1**:

Figure 1. curatedMetagemomicData (cMD) 3 is an open-source software package providing uniformly processed metagenomic data and manually-curated metadata, available in R/Bioconductor and via the command line. a) cMD 3 distributes more than 22K metagenomes derived from 94 cohorts and 90 publications (42 countries and ~160 more detailed locations). cMD 3 metagenomic data are profiled with the MetaPhlan 3 and HUMAnN 3 tools from the bioBakery3 suite. cMD 3 contains standardized, manually curated, and machine-validated sample-level metadata on host's characteristics and specimen's technical attributes. **b)** Barplot showing the top 30 most complete metadata attributes among the samples from individuals above 12 years (left) and below (right), where the red lines mark 500 and 250 samples, respectively. Blue lines mark the total number of samples in the two categories. Metadata are colored according to whether they are required (red), recommended (blue), or optional (orange).

We have also added the following paragraph, adding some considerations on the current state of WGS sequencing, in line with the paper mentioned by the reviewer.

New text at the end of the Discussion:

“Likely, the data provided by cMD 3 reflects strengths and limitations of shotgun metagenomic sequencing of human hosts by the scientific community. Stool is the most commonly studied body site. Africa, South America, Middle-East countries, eastern Europe, southern Asia and Polynesia are still under-represented. Greater sampling in under-represented populations is needed to discover the full diversity of the human microbiome and to uncover roles of the microbiome in health and disease in those populations and elsewhere (Fig. 1a).”

3. While the authors provide a considerable amount of data in this manuscript, interpreting the results in an insightful manner is limited by several factors detailed in the following points. Notably, each of the analyzed parameters (sex, age, BMI, some of the diseases including CRC and T2D) was previously included in at least one microbiome-focused meta-analysis. While some of these works are cited in the manuscript, there is little discussion of the current findings in context of previous meta-analyses. Most importantly, do the meta-analyses agree? Are there new microbial signals that were not previously detected? Or some that may be attributed to noise or confounders? And which components of the authors' approach underlie these differences?

We agree with the importance of comparing our results to the previous meta-analyses and have added a number of references:

The meta-analysis of microbiome in relation with sex and age by Zhang et al from *Nature Aging*, 2021, revealed several biomarkers that we confirmed, such as the increased presence of *Akkermansia muciniphila* and *Prevotella copri* in adult females and adult males, respectively.

For BMI, the closest we could find was a systematic review and meta-analysis of 16S amplicon sequencing studies of obesity (Pinart et al. 2021). Although that study reported findings only at the genus and phylum rank and of alpha diversity, and did not control for age or sex, we replicate several essential observations: 1) a greater number of taxonomic features with increased relative abundance in high BMI (previously reported in obese vs lean participants), 2) more *Firmicutes* and fewer *Bacteroidetes* in high BMI participants (68/116 [59%] biomarkers in higher BMI participants and 18/34 (53%) in low BMI participants; instead, *Bacteroidetes* were respectively 10 and 14 (41% and 9%), therefore they were more numerous overall in the low BMI individuals). In line with the same review based on 16S amplicon sequencing, we could also confirm the presence of Actinobacteria (21 species), Proteobacteria (10 species), and Fusobacteria (4 species) positively associated with BMI while no species from these phyla were found to negatively associated with it. We are not aware of other meta-analyses of association between the stool microbiome and BMI using shotgun sequencing.

We are aware of several other microbiome meta-analyses that also search for a common signature of multiple diseases: Duvallet *et al.* (16S amplicon sequencing), Tierney *et al.*, and Pasolli *et al.* (the latter sharing authors with this paper). Relative to these 3 studies, our cross-disease analysis encompasses respectively 1.5, 2.1, and 3 times more phenotypes and 1.07, 2.3, 3.7 times more cohorts. A recent effort by Su *et al.* analyzed 9 phenotypes in a single cohort and partially agrees with our findings, which we now cite.

Only a few large-scale studies have sought to identify associations with the human microbiome at the species-level, and with the statistical power and generalizability made possible by meta-analysis spanning data from >10 countries, comparable to this work. Cross-disease analysis has received significant attention and resulted in numerous studies, but only the work by Su *et al.* was shotgun-based and specifically aimed at species-level analysis. We discuss these innovative aspects of our analysis as follows:

Text at the beginning of the cross-disease analysis:

“Biomarkers of “unhealthy” or dysbiotic microbiomes, which are associated with multiple diseases as opposed to being biomarkers of specific diseases, have been reported. (Armour et al. 2019; Duvall et al. 2017; Pasolli et al. 2016; Tierney et al. 2022; Su et al. 2022). Here, we add evidence of such biomarkers at higher resolution. The study Duvall et al. (Duvall et al. 2017) analyzed 28 published case-control gut microbiome studies spanning 10 diseases to characterize disease-associated microbiome changes. However, they used only 16S rRNA sequencing data, limiting the resolution. We analyzed 4,646 samples from 15 diseases, composed of 2,300 cases and 2,346 controls (see **Methods**). We adopted a hierarchical random-effects meta-analysis to synthesize effect sizes: at the first level for diseases studied in more than one dataset (CRC, T2D, CD, UC), and at the second level across 15 diseases (4 from meta-analyses, 11 from individual datasets).”

Text comparing ours with the Sun et al. cross-disease analysis:

“Our results identified *Actinomyces odontolyticus*, *Eubacterium*, and *Roseburia* as health-associated species, agreeing with a previous study that analyzed 2,320 samples associated with 9 disease phenotypes (Su et al. 2022).”

In the Discussion section, highlighting the differences between our and previous meta-analyses:

“This integrative analysis expands on previous meta-analytical efforts focused on specific conditions (Zhang et al. 2021; Thomas et al. 2019; Wirbel et al. 2019; de la Cuesta-Zuluaga et al. 2019; McCulloch et al. 2022; Lee et al. 2022; Derosa et al. 2022, 2021), or considering a limited number of samples and conditions (Duvall et al. 2017; Schmidt et al. 2019; Tierney et al. 2022; Su et al. 2022). We provide tables of replicated associations that can assist researchers in interpreting putative microbiome biomarkers.”

In the oral to gut introgression score section:

“Increased oral-to-gut introgression of microbes (i.e. increased passage of oral-typical species in the intestine) in response to physico-chemical alterations of the intestinal lumen in inflammation has been postulated in several diseases (Thomas et al. 2019; Jie et al. 2017; Flemer et al. 2018).”

Paragraph comparing our results on age and BMI with existing literature (single-dataset analyses, and meta-analyses when possible)

“Taxa showing the strongest correlations with age were the negatively associated *Bacteroides* species “OM05 12” (meta-analysis correlation = $-.11$, 95% CI $[-.16, -.07]$, $Q < 0.0001$) and the positively correlated *Bifidobacterium dentium* (meta-analysis correlation = $.12$, 95% CI $[.8, .18]$, $Q < 0.0001$), consistent with the previous study (Zhang et al. 2021). The two strongest correlations with BMI were the negatively associated *Firmicutes* species “CAG 95” (meta-analysis correlation = $-.11$, 95% CI $[-.16, -.07]$, $Q < 0.0001$), and the positively correlated

Gemella haemolysans (meta-analysis correlation = .11, 95% CI [.06, .15] and .10, 95% CI [.08, .13], $Q = 0.0001$). The increased presence of several putative oral taxa, such as *Streptococcus gordonii* in older ages and of *Streptococcus mitis* in higher BMI, also agrees with the previous studies (Zhang et al. 2021; Schirmer et al. 2016). We confirmed the decreased abundance of *A. muciniphila* at lower BMI (Depommier et al. 2019; Karcher et al. 2021), when adjusted for sex and age. We could also replicate the previously reported positive association of the *Blautia* genus with increased BMI (Castañer and Schröder 2018)."

Sentences confirming the previously found relationship between alpha diversity and sex, age, and BMI in existing literature (both single-dataset analysis and meta-analyses):

"Alpha diversity (Shannon entropy) was significantly increased in females with the Standardized Mean Difference (SMD) of -0.16 (95% CI [-0.21, -0.11], $P < 4.6 \times 10^{-9}$) relative to males, as previously reported (Sinha et al. 2019)."

"Shannon diversity was positively associated with age (meta-analysis correlation coefficient = .09, 95% CI [.06, .12], $P = 1.6 \times 10^{-10}$), in line with the observations reported by (de la Cuesta-Zuluaga et al. 2019; Zhang et al. 2021), and negatively associated with BMI (meta-analysis correlation = -.08, 95% CI [-.12, -.05], $P = 7 \times 10^{-5}$) as previously reported (Castañer and Schröder 2018)."

Sex-related confirmation of previous analyses:

"*P. succinatutens* was previously observed to increase in males after a weight-loss intervention (Cuevas-Sierra et al. 2021), and *P. copri* has been previously associated with non-Westernized lifestyles (Tett et al. 2019), and fiber-rich and Mediterranean diet-related nutritional habits (De Filippis et al. 2019; Wang et al. 2021). Conversely, *A. muciniphila* and *I. butyriciproducens* are important butyrate producers (Bui et al. 2020). *A. muciniphila* has been shown to play a role in decreasing the risk of metabolic syndrome (Dao et al. 2016). Our findings are potentially explainable by the longer transit time previously observed in females (Degen and Phillips 1996), confirmed in a recent investigation (Asnicar, Leeming, et al. 2021), and they were partially mirrored in the functional potential analysis, where for example the *L-lysine fermentation to acetate and butyrate* was increased in females (SMD = -0.11, 95% CI [-0.18, -0.05], $Q = 0.009$) (**Suppl. Fig. 2**), and maltose-6'-phosphate glucosidase was enriched in males (SMD = 0.17, 95% CI [0.12, 0.22], $FDR = 0.0001$) (**Suppl. Fig. 3, Suppl. Tab. 2**)."

4. Functional analyses: The functional genes presented in the supplementary figures throughout the paper should be filtered or caveated - many of them make little sense in the context of the microbiome, are primarily eukaryotic, or represent universal bacterial proteins (see the number of universal ribosomal proteins in Figure S5 - how should such a result even be interpreted?). Further, uninterpretable pathway codes, e.g. "PWY-5136, PWY-5138, PWY-7288, PWY66-391" should not be listed in the main text.

We have:

- Lowered the False Discovery Rate to 0.01 in all meta-analysis, including the functional ones (see also general comments).
- Removed pathway and KEGG codes from the text.
- Improved interpretation of the results in most parts of the text.

As a result, many universal and ribosomal proteins are no longer there. Regarding the presence of eukaryotic pathways, pathways identified by the HUMAnN 3.0 software in the microbiome are not considered to be the prerogative of bacteria only, but rather as pathways in which bacteria are estimated by the HUMAnN algorithm to possess the genes necessary to cover a minimum of the 20% of the reactions that constitute a given pathway. Furthermore, some eukaryotic pathways may remain under-studied in bacteria. For these reasons it is therefore possible to correctly identify pathways that are currently attributed mostly to Eukaryotes.

5. Throughout the manuscript, the authors should better connect key results with biological interpretation. For example, they say “Many of the taxa (species and genera) identified as negatively correlated with BMI and positively with age are known for being short-chain fatty acids producers”: but which SCFAs do they mean? How do they know which taxa are truly SCFA producers, a trait that varies dramatically between species and strains? There is another example of this: “Female-associated microbiomes contained increased potential for tryptophan-derived indole synthesis... in line with the increased abundance of *A. muciniphila* (Chappell et al., 2016)” But the reference provided does not clearly support this statement.

We have improved text readability and interpretability by sticking with supporting literature and functional-level supported findings. On SCFAs, we mostly referred to butyrate, but usually butyrate producers are also involved in the metabolism of other SCFA so we wanted to be more general, but we agree we should try to be as specific as possible. It is also true that several phenotypes are strain-specific but for the species in which several strains have been experimentally analyzed and are consistently associated with a phenotype it is plausible to consider that phenotype as a core characteristic of the species. With this reasoning in mind we amended the code to improve and more accurately address the biological interpretation, including more exhaustive citations of previous findings.

We point to previous answers to the same reviewers which include surveying this aspect.

6. In all figures, it is crucial to visualize or list the n of each study, as the n will have a large impact on whether a result in a study is significant or strongly predicted in the ML analyses. Which studies were class balanced? How are class imbalances (particularly among diseases) handled by the authors?

We thank the reviewer for this suggestion. We have added the n of each study in the updated Figure 2 and in Figure 4, in both as a novel part of the ML heatmaps. This allows us to know the cardinality of each class in the two main binary outcome analyses.

Figure 2, panel C

Figure 2. Meta-analysis of the gut microbiome from 5,505 individuals (3,288 females and 2,217 males) across 18 datasets, revealing sex-associated microbial differences in healthy adults based on stool samples. [...] c) AUROCs of a leave-one-dataset-out (LODO) validation predicting sex using a Random Forest algorithm trained on various features: (i) age, BMI, sequencing depth of the samples; (ii) species relative-abundances, with and without (iii) age, BMI and sequencing depth as features; (iv) KEGG-level-collapsed UniRef90 gene families abundances with and without (v) age, BMI and sequencing depth as features. On top: the number of female and male participants in each study.

Figure 4, panel B:

Figure 4. Meta-analysis of 15 diseases and 30 cohorts reveals microbial markers of disease or health in 2,346 controls and 2,300 diseased patients - [...] b) heatmaps showing AUROCs of different Random Forest experiments on the binary discrimination “disease (generic) vs. healthy”. Numbers next to study names report the number of cases and controls for each cohort. Cross validations are 10-fold repeated 10 times. Four different “LODO” (Leave-one-dataset-out) AUROCs are shown: 1,2) models trained using all independent datasets (of the same and different diseases) as the test set, using either 1) microbial species relative abundance, or 2) relative abundance plus age, sex, BMI and depth of the sample as features; 3) models trained only on different diseases than the test set; 4) models trained only on the same disease as the test set. The cyan-red-gray bar indicates whether the samples possibly took antibiotics or this information was not available.

The reviewer has two concerns: i) how do we treat class imbalance; ii) studies with large n will guide the meta-analysis or will result better predicted in ML.

In the meta-analysis, we apply 3 techniques: a) we scan cMD 3 for studies with a chosen size in both the classes considered, we then b) use linear models, meaning that in classification tasks the result will be based on difference of means which is independent from the size of the dataset; c) we perform random-effects meta-analysis applying the Paule-Mandel heterogeneity estimator which mitigates the power of larger studies. In general, these three strategies combined are a formally correct way of dealing with the problem of class imbalance. We have reported this in the Methods, as follows:

“Inclusion and exclusion criteria

For sex, age, and BMI analyses, datasets *AsnicarF_2021* and *MetaCardis_2020_a* were divided into UK/US parts and Germany/France parts, respectively. Meta-cohorts for analysis of these attributes were selected based on the following common criteria: reported age greater than 16 years (“*age!=NA*” & “*age≥16*”), data collected from stool sample (“*body_site==stool*”), generally healthy or control in a case-control study (“*study_condition==control*”), baseline or single time point (“*days_from_first_collection in [0.0, NA]*”), and reported BMI (“*BMI!=NA*”). Some additional, feature-specific selection criteria were applied:

1. For analysis of sex-associated microbiomes: at least 25% and 40 samples of each male and female, resulting in 5,505 healthy samples from 21 studies.
2. For analysis of age-associated microbiomes: at least 40 samples and an interquartile range (IQR) of age greater or equal to 15 years, resulting in 4,723 samples from 18 datasets.
3. For analysis of BMI-associated microbiomes: at least 40 samples and an IQR greater than or equal to 3, resulting in 6,361 samples from 32 datasets.
4. For analysis of disease-associated microbiomes: case-control studies with at least 10 cases and 10 controls after excluding individuals with pre-pathological conditions (glucose tolerance and colorectal adenoma), including only samples annotated for age, sex, and BMI. This resulted in 2,300 cases and 2,346 baseline controls from 30 cohorts spanning 15 diseases, including four cardiometabolic diseases.

Two studies (*HMP_2019_ibdmdb* and *NielsenHB_2014*) included samples for both Ulcerative Colitis (UC) and Crohn’s Disease (CD). These diseases were analyzed separately, each with the common control group. Another study (*XieH_2016*) included samples for both asthma and migraine; these were also analyzed separately using the shared controls.

This filtering resulted in the following meta-cohorts:

- Four cardiometabolic diseases (type 2 diabetes (T2D, n=1427), atherosclerotic cardiovascular disease (ACVD, n = 305), coronary artery disease (CAD, n = 563), and coronary artery disease with heart failure (HF, n = 469))
- One psychological pathology (schizophrenia, n = 167)
- One gastrointestinal tract disease having a tumoral character (colorectal cancer (CRC), n = 1300)
- Two gastrointestinal tract autoimmune diseases (Crohn’s disease (CD, n = 309), ulcerative colitis (UC, n = 346),
- One autoimmune non-gastrointestinal tract disease (asthma (n = 200))
- One multisystem inflammatory disease (Behcet disease, BD, n = 65)
- One liver disease (cirrhosis, n = 237)
- Soil-transmitted Helminths (STH, n = 159)

- A partially uncovered pathology ((Institute of Medicine, Board on the Health of Select Populations, and Committee on the Diagnostic Criteria for Myalgic Encephalomyelitis/Chronic Fatigue Syndrome 2015), myalgic encephalomyelitis or chronic fatigue syndrome (ME/CFS), n = 100)
- A partially uncovered etiology disease that involves the brain, though not considered a nervous system disease (migraine, n = 226)
- Parkinson's disease (n = 45).

Per-dataset regression models

For each disease outcome, we fit Ordinary Least Square (OLS) models using the model formula:

$$\text{clr}(\text{feature abundance}) \sim \text{outcome} + \text{sequencing depth} + \text{sex} + \text{age} + \text{BMI}$$

Sequencing depth was included to control for the effect of pseudocount addition in clr transformation. Another single model was fit for analysis of sex, age, and BMI as outcomes, as above but without disease outcome.

Standardized Mean Difference (SMD) for binary outcomes

We used SMD (Nakagawa and Cuthill 2007) as a scale-free measure of association that can be synthesized across multiple studies. SMD was calculated for each study from the relevant regression coefficient as:

$$SMD = \frac{t \times (n1 + n2)}{\sqrt{n + n2} \times \sqrt{n1 + n2 - 2}} \quad (1)$$

where t is the t-score, and $n1$ and $n2$ are sample sizes for control and disease, respectively. The standard error (SE) of SMD was computed as:

$$SE = \sqrt{\frac{n1 + n2 - 1}{n1 + n2 - 3} \cdot \frac{4}{n1 + n2} \cdot \left(1 + \frac{SMD^2}{8}\right)} \quad (2)$$

Correlation for continuous outcomes

Correlation coefficients were computed from regression tables as (Nakagawa and Cuthill 2007):

$$r = \frac{\sqrt{t}}{(t^2 + n - 1)} \quad (3)$$

where t is the t-value and n is the number of samples in the control group. The correlation coefficients were Fisher-Z transformed for meta-analysis, then the inverse of the Fisher-Z function was applied to the synthesized estimate and its confidence intervals.

Meta-analysis

We used random-effects models to synthesize SMD and partial correlation coefficients, and estimated between-dataset heterogeneity using Paule and Mandel's and Cochran's generalized Q-statistic to

assess significant heterogeneity (Veroniki et al. 2016), using the Python packages *statsmodels*, *skbio*, and *scipy*.

The standard error for combined effect size (SEm) was calculated as:

$$SEm = \frac{1}{\sqrt{\sum_{i=1}^k W_i}} \quad (4)$$

where k is the number of studies, and W_i is the weight of the i -th study, which corresponds to the inverse of the coefficient estimated variance plus the between-study variance estimation (Paule and Mandel random-effect meta-analysis).

Meta-analysis of diseases Whereas microbiome associations with age, sex, and BMI were synthesized using standard meta-analysis methods as described above, we employed a two-step method to identify biomarkers of disease vs. health. In step 1, we performed a meta-analysis of four multi-dataset diseases (10 CRC, 3 CD, 2 UC, 4 T2D) to generate a single estimate and standard error for each of these four diseases. In step 2, we performed a meta-analysis to synthesize these 4 coefficients and of 11 diseases represented by a single dataset, treating disease as a single outcome (ie, “disease” vs control).”

In ML, we treat class imbalance in two ways: first, we use the AUROC, which is an unbiased estimator, meaning that it will never reach a high value just due to a particularly high accuracy only in one class. Second, we use LODO assessments mainly, meaning that the biggest dataset is usually smaller than the amount of samples used in the training set. The result therefore tends not to be influenced by the presence/absence of a large dataset in the training set (as visible in the above figure for example in the case of the dataset MetaCardis, a large one, in the diabetes analysis (see figure above)). This applies to the point ii) raised by the reviewer.

7. As currently presented in the manuscript, the meta-analysis of sex differences in the microbiome is largely uninformative. These differences can be due to behavioral, social, and dietary differences between sexes as opposed to biological differences. It is unclear what the value of the work as presented therefore is. The authors note that their result indicates that it may be important to control for sex in a microbiome study of disease (they point out Akkermansia) - however this is surely already well-established. This point could perhaps be better made in their disease analyses, where they could demonstrate that controlling for sex across studies significantly changes results and interpretation of the studies.

We agree with the reviewer that interpretation of microbiome differences by sex is multifactorial and not straightforward. We include this meta-analysis because it has been the subject of numerous studies, and we are able to synthesize results across a larger and more diverse sample than those studies, presumably identifying more widely shared differences by sex.

As stated before, we have rephrased the interpretation of sex highlighting that is likely to be due to behavioral and nutritional differences (although this needs further validation).

We have as well performed an analysis of multiple diseases without adjusting in the single datasets for the nuisance sex variable as suggested by the reviewer. Overall, analysis is highly correlated with the one including the nuisance variable sex (Kendall's tau = 0.78, **Rebuttal Fig. 1a**). Importantly the correlation of the species which appear in the top 30 effect-sizes of the two analyses is still significant but much weaker (Kendall's tau = 0.6, **Rebuttal Fig. 1b**). Top species are the ones that represents the most important differences and therefore adjusting by sex can lead to different conclusions: for example, when not adjusting by sex the opportunistic nosocomial pathogen *Streptococcus anginosus* (likely due to hospitalization, as reported by the same reviewer at comment 15) appears to be the top species, while when sex is included it does not.

We have included a new supplementary figure in the paper with this new analysis. This figure is mentioned in the Results, when describing the disease analysis, by the following sentence:

“We performed a similar analysis not adjusting by sex. While the overall correlation between the two analyses was high (Spearman's rho = 0.93, $P < 10^{-20}$). Interestingly, however, the top 30 species differed: for example, they included the nosocomial pathogen *Streptococcus anginosus* as the top disease-associated species, suggesting a possible interaction with sex and the presence among the top 10 disease-associated species, of several CRC-related biomarkers such as *Peptostreptococcus stomatis* and *Gemella morbillorum* (Thomas et al. 2019) (**Suppl. Fig. 8**).”

We have also provided extensive statistics on the species, genera, and functions that are the most correlated with sex in the healthy adult microbiome as machine-readable tables, and this information constitutes an additive value for further researchers, able them to adjust for sex by analyzing our data and question findings on single datasets evidences.

Rebuttal Figure 1. Meta-analysis of 15 diseases and 30 cohorts reveals hallmarks of disease in the microbiomes of 2,346 controls and 2,300 diseased patients returns different results if not adjusting by sex - left) scatterplot of of the synthetic standardized mean differences from two hierarchical meta-analyses on CLR-transformed microbial species relative abundances. On the X axis are reported the meta-analysis coefficients from analysis in which linear models are adjusted by age, BMI, and sample's depth; on the Y axis are reported the meta-analysis coefficients from analysis in which linear models are adjusted by the nuisance variable sex also. right) same plot but considering only species among the top 30 effect-sizes in one of the analyses.

Supplementary Figure 8. Meta-analysis of 15 diseases and 30 cohorts reveals microbial markers of disease or health in 2,346 controls and 2,300 diseased patients slightly different with respect to the same analysis performed adjusting also by sex - a) the 30 microbial species with highest meta-analysis coefficient and ($Q < 0.01$) of disease-associated vs. control samples, with a prevalence of at least 1% in the cohort. Effect-sizes are computed as Standardized Mean Differences from a linear model controlling for age, BMI, and sequencing depth, and usage of antibiotics in the type-2 diabetes datasets, on centered log-ratio transformed species relative-abundances. Acronyms are for: colorectal cancer (CRC), Crohn's disease (CD), ulcerative colitis (UC), type-2 diabetes (T2D), atherosclerotic cardiovascular disease (ACVD), Behcet disease (BD), soil-transmitted helminths (STH), myalgic encephalomyelitis or chronic fatigue syndrome (ME/CFS), coronary artery disease (CAD), coronary artery disease with heart failure (CAD+HF), Parkinson's disease (PD). Effect-sizes of CRC, CD, UC, & T2D are synthesized in a meta-analysis prior to be include in the second meta-analysis, so that these more frequently studied diseases do not dominate the results over diseases for which a single dataset is available (ACVD, asthma, migraine, STH, cirrhosis, ME/CFS, schizophrenia, BD, CAD, CAD+HF, PD). Yellow shape: individually significant effect ($Q < 0.2$). Light-blue shape: non individually significant effect. Black diamonds: Standardized Mean Differences between all cases and controls synthesized by meta-analysis. Yellow horizontal lines show 95% confidence intervals of the synthesized effect size.

So while we agree the sex meta-analysis is not easy to interpret and there were other attempts at such analysis, we think that with these additional results and assessment it will be useful for other studies in the field.

8. In figure 2, the predictive effect is strikingly stronger in two studies, regardless of which features are included in the analysis. The predictive effect in these two studies is stronger than in any other comparison. The authors should discuss the possible explanations for this.

The most likely explanation for a scenario like this is the presence of a bias induced by severely diseased and particularly healthy individuals, which lead to optimistic results. Other sources of optimism are difficult to be observed here thanks to the validation applied. One drawback might be represented by the fact that these optimistic datasets are included in the training, but the LODO validation is a safety measure against over-optimism in real-World scenarios. In the text, we inserted the following sentence:

“The presence of two optimistic datasets (JieZ_2017 and QinJ_2012), likely due to severely diseased individuals, was found, although the LODO validation here applied is a robust safety measure to assess the algorithm behavior in real scenarios.”

Figure 2. Meta-analysis of the gut microbiome from 5,505 individuals (3,288 females and 2,217 males) across 18 datasets, revealing sex-associated microbial differences in healthy adults based on stool samples. [...] c) AUROCs of a leave-one-dataset-out (LODO) validation predicting sex using a Random Forest algorithm trained on various features: (i) age, BMI, sequencing depth of the samples; (ii) species relative-abundances, with and without (iii) age, BMI and sequencing depth as features; (iv) KEGG-level-collapsed UniRef90 gene families abundances with and without (v) age, BMI and sequencing depth as features. On top: the number of female and male participants in each study.

9. Can the authors connect their results in BMI and age to existing literature? For example, what about the Bacteroides:Firmicutes ratio? Alpha diversity and BMI or age? Throughout the manuscript, there is a lot of presentation of results without biological interpretation of these results, or sufficient discussion of how these results relate to what is already known in the literature. Further, the limits of BMI as a health indicator should be noted.

We thank the reviewer for this suggestion. We point to the previous answers to the same reviewer for greater adherence to existing literature about BMI (as a similar concern was expressed in another point). We have added a sentence highlighting the limited value of BMI as an indicator.

“Despite the limitation of BMI as an indicator of obesity, at the functional level we also identified the *phosphatidylglycerol biosynthesis I* and *II*, previously linked with obesity (Kayser et al. 2019), and the *glycogen biosynthesis I* (from ADP-D-Glucose) positively correlated with BMI (meta-analysis correlation 0.09 and 0.07, 95% CI [0.06, 0.11] and [0.04, 0.1], FDR = 0.0001), indicating a putative causative role of the microbiome in obesity (Suppl. Fig. 5,7, Suppl. Tab. 4).”

As an aside, we think the Firmicutes/Bacteroidetes ratio has not proven to be a hallmark of obesity, see for example:

<https://www.ncbi.nlm.nih.gov/pmc/articles/PMC7285218/pdf/nutrients-12-01474.pdf>
<https://www.ncbi.nlm.nih.gov/pmc/articles/PMC2629490/pdf/zpq2365.pdf>
<https://onlinelibrary.wiley.com/doi/full/10.1038/oby.2009.167>

As such, we preferred to focus on the other features highlighted by the reviewer.

10. What other metadata exists for each study beyond age, sex, and BMI? Is dietary information available for any? There should be a supplementary figure (or perhaps worked into Figure 1) describing the extent of the metadata that was compiled for these studies.

curatedMetagenomicData provides approximately 140 different metadata variables, and we now highlight the most commonly available ones in two new panels of Figure 1 (**Fig. 1b**). It shows the 30 most available metadata variables by the number of metagenomic samples by which they are present in the package. The panel is divided into samples above 12 years of age and infants. A red line marks the threshold of 500 adult, and 250 infant samples. A blue line marks the total n of samples. This new panel is useful as it shows which metadata are present in the two categories with a high number of samples, which have never been actually employed in any meta-analysis.

Figure 1. curatedMetagenomicData (cMD) 3 is an open-source software package providing uniformly processed metagenomic data and manually-curated metadata, available in R/Bioconductor and via the command line. a) cMD 3 distributes more than 22K metagenomes derived from 94 cohorts and 90 publications (42 countries and ~160 more detailed locations). cMD 3 metagenomic data are profiled with the MetaPhlAn 3 and HUMAnN 3 tools from the bioBakery3 suite. cMD 3 contains standardized, manually curated, and machine-validated sample-level metadata on host's characteristics and specimen's technical attributes. **b)** Barplot showing the top 30 most complete metadata attributes among the samples from individuals above 12 years (left) and below (right), where the red lines mark 500 and 250 samples, respectively. Blue lines mark the total number of samples in the two categories. Metadata are colored according to whether they are required (red), recommended (blue), or optional (orange).

11. Summarizing some of the above points regarding age, sex, BMI – in my opinion, the interesting parts of this manuscript are the diseases meta-analysis, and the oral to gut introgression analysis. However, the reader is presented with a considerable amount of data with limited contribution to knowledge before reaching these parts. It will be helpful if rather than these three sections, the authors would focus on one, for which they have the most relevant studies, and sufficient metadata to address the aforementioned confounders, and discuss it in context of previous works, as suggested above.

We thank the reviewer for this remark. We share the excitement for the analysis of diseases and for the oral introgression, but we also think that the analysis of sex, age, and BMI are also relevant, for two reasons. First, these analyses illustrate the cMD 3 resource more broadly by utilizing datasets that do not have diseased individuals, and by pooling controls from numerous datasets that have relevant ranges of age and BMI. Second, research on the link between the microbiome and both obesity and healthy aging is ongoing, and thus providing a large-scale baseline for such study expands the value of the manuscript. The focus of this analysis is not biological interpretation but identifying reliably associated features for comparison and follow-up by further studies with more targeted experimental setups. We therefore would prefer to maintain these sections in the manuscript; however, we have shortened the sections on sex, age, and BMI. The first section on sex was shortened from 5,667 characters to 4,643. The section about age and BMI was shortened from 3,147 to 2,830 characters.

12. The methodology used to interrogate the microbiome in this work were MetaPhlAn3 for bacterial taxonomy and HUMAnN for functional gene content. The basis of these methods are sound, but limited. The work would be substantially improved with the inclusion of different metagenomic methodology, particularly for the purposes of verifying important conclusions. Furthermore, it would be helpful to address some of the technical variation between studies and how it affected the results. For example, information regarding differences in sequencing platforms should be included in the metadata, and integrated into the analysis.

We thank the reviewer for these observations. For practical reasons we have had to limit taxonomic and metabolic profiling to MetaPhlAn and HUMAnN, widely-used profilers that have been found to have low false-positive rates and high precision in numerous independent benchmarking studies. However, we now include the following technical data in curatedMetagenomicData: read depth, DNA extraction kit, and sequencing platform. We note that by performing analysis per-study and then synthesizing fold-changes by meta-analysis, direct comparisons are **only** made within studies where the DNA extraction kit and sequencing platform are uniform. Despite these potential sources of heterogeneity across studies, we do not see strong evidence of heterogeneity either in I^2 or LODO where training sets consist of pooled data. We have similarly observed previously, in microarray studies, that relevant

sources of heterogeneity in high-dimensional data may not be the obvious technical factors (Y. Zhang et al. 2018).

We include read depth as a variable in all our linear models, with the following justification in the methods:

“Per-dataset regression models

For each disease outcome, we fit Ordinary Least Square (OLS) models using the model formula:

$$\text{clr}(\text{feature abundance}) \sim \text{outcome} + \text{sequencing depth} + \text{sex} + \text{age} + \text{BMI}$$

Sequencing depth was included to control for the effect of pseudocount addition in clr transformation. Another single model was fit for analysis of sex, age, and BMI as outcomes, as above but without disease outcome.”

13. The disease analysis could potentially be the promising part of the manuscript, but it has several conceptual flaws. First, there are only 25 cohorts included in the figure, out of the original 90 included in the study. First, this should be clearly state in the main text. Furthermore, the authors should better describe in the main text the exclusion criteria that resulted in the loss of the majority of studies. Finally, considering that all the meta-analyses in this study focused on a subset of the original 90 studies, figure 1 is somewhat misleading. Notably, it is confusing as to why so many metagenomic studies of the analyzed diseases are missing.

We thank the reviewer for identifying this unclear aspect of the manuscript. To clarify, we first present an overview of all 94 datasets available in curatedMetagenomicData, and then focus on the subsets relevant for each analysis, such as the subset of only healthy individuals used for analysis of age, and relevant case/control cohorts for meta-analyses of diseases. We better clarify these points in the revised manuscript by clearly stating the inclusion criteria for each analysis:

Results:

“For this analysis, we queried cMD 3 for the stool microbiomes from at least 10 adults with specific diseases and 10 controls. These criteria resulted in 15 diseases (from 30 cohorts), which are: colorectal cancer (CRC), type 2 diabetes (T2D), Crohn’s disease (CD), ulcerative colitis (UC), atherosclerotic cardiovascular disease (ACVD), coronary artery disease (CAD), cirrhosis, schizophrenia, Parkinson’s disease (PD), asthma, migraine, soil-transmitted helminths (STH), coronary artery disease with an episode of heart failure (HF), myalgic encephalomyelitis or chronic fatigue syndrome (ME/CFS), and Behcet disease (BD).”

Methods:

“Inclusion and exclusion criteria

For sex, age, and BMI analyses, datasets *AsnicarF_2021* and *MetaCardis_2020_a* were divided into UK/US parts and Germany/France parts, respectively. Meta-cohorts for analysis of these attributes were selected based on the following common criteria: reported age greater than 16 years (“*age!=NA*” & “*age≥16*”), data collected from stool sample (“*body_site==stool*”),

generally healthy or control in a case-control study (“*study_condition==control*”), baseline or single time point (“*days_from_first_collection in [0.0, NA]*”), and reported BMI (“*BMI!=NA*”). Some additional, feature-specific selection criteria were applied:

1. For analysis of sex-associated microbiomes: at least 25% and 40 samples of each male and female, resulting in 5,505 healthy samples from 21 studies.
2. For analysis of age-associated microbiomes: at least 40 samples and an interquartile range (IQR) of age greater or equal to 15 years, resulting in 4,723 samples from 18 datasets.
3. For analysis of BMI-associated microbiomes: at least 40 samples and an IQR greater than or equal to 3, resulting in 6,361 samples from 32 datasets.
4. For analysis of disease-associated microbiomes: case-control studies with at least 10 cases and 10 controls after excluding individuals with pre-pathological conditions (glucose tolerance and colorectal adenoma), including only samples annotated for age, sex, and BMI. This resulted in 2,300 cases and 2,346 baseline controls from 30 cohorts spanning 15 diseases, including four cardiometabolic diseases.

Two studies (*HMP_2019_ibdmdb* and *NielsenHB_2014*) included samples for both Ulcerative Colitis (UC) and Crohn’s Disease (CD). These diseases were analyzed separately, each with the common control group. Another study (*XieH_2016*) included samples for both asthma and migraine; these were also analyzed separately using the shared controls.

This filtering resulted in the following meta-cohorts:

- Four cardiometabolic diseases (type 2 diabetes (T2D, n=1427), atherosclerotic cardiovascular disease (ACVD, n = 305), coronary artery disease (CAD, n = 563), and coronary artery disease with heart failure (HF, n = 469))
- One psychological pathology (schizophrenia, n = 167)
- One gastrointestinal tract disease having a tumoral character (colorectal cancer (CRC), n = 1300)
- Two gastrointestinal tract autoimmune diseases (Crohn’s disease (CD, n = 309), ulcerative colitis (UC, n = 346),
- One autoimmune non-gastrointestinal tract disease (asthma (n = 200))
- One multisystem inflammatory disease (Behcet disease, BD, n = 65)
- One liver disease (cirrhosis, n = 237)
- Soil-transmitted Helminths (STH, n = 159)
- A partially uncovered pathology ((Institute of Medicine, Board on the Health of Select Populations, and Committee on the Diagnostic Criteria for Myalgic Encephalomyelitis/Chronic Fatigue Syndrome 2015), myalgic encephalomyelitis or chronic fatigue syndrome (ME/CFS), n = 100)
- A partially uncovered etiology disease that involves the brain, though not considered a nervous system disease (migraine, n = 226)

Parkinson's disease (n = 45)."

14. Diseases represented by a single study should not be included in this figure, as in this case this is not really a "meta-analysis". Inclusion of several conditions where the microbiome is almost certainly not causal (e.g., Soil-transmitted Helminths) seems unwarranted.

To clarify, the meta-analysis we are referring to here is the cross-disease analysis, rather than of individual diseases where diseases with a single study could not be part of a meta-analysis. We agree that this analysis is not testing causality of associations, and that clearly infections with helminths is not *caused* by the microbiome. However we have noted widespread discussion of the idea of "dysbiotic" microbiomes and a lack of consensus about what that means. We propose that one quantitative definition of a dysbiotic microbiome is one that either results from or increases susceptibility for multiple adverse health outcomes. For example, a dysbiotic gut microbiome could directly (e.g. through the absence of competitive protective bacteria) and indirectly (e.g. through immune stimulation) affect the likelihood that the pathogen will cause infection, and conversely, infection will likely impact the gut microbiome. We cannot disentangle these causal relationships from cross-sectional observational studies, but identifying reproducible associative links can be formative for future hypothesis building and testing. We have added such a disclaimer to the second paragraph of the Discussion:

"While meta-analysis can, in some situations, reduce the effects of confounding by combining populations with different distributions and effects of confounders, we cannot assess the causality of these associations beyond controlling for age, sex, and BMI. Instead, we present taxonomic and functional shifts in the gut microbiome that can be considered as replicable markers of age, sex, BMI, and disease."

15. Most importantly, it makes little sense to analyze all diseases in this figure together, as their etiology is almost certainly different, and causal microbial associations almost certainly differ between them. The "cross-disease" signal demonstrated is almost certainly a confounding signal - for example, increased use of antibiotics, hospital-exposure, or dietary restriction among sick individuals. It is possible to see this confounding effect in the data, as *Streptococcus anginosus*, a nosocomial pathogen is the most significant 'disease-associated' species. Indeed, for these reasons the cross-disease signal may actually represent the least important species for microbiome causality for any individual disease. This analysis would have to be reworked, or at the least controlled for antibiotic-usage and hospitalization between patients.

We agree with the reviewer that the "cross-disease" signal is not plausibly a common causal factor across these disparate diseases, and the presence of a nosocomial pathogen *Streptococcus anginosus* of the most important disease-associated species is a good example making this point. However, we believe this analysis is valuable for two reasons: 1) it can be used as a quantitative definition of "dysbiosis", as predictive microbiome profiles that are associated disease even if they may have been caused by that disease, by treatments of the disease, or by confounding factors such cause both the disease and alterations of the gut microbiome. 2) It will enable researchers of a single disease to identify elements of their results that are not specific to their disease and are therefore unlikely to be involved in its specific etiology. We have clarified the purpose and interpretation of this analysis, and taken steps

to control for recent exposure to antibiotics, whose strong effects are better addressed by more specific analyses. Specifically, we have:

- Reviewed all the papers of the datasets employed. We found clear information of antibiotics-usage at time of sampling as an exclusion criteria in 18/30 cohorts. We could not find any information in 6 out of the remaining 12. However, 3 out of these 6 were CRC datasets, which are analyzed together with other 7 CRC datasets for which it was known that the patient did undergo any ABX treatment. 6 cohorts in total contain subjects which took antibiotics.
- Data from the MetaCardis_2020 (Forslund et al. 2021) dataset accounted for the largest proportion of samples in three diseases, T2D, CAD, and CAD + HF, and did take ABX. So we adjusted any linear model in which this dataset appears (in the meta-analysis on T2D, also the SankaranarayananK_2015 (Sankaranarayanan et al. 2015) did not exclude samples with antibiotics) by the usage of antibiotics. This has an interesting effect, as it allows to disentangle the effect of the disease to the effect of nosocomial pathogens arising from antibiotic exposure and subsequent ABX-resistance related enhancement.

The new Figure 4 is the following:

Figure 4. Meta-analysis of 15 diseases and 30 cohorts reveals microbial markers of disease or health in 2,346 controls and 2,300 diseased patients - a) the 30 microbial species with the highest meta-analysis coefficient and FDR = 0.01 of disease-associated vs. control samples, with a prevalence of at least 1% in the cohort. Effect sizes are computed as Standardized Mean Differences (SMDs) from a linear

model controlling for sex, age, BMI, sequencing depth, and usage of antibiotics in the type-2 diabetes datasets on centered log-ratio transformed species relative-abundances. Acronyms are colorectal cancer (CRC), Crohn's disease (CD), ulcerative colitis (UC), type-2 diabetes (T2D), atherosclerotic cardiovascular disease (ACVD), Behcet disease (BD), soil-transmitted helminths (STH), myalgic encephalomyelitis or chronic fatigue syndrome (ME/CFS), coronary artery disease (CAD), coronary artery disease with heart failure (CAD+HF), Parkinson's disease (PD). Effect sizes of CRC, CD, UC, & T2D are synthesized in a meta-analysis before the second meta-analysis so that these more frequently studied diseases do not dominate the results over diseases for which a single dataset is available (ACVD, asthma, migraine, STH, cirrhosis, ME/CFS, schizophrenia, BD, CAD, CAD+HF, PD). Yellow shape: individually significant effect ($Q < 0.2$). Light-blue shape: non-individually significant effect. Black diamonds: SMDs between cases and controls synthesized by meta-analysis. Yellow horizontal lines show 95% confidence intervals of the synthesized effect size. **b)** heatmaps showing AUROCs of different Random Forest experiments on the binary discrimination "disease (generic) vs. healthy". Numbers next to study names report the number of cases and controls for each cohort. Cross validations are 10-fold repeated 10 times. Four different "LODO" (Leave-one-dataset-out) AUROCs are shown: 1,2) models trained using all independent datasets (of the same and different diseases) as the test set, using either 1) microbial species relative abundance, or 2) relative abundance plus age, sex, BMI and depth of the sample as features; 3) models trained only on different diseases than the test set; 4) models trained only on the same disease as the test set. The cyan-red-gray bar indicates whether the samples possibly took antibiotics or this information was not available.

16. Figure 5 / oral introgression: Why are p-values used here instead of q-values? The results should be corrected for the number of hypotheses tested (across the 25 studies in this case).

We have amended this.

17. The lack of paired-sample analysis (e.g. analysis of oral and gut samples from the same patient) in this meta-analysis as done in Schmidt et al. 2019 limits interpretation. The authors say "but a quantitative, reproducibly calculable definition is lacking" - but Schmidt et al. 2019 presented a quantitative definition for oral introgression: are the authors able to describe how many paired samples they have across their datasets? Further, discussion of interpretation of oral introgression is lacking. Is this effect simply identifying a decrease in healthy gut anaerobes and therefore an increase in transient oral microbes passing through the gut?

We recognize that the work by Schmidt et al. 2019 was not appropriately acknowledged. To recognize the relevance of the finding of that work, and in parallel to explain what we meant by "quantitative, reproducible definition", we have added a the following paragraph to the Introduction:

"Oral microbiome species can, in some cases, transit across the barrier of the stomach and reach the gut, where they integrate into the gut ecology (Schmidt et al. 2019). Introgression of oral bacterial species in the gut has been associated with colorectal cancer (Thomas et al. 2019; Flemer et al. 2018; Drewes et al. 2017), atherosclerotic cardiovascular disease (ACVD) (Jie et al. 2017), and inflammatory bowel disease (Gevers et al. 2014). Schmidt *et al.* (Schmidt et al. 2019) observed evidence of oral-to-gut transition of microbial strains in both healthy and diseased subjects. These findings motivate a quantitative definition of the extent of oral-to-gut microbial introgression and a systematic investigation of its potential role across a range of diseases."

We have stated clearly a difference between our and previous strain-level methodologies:

"Increased oral-to-gut introgression of microbes (i.e. increased passage of oral-typical species in the intestine) in response to physico-chemical alterations of the intestinal lumen in inflammation has been postulated in several diseases (Thomas et al. 2019; Jie et al. 2017; Flemer et al. 2018). Recent strain-level analysis has clearly demonstrated the introgression

of individual strains in individuals (Schmidt et al. 2019), but a quantitative and conveniently calculable definition of the total accumulated amount of introgression, based on microbiome taxonomic profiling, is lacking.”

In comparison with the studies mentioned in this new paragraph, we have expanded this notion to many more datasets and diseases, showing that a simple (i.e. general) definition of oral to gut introgression is sufficient to capture the association of this phenomenon with multiple adverse health outcomes.

We also tested a plausible alternative to this score: select all oral species that are present in fewer than 10% of matched gut samples. This identified 252 species (vs the 305 species currently used in the manuscript) from 1,053 samples from same-individual paired gut/oral samples. Compared to our current score, this alternative one is similar (Spearman’s rho = 0.72).

Rebuttal Figure 3. Cumulative relative abundance of typically oral taxa in the human gut microbiome from all oral samples and from paired oral/gut samples as potential indicators of disease.

Rebuttal Fig. 4 represents the same concepts of the newly designed **Fig.5**, computing the oral to gut introgression score with this other rationale. While slightly more conservative, this new score may be less transportable, as it is based on fewer species identified from fewer individuals.

Rebuttal Figure 4. Cumulative relative abundance of typically oral taxa in the human gut microbiome from paired oral/gut samples is a potential indicator of disease - score is computed by summing relative abundances of typical oral species from oral/gut paired samples; once a species is chosen, it is excluded if present in more than 10% of the corresponding gut samples **a)** distribution of the per-population mean score in 6,891 gut microbiomes from adult, healthy individuals (25 studies), and 3,632 gut microbiomes (48 studies) from adults who have received a specific diagnosis. Asterisks mark the between-distribution Mann-Whitney $P < 0.05$. Standardized Mean Difference of the per-threshold average score between the two conditions is reported. **b)** boxplots showing the log-10 distributions of oral-cavity typical microbial species cumulative abundance in 14 diseases (20 cohorts), divided by disease (orange) and controls (blue). Asterisks mark the Mann-Whitney $Q < 0.1$. Standardized Mean Differences of the introgression score against disease versus healthy conditions are presented. **c)** boxplots showing the log-10 distributions of oral-cavity typical microbial species cumulative abundance in 10 colorectal cancer cohorts, divided by the disease (orange) and control (blue) groups. Asterisks mark the Mann-Whitney $Q < 0.1$. Standardized Mean Differences of the introgression score against disease versus healthy conditions are presented.

Additional comments

Figure 2: The meta-analysis plots in this Figure and Figure 3-4 are extremely busy and difficult to interpret. Many points are overlaid upon each other, and the horizontal bars are almost impossible to see. These figures need to be altered for readability.

We thank the reviewer for having pointed this out. We have altered the figure to improve readability.

Figure 2. Meta-analysis of the gut microbiome from 5,505 individuals (3,288 females and 2,217 males) across 18 datasets, revealing sex-associated microbial differences in healthy adults based on stool samples. a) The 30 microbial species and genera with the highest SMD meta-analysis coefficient (FDR = 0.01) between sexes. Effect sizes are calculated as SMDs from a linear model controlling for age, BMI, and sequencing depth, applied to centered log-ratio transformed species relative abundances. Yellow: significant effect size (FDR = 0.2). Light-blue: non-significant effect size. Black diamonds: SMD between male and female. Yellow horizontal lines

indicate the 95% confidence intervals of the effect size from the meta-analysis. **b)** Mean relative abundance distribution of the 30 taxa in the 18 datasets, grouped by sex. The Y-axis is in log10 scale. **c)** AUROCs of a leave-one-dataset-out (LODO) validation predicting sex using a Random Forest algorithm trained on various features: (i) age, BMI, sequencing depth of the samples; (ii) species relative-abundances, with and without (iii) age, BMI and sequencing depth as features; (iv) KEGG-level-collapsed UniRef90 gene families abundances with and without (v) age, BMI and sequencing depth as features. On top: the number of female and male participants in each study.

AUROC should be written as AUROC throughout the text.

We have changed any occurrence of AUROC with AUROC.

Reviewer #3 (Remarks to the Author):

The article “Meta-analysis of 29,533 human metagenomes defines an index of oral to gut microbial introgression and associations with age, sex, BMI, and diseases” by Manghi and colleagues reanalyzes existing microbiome data sets, and performs several supervised correlation analyses across the different individual studies comprised in the latest iteration of the curated Metagenomic Data set version 3.

The paper presents incremental new insights into the associations between age, sex, BMI and several disease conditions with bacterial populations in stool samples. Furthermore, the authors propose the sum of 130 bacterial species’ relative abundances as a novel microbiome health score.

I found the paper of modest interest. I largely believe that most analyses were conducted correctly, but some of the applied statistical analyses could benefit from a better justification and explanation. I found the microbiome health score unconvincing. Overall, novel biological insights are limited. The paper could benefit from editorial input regarding clarity, the many very long sentences, and sometimes off-putting formatting.

The term “microbial introgression” is used in the title and name of the microbiome health score. The term “introgression” is unfamiliar to me in a microbiome context, the provided references discussing oral microbiome taxa in the gut use “transmission”, which seems more appropriate, and the term introgression has different meanings elsewhere. Perhaps the authors can at least define what they mean by it, or better still, consider replacing with a more appropriate term: they mean that DNA of oral species are found in the stool.

We would prefer to avoid mentioning “microbial transmission”, as the present analysis is not based on strain-level approaches such as others we have performed (Valles-Colomer et al. 2023; Ianiro et al. 2022). We don’t believe the approach here can be used to infer transmission.

In order to clarify this aspect, we now state at the beginning of this paragraph:

“Increased oral-to-gut introgression of microbes (i.e. increased passage of oral-typical species in the intestine) in response to physico-chemical alterations of the intestinal lumen in inflammation has been postulated in several diseases (Thomas et al. 2019; Jie et al. 2017; Flemer et al. 2018).”

In the revised Introduction, a new paragraph now describes the process we are referring to and the premise for the proposed score:

“Oral microbiome species can, in some cases, transit across the barrier of the stomach and reach the gut, where they integrate into the gut ecology (Schmidt et al. 2019). Introgression of oral bacterial species in the gut has been associated with colorectal cancer (Thomas et al. 2019; Flemer et al. 2018; Drewes et al. 2017), atherosclerotic cardiovascular disease (ACVD) (Jie et al. 2017), and inflammatory bowel disease (Gevers et al. 2014). Schmidt *et al.* (Schmidt et al. 2019) observed evidence of oral-to-gut transition of microbial strains in both healthy and diseased subjects. These findings motivate a quantitative definition of the extent of oral-to-gut microbial introgression and a systematic investigation of its potential role across a range of diseases.”

I have a few specific comments:

L66: “increase” relative to what?

We thank the reviewer for pointing out this. To clarify better, we have substituted this sentence with the following:

“[...] One of the most interesting and consistent findings has been a progressive increase in the gut microbial diversity of longevous populations (Wilmanski et al. 2021; Zhang et al. 2021), [...]”

L70: the sentence is too long, and the “or to the beneficial role...” part is phrased confusingly in the context of the rest of the sentence.

We agree with the reviewer about the confusion generated by this sentence. We have revised to clarify that elderly microbiomes may appear as beneficial as a result of selection bias. We have splitted the sentence and modified the problematic statement as follows:

Before:

“The relationship between aging and the human microbiome has been extensively studied; the adjustment for age, for example, can improve the identification of gut microbiome associations with disease (Ghosh et al. 2020), but consistent age-associated microbiome changes remain difficult to define. One of the most interesting and consistent findings is an increase in the gut microbial diversity of longevous populations (Wilmanski et al. 2021; X. Zhang et al. 2021), although it can be attributed to increased presence of pathobionts and a possible introgression of oral microbes (Biagi et al. 2010; X. Zhang et al. 2021), or to the beneficial role of the elderly gut microbiome (Kong et al. 2019; Biagi et al. 2016).”

Now:

“The relationship between aging and the human microbiome has been extensively studied; adjustment for age, for example, can improve the identification of gut microbiome associations with disease (Ghosh et al. 2020). Consistent age-associated microbiome changes, however, remain difficult to define. One of the most interesting and

consistent findings has been a progressive increase in the gut microbial diversity of longevous populations (Wilmanski et al. 2021; Zhang et al. 2021), although this can be attributed to increased presence of pathobionts and a possible introgression of oral microbes (Biagi et al. 2010; Zhang et al. 2021), or to selection bias (Kong et al. 2019; Biagi et al. 2016).”

Figure 2: How did you standardize? Did you z-score the CLR?

We standardized for meta-analysis using standardized mean differences of CLR-transformed relative abundances (difference in samples means divided by standard deviation). CLR is computed over sample vectors, while standardized mean difference is computed over the feature vectors, which are approximately normal following CLR transformation.

Figure 2: what are “cases” and “controls” in an analysis of “sex”?

We thank the reviewer. Cases and controls were substituted in the caption of figure 2 with males and females, respectively.

Figure 2: what are raw abundances? The figure axis says log₂, the caption log₁₀

We thank the reviewer. Now both say log₁₀.

L657: “raw abundances were scaled to the 0 1 range” --- Did you scale within a taxon (i.e. min max scaling) or per sample? In the latter case, did you include a “other taxa” category as a sum of the excluded taxa, and if not, how do you justify the CLR transform? In the former case of “min max” scaling, why apply the CLR?

The compositional vector (of the sample) is provided by MetaPhlAn as *per-sample* normalized relative abundances between 0 and 100, which we divided by 100 to produce a 0 1 range as per the default of scikit-bio. The MetaPhlAn relative abundance output normalizes for factors such as library size and number and length of DNA marker sequences in each species, whereas CLR attempts to correct for compositionality of microbiome samples. As a side note, as this is the implicit behavior of the scikit-bio library, this statement is no longer in the manuscript.

L189: where is the figure presenting these results? Besides significance, what is the meaning of the effect sizes of difference that seem very small between the sexes.

The effect sizes presented are standardized mean differences, which equals the proportion of difference between the two means divided by pooled standard deviation. While counterintuitive at first, it is preferable to direct use of CLR transformed values for which there is no direct interpretation of the effect size. The reviewer’s observation that these effect sizes are small is correct: unexplained person-to-person variability in the observed gut microbiome species is larger than any of these associations. This is in a way similar to the small effect sizes observed in GWAS, and also similarly, polymicrobial scores can be more predictive, as seen in AUROCs in our machine learning analysis.

L222: Related to the above, given that the authors largely find male and female microbiomes to be very similar, would perhaps the opposite conclusion, that sex is not an important variable in microbiome studies, be justified? Can the authors discuss this better for future reference?

Although the effect sizes of individual taxa are small, machine learning analysis supports sex as a relevant covariate of the overall microbiome (AUROC = 0.7 in LODO). So we take the reviewer's point, and conclude that sex (as well as age and BMI) may or may not be important depending on the purpose of an analysis. And of course the causality of this relationship is unclear, and could be caused by differences in diet, lifestyle, social exposures, and/or differences in host biology. We have elaborated on effect size in the Discussion:

“The effect sizes for individual taxa are generally small for age, BMI, and sex: less than 0.2 Standardized Mean Difference (SMD), meaning for example that the relative abundance of a taxon in one population exceeds the median of the other population in 58% of individuals instead of 50%⁷³ However, with the large number of associated taxa (or functional modules), it is still possible to construct reasonably accurate machine learning models, with mean AUROC in LODO independent validation in the range of 0.7. Associations with any disease state compared to healthy control participants were stronger, with the largest SMD at 0.35, mean AUROC in LODO of 0.7 across all diseases, above 0.80 for the most predictable diseases, and 0.80 across all diseases for the “oral introgression” score. While meta-analysis can, in some situations, reduce the effects of confounding by combining populations with different distributions and effects of confounders, we cannot assess the causality of these associations beyond controlling for age, sex, and BMI. Instead, we present taxonomic and functional shifts in the gut microbiome that can be considered as replicable associations, across multiple studies in diverse locations, with age, sex, BMI, and disease.”

L277: What does “correlations were synthesized” mean?

We borrow the term “synthesize” from the meta-analysis literature (e.g. (Bakbergenuly and Kulinskaya 2018; Waldron et al. 2014; Borenstein et al. 2011)) to refer to the operation of weighted averaging over different studies to synthesize their effect-size estimates into a single estimate (by inverse variance weighting, see answer to comment N.2, 6 of reviewers N. 2).

L282: if diversity goes up with age, would therefore a negative association of many species with age be expected due to compositional effects?

Since we compute linear models after having performed Centered Log-ratio Transformation, we do not expect these results to be driven by compositionality, even in the presence of dominant species. However, on untransformed relative abundances, one could expect negative association for species that are dominant at younger ages, and positive association for lower-abundance species that become more-observed with the loss of dominant species.

L330: rephrase “a hallmark of good health” cannot be established with the current study

We have eliminated references to “a hallmark of good health” and now only refer to associations with disease state, adverse health outcomes, or healthy control participants.

L341: the model setup should be described in more detail in the methods

We thank the reviewer for this suggestion. We agree on providing a clearer methodological description of this procedure. We have accordingly modified the text of the Methods section “**Meta-analysis of 15 diseases**” as follows:

Before:

“We performed 2 analyses, on species-level taxonomic entries and on MetaCyc pathways. Briefly, the diseases for which more than one dataset was available through cMD 3 (10 CRC datasets, 3 CD datasets, 2 UC datasets and 4 T2D datasets) were analyzed via the procedure described. The remaining 11 diseases for which only a single dataset was available (asthma, migraine, ACVD, STH, cirrhosis, schizophrenia, ME/CFS, BD, CAD, CAD + HF, PD) were analyzed following the formulas (1.) and (2.). The 15 coefficients (4 from a meta-analysis, 11 from a single linear model) were then summarized in a meta-analysis in which the diseases for which multiple datasets appear with a single coefficient (Standardized Mean Difference) which corresponds to the best linear unbiased estimator (BLUE) (Borenstein et al. 2011).”

Now:

“Standardized Mean Difference (SMD) for binary outcomes

We used SMD (Nakagawa and Cuthill 2007) as a scale-free measure of association that can be synthesized across multiple studies. SMD was calculated for each study from the relevant regression coefficient as:

$$SMD = \frac{t \times (n1 + n2)}{\sqrt{n + n2} \times \sqrt{n1 + n2 - 2}} \quad (1)$$

where t is the t-score, and $n1$ and $n2$ are sample sizes for control and disease, respectively. The standard error (SE) of SMD was computed as:

$$SE = \sqrt{\frac{n1 + n2 - 1}{n1 + n2 - 3} \cdot \frac{4}{n1 + n2} \cdot \left(1 + \frac{SMD^2}{8}\right)} \quad (2)$$

Correlation for continuous outcomes

Correlation coefficients were computed from regression tables as (Nakagawa and Cuthill 2007):

$$r = \frac{\sqrt{t}}{(t^2 + n - 1)} \quad (3)$$

where t is the t-value and n is the number of samples in the control group. The correlation coefficients were Fisher-Z transformed for meta-analysis, then the inverse of the Fisher-Z function was applied to the synthesized estimate and its confidence intervals.

Meta-analysis

We used random-effects models to synthesize SMD and partial correlation coefficients, and estimated between-dataset heterogeneity using Paule and Mandel's and Cochran's generalized Q-statistic to assess significant heterogeneity (Veroniki et al. 2016), using the Python packages *statsmodels*, *skbio*, and *scipy*.

The standard error for combined effect size (SEm) was calculated as:

$$SEm = \frac{1}{\sqrt{\sum_{i=1}^k W_i}} \quad (4)$$

where k is the number of studies, and W_i is the weight of the i -th study, which corresponds to the inverse of the coefficient estimated variance plus the between-study variance estimation (Paule and Mandel random-effect meta-analysis).

Meta-analysis of diseases Whereas microbiome associations with age, sex, and BMI were synthesized using standard meta-analysis methods as described above, we employed a two-step method to identify biomarkers of disease vs. health. In step 1, we performed a meta-analysis of four multi-dataset diseases (10 CRC, 3 CD, 2 UC, 4 T2D) to generate a single estimate and standard error for each of these four diseases. In step 2, we performed a meta-analysis to synthesize these 4 coefficients and of 11 diseases represented by a single dataset, treating disease as a single outcome (ie, "disease" vs control). "

Other aspects of the usage of linear regression are discussed in previous sections of the Methods.

L379: rephrase to "health associated microbiome"

We have made this correction.

Figure 5: please resolve colors in a figure legend

We have made this correction.

L 413: why write "cumulative" instead of "sum"?

We have replaced "cumulative" with "summed relative abundance".

L418: 3/10 and 5/15 are not very convincing numbers. Can the authors perhaps create a randomized control score for comparison? E.g. multiple times, select 130 other taxa with somewhat similar abundance profiles to the 130 oral species, and count in how many disease studies these control scores are significant predictors? This could help the authors convince the reader of the impact of their novel score and finding.

The numbers 3/10 and 5/15 (now: 9/10 and 8/20, reflecting the greater transportability of the newly defined/validated score, currently based on 305 species, 2.4 times larger) are not indicative of the general tendency of the score. That is indicated by the binomial-test P-value (whose null hypothesis is that the the score is higher in cases 50% of the times), and by the meta-analysis P-value (whose null hypothesis is that the score obtaining as the average of all scores is zero, when considering also the

intra- and inter-dataset dispersions). As we performed at least 2 different tests on the overall association in the 30 studies (binomial test and meta-analysis Z-score), we would prefer not to add other tests.

L419 and L431: What is the difference between the data from the 15 non CRC disease cohorts (L419) and 12 diseases (L432)?

The 12 diseases included also CRC, which was initially separated for clarity. This has been partially changed in the updated manuscript which accounts for 15 diseases.

It was not clear to me if this paper is the official release publication of cMD 3. Please clarify.

We agree with the reviewer that this was not very clear. We now make explicit in the Introduction that this is the official presentation paper of curatedMetagenomicData 3. The sentence referring to this, placed in the last Introduction paragraph, is:

“Here we present version 3 of curatedMetagenomicData (Pasolli et al. 2017) (cMD 3), an expansion and refactoring of the original resource, providing 94 shotgun metagenomic datasets with manually-curated metadata from 42 countries and 6 continents. This version provides 22,710 samples (3.6 times larger than version 1, including 3.5 times more studies) with updated taxonomic and functional potential dedicated tools, with expanded manually curated metadata on more than 100 different individual-participant characteristics. cMD 3 provides a higher level of manual curation than alternatives (Dai et al. 2022; Kasmanas et al. 2020), allowing adjustment for some potential confounding factors in meta-analysis.”

Bibliography

- Armour, Courtney R., Stephen Nayfach, Katherine S. Pollard, and Thomas J. Sharpton. 2019. “A Metagenomic Meta-Analysis Reveals Functional Signatures of Health and Disease in the Human Gut Microbiome.” *mSystems* 4 (4). <https://doi.org/10.1128/mSystems.00332-18>.
- Asnicar, Francesco, Sarah E. Berry, Ana M. Valdes, Long H. Nguyen, Gianmarco Piccinno, David A. Drew, Emily Leeming, et al. 2021. “Microbiome Connections with Host Metabolism and Habitual Diet from 1,098 Deeply Phenotyped Individuals.” *Nature Medicine* 27 (2): 321–32.
- Asnicar, Francesco, Emily R. Leeming, Eirini Dimidi, Mohsen Mazidi, Paul W. Franks, Haya Al Khatib, Ana M. Valdes, et al. 2021. “Blue Poo: Impact of Gut Transit Time on the Gut Microbiome Using a Novel Marker.” *Gut* 70 (9): 1665–74.
- Bakbergenuly, Ilyas, and Elena Kulinskaya. 2018. “Meta-Analysis of Binary Outcomes via Generalized Linear Mixed Models: A Simulation Study.” *BMC Medical Research Methodology* 18 (1): 70.
- Beghini, Francesco, Lauren J. McIver, Aitor Blanco-Míguez, Leonard Dubois, Francesco Asnicar, Sagun Maharjan, Ana Mailyan, et al. 2021. “Integrating Taxonomic, Functional, and Strain-Level Profiling of Diverse Microbial Communities with bioBakery 3.” *eLife* 10 (May). <https://doi.org/10.7554/eLife.65088>.
- Biagi, Elena, Claudio Franceschi, Simone Rampelli, Marco Severgnini, Rita Ostan, Silvia Turroni, Clarissa Consolandi, et al. 2016. “Gut Microbiota and Extreme Longevity.” *Current Biology: CB* 26 (11): 1480–85.
- Biagi, Elena, Lotta Nyland, Marco Candela, Rita Ostan, Laura Bucci, Elisa Pini, Janne Nikkila, et al. 2010. “Through Ageing, and beyond: Gut Microbiota and Inflammatory Status in Seniors and Centenarians.” *PloS One* 5 (5): e10667.

- Borenstein, Michael, Larry V. Hedges, Julian P. T. Higgins, and Hannah R. Rothstein. 2011. *Introduction to Meta-Analysis*. John Wiley & Sons.
- Bui, Thi Phuong Nam, Antonio Dario Troise, Bart Nijssse, Giovanni N. Roviello, Vincenzo Fogliano, and Willem M. de Vos. 2020. "Intestinimonas-like Bacteria Are Important Butyrate Producers That Utilize N ϵ -Fructosyllysine and Lysine in Formula-Fed Infants and Adults." *Journal of Functional Foods*. <https://doi.org/10.1016/j.jff.2020.103974>.
- Castañer, Olga, and Helmut Schröder. 2018. "Response to: Comment on "The Gut Microbiome Profile in Obesity: A Systematic Review."" *International Journal of Endocrinology*.
- Cuesta-Zuluaga, Jacobo de la, Scott T. Kelley, Yingfeng Chen, Juan S. Escobar, Noel T. Mueller, Ruth E. Ley, Daniel McDonald, et al. 2019. "Age- and Sex-Dependent Patterns of Gut Microbial Diversity in Human Adults." *mSystems* 4 (4). <https://doi.org/10.1128/mSystems.00261-19>.
- Cuevas-Sierra, Amanda, Ana Romo-Hualde, Paula Aranaz, Leticia Goni, Marta Cuervo, J. Alfredo Martínez, Fermín I. Milagro, and José I. Riezu-Boj. 2021. "Diet- and Sex-Related Changes of Gut Microbiota Composition and Functional Profiles after 4 Months of Weight Loss Intervention." *European Journal of Nutrition* 60 (6): 3279–3301.
- Dai, Die, Jiaying Zhu, Chuqing Sun, Min Li, Jinxin Liu, Sicheng Wu, Kang Ning, Li-Jie He, Xing-Ming Zhao, and Wei-Hua Chen. 2022. "GMrepo v2: A Curated Human Gut Microbiome Database with Special Focus on Disease Markers and Cross-Dataset Comparison." *Nucleic Acids Research* 50 (D1): D777–84.
- Dao, Maria Carlota, Amandine Everard, Judith Aron-Wisnewsky, Nataliya Sokolovska, Edi Prifti, Eric O. Verger, Brandon D. Kayser, et al. 2016. "Akkermansia Muciniphila and Improved Metabolic Health during a Dietary Intervention in Obesity: Relationship with Gut Microbiome Richness and Ecology." *Gut* 65 (3): 426–36.
- De Filippis, Francesca, Edoardo Pasolli, Adrian Tett, Sonia Tarallo, Alessio Naccarati, Maria De Angelis, Erasmo Neviani, et al. 2019. "Distinct Genetic and Functional Traits of Human Intestinal Prevotella Copri Strains Are Associated with Different Habitual Diets." *Cell Host & Microbe* 25 (3): 444–53.e3.
- Degen, L. P., and S. F. Phillips. 1996. "Variability of Gastrointestinal Transit in Healthy Women and Men." *Gut* 39 (2): 299–305.
- Depommier, Clara, Amandine Everard, Céline Druart, Hubert Plovier, Matthias Van Hul, Sara Vieira-Silva, Gwen Falony, et al. 2019. "Supplementation with Akkermansia Muciniphila in Overweight and Obese Human Volunteers: A Proof-of-Concept Exploratory Study." *Nature Medicine* 25 (7): 1096–1103.
- Derosa, Lisa, Bertrand Routy, Andrew Maltez Thomas, Valerio Iebba, Gerard Zalcman, Sylvie Friard, Julien Mazieres, et al. 2022. "Intestinal Akkermansia Muciniphila Predicts Clinical Response to PD-1 Blockade in Patients with Advanced Non-Small-Cell Lung Cancer." *Nature Medicine*, February. <https://doi.org/10.1038/s41591-021-01655-5>.
- Derosa, Lisa, Bertrand Routy, Laurence Zitvogel, Andrew M. Thomas, Gerard Zalcman, Sylvie Friard, Julien Mazieres, et al. 2021. "Intestinal Akkermansia Muciniphila Predicts Overall Survival in Advanced Non-Small Cell Lung Cancer Patients Treated with Anti-PD-1 Antibodies: Results a Phase II Study." *Journal of Clinical Oncology*. https://doi.org/10.1200/jco.2021.39.15_suppl.9019.
- Drewes, Julia L., James R. White, Christine M. Dejea, Payam Fathi, Thevambiga Iyadorai, Jamuna Vadivelu, April C. Roslani, et al. 2017. "High-Resolution Bacterial 16S rRNA Gene Profile Meta-Analysis and Biofilm Status Reveal Common Colorectal Cancer Consortia." *NPJ Biofilms and Microbiomes* 3 (November): 34.
- Duvallet, Claire, Sean M. Gibbons, Thomas Gurry, Rafael A. Irizarry, and Eric J. Alm. 2017. "Meta-Analysis of Gut Microbiome Studies Identifies Disease-Specific and Shared Responses." *Nature Communications* 8 (1): 1784.
- Flemer, Burkhardt, Ryan D. Warren, Maurice P. Barrett, Katryna Cisek, Anubhav Das, Ian B. Jeffery, Eimear Hurley, Micheal O'Riordain, Fergus Shanahan, and Paul W. O'Toole. 2018. "The Oral Microbiota in Colorectal Cancer Is Distinctive and Predictive." *Gut* 67 (8): 1454–63.

- Forslund, Sofia K., Rima Chakaroun, Maria Zimmermann-Kogadeeva, Lajos Markó, Judith Aron-Wisniewsky, Trine Nielsen, Lucas Moitinho-Silva, et al. 2021. "Combinatorial, Additive and Dose-Dependent Drug-Microbiome Associations." *Nature* 600 (7889): 500–505.
- Gevers, Dirk, Subra Kugathasan, Lee A. Denson, Yoshiki Vázquez-Baeza, Will Van Treuren, Boyu Ren, Emma Schwager, et al. 2014. "The Treatment-Naive Microbiome in New-Onset Crohn's Disease." *Cell Host & Microbe* 15 (3): 382–92.
- Ghosh, Tarini S., Mrinmoy Das, Ian B. Jeffery, and Paul W. O'Toole. 2020. "Adjusting for Age Improves Identification of Gut Microbiome Alterations in Multiple Diseases." *eLife* 9 (March).
<https://doi.org/10.7554/eLife.50240>.
- Ianiro, Gianluca, Michal Punčochář, Nicolai Karcher, Serena Porcari, Federica Armanini, Francesco Asnicar, Francesco Beghini, et al. 2022. "Variability of Strain Engraftment and Predictability of Microbiome Composition after Fecal Microbiota Transplantation across Different Diseases." *Nature Medicine* 28 (9): 1913–23.
- Institute of Medicine, Board on the Health of Select Populations, and Committee on the Diagnostic Criteria for Myalgic Encephalomyelitis/Chronic Fatigue Syndrome. 2015. *Beyond Myalgic Encephalomyelitis/Chronic Fatigue Syndrome: Redefining an Illness*. National Academies Press.
- Jie, Zhuye, Huihua Xia, Shi-Long Zhong, Qiang Feng, Shenghui Li, Suisha Liang, Huanzi Zhong, et al. 2017. "The Gut Microbiome in Atherosclerotic Cardiovascular Disease." *Nature Communications* 8 (1): 845.
- Karcher, Nicolai, Eleonora Nigro, Michal Punčochář, Aitor Blanco-Míguez, Matteo Ciciani, Paolo Manghi, Moreno Zolfo, et al. 2021. "Genomic Diversity and Ecology of Human-Associated Akkermansia Species in the Gut Microbiome Revealed by Extensive Metagenomic Assembly." *Genome Biology* 22 (1): 209.
- Kasmanas, Jonas Coelho, Alexander Bartholomäus, Felipe Borim Corrêa, Tamara Tal, Nico Jehmlich, Gunda Herberth, Martin von Bergen, Peter F. Stadler, André Carlos Ponce de Leon Ferreira de Carvalho, and Ulisses Nunes da Rocha. 2020. "HumanMetagenomeDB: A Public Repository of Curated and Standardized Metadata for Human Metagenomes." *Nucleic Acids Research*, November.
<https://doi.org/10.1093/nar/gkaa1031>.
- Kayser, Brandon D., Marie Lhomme, Edi Prifti, Carla Da Cunha, Florian Marquet, Florian Chain, Isabelle Naas, et al. 2019. "Phosphatidylglycerols Are Induced by Gut Dysbiosis and Inflammation, and Favorably Modulate Adipose Tissue Remodeling in Obesity." *FASEB Journal: Official Publication of the Federation of American Societies for Experimental Biology* 33 (4): 4741–54.
- Kong, Fanli, Feilong Deng, Ying Li, and Jiangchao Zhao. 2019. "Identification of Gut Microbiome Signatures Associated with Longevity Provides a Promising Modulation Target for Healthy Aging." *Gut Microbes* 10 (2): 210–15.
- Lee, Karla A., Andrew Maltez Thomas, Laura A. Bolte, Johannes R. Björk, Laura Kist de Ruijter, Federica Armanini, Francesco Asnicar, et al. 2022. "Cross-Cohort Gut Microbiome Associations with Immune Checkpoint Inhibitor Response in Advanced Melanoma." *Nature Medicine* 28 (3): 535–44.
- McCulloch, John A., Diwakar Davar, Richard R. Rodrigues, Jonathan H. Badger, Jennifer R. Fang, Alicia M. Cole, Ascharya K. Balaji, et al. 2022. "Intestinal Microbiota Signatures Predict Clinical Outcome and Immune-Related Adverse Events in PD-1 Treated Melanoma Patients." *Nature Medicine*, March.
- Nakagawa, Shinichi, and Innes C. Cuthill. 2007. "Effect Size, Confidence Interval and Statistical Significance: A Practical Guide for Biologists." *Biological Reviews of the Cambridge Philosophical Society* 82 (4): 591–605.
- Pasolli, Edoardo, Lucas Schiffer, Paolo Manghi, Audrey Renson, Valerie Obenchain, Duy Tin Truong, Francesco Beghini, et al. 2017. "Accessible, Curated Metagenomic Data through ExperimentHub." *Nature Methods* 14 (11): 1023–24.
- Pasolli, Edoardo, Duy Tin Truong, Faizan Malik, Levi Waldron, and Nicola Segata. 2016. "Machine Learning Meta-Analysis of Large Metagenomic Datasets: Tools and Biological Insights." *PLoS Computational Biology* 12 (7): e1004977.

- Pinart, Mariona, Andreas Dötsch, Kristina Schlicht, Matthias Laudes, Jildau Bouwman, Sofia K. Forslund, Tobias Pischon, and Katharina Nimpf. 2021. "Gut Microbiome Composition in Obese and Non-Obese Persons: A Systematic Review and Meta-Analysis." *Nutrients*. <https://doi.org/10.3390/nu14010012>.
- Sankaranarayanan, Krithivasan, Andrew T. Ozga, Christina Warinner, Raul Y. Tito, Alexandra J. Obregon-Tito, Jiawu Xu, Patrick M. Gaffney, et al. 2015. "Gut Microbiome Diversity among Cheyenne and Arapaho Individuals from Western Oklahoma." *Current Biology: CB* 25 (24): 3161–69.
- Schirmer, Melanie, Sanne P. Smekens, Hera Vlamakis, Martin Jaeger, Marije Oosting, Eric A. Franzosa, Rob Ter Horst, et al. 2016. "Linking the Human Gut Microbiome to Inflammatory Cytokine Production Capacity." *Cell* 167 (7): 1897.
- Schmidt, Thomas Sb, Matthew R. Hayward, Luis P. Coelho, Simone S. Li, Paul I. Costea, Anita Y. Voigt, Jakob Wirbel, et al. 2019. "Extensive Transmission of Microbes along the Gastrointestinal Tract." *eLife* 8 (February). <https://doi.org/10.7554/eLife.42693>.
- Sinha, Trishla, Arnau Vich Vila, Sanzhima Garmaeva, Soesma A. Jankipersadsing, Floris Imhann, Valerie Collij, Marc Jan Bonder, et al. 2019. "Analysis of 1135 Gut Metagenomes Identifies Sex-Specific Resistome Profiles." *Gut Microbes* 10 (3): 358–66.
- Su, Qi, Qin Liu, Raphaela Iris Lau, Jingwan Zhang, Zhilu Xu, Yun Kit Yeoh, Thomas W. H. Leung, et al. 2022. "Faecal Microbiome-Based Machine Learning for Multi-Class Disease Diagnosis." *Nature Communications* 13 (1): 6818.
- Taur, Ying, and Eric G. Pamer. 2016. "Microbiome Mediation of Infections in the Cancer Setting." *Genome Medicine* 8 (1): 40.
- Tett, Adrian, Kun D. Huang, Francesco Asnicar, Hannah Fehlner-Peach, Edoardo Pasolli, Nicolai Karcher, Federica Armanini, et al. 2019. "The Prevotella Copri Complex Comprises Four Distinct Clades Underrepresented in Westernized Populations." *Cell Host & Microbe* 26 (5): 666–79.e7.
- Thomas, Andrew Maltez, Paolo Manghi, Francesco Asnicar, Edoardo Pasolli, Federica Armanini, Moreno Zolfo, Francesco Beghini, et al. 2019. "Metagenomic Analysis of Colorectal Cancer Datasets Identifies Cross-Cohort Microbial Diagnostic Signatures and a Link with Choline Degradation." *Nature Medicine* 25 (4): 667–78.
- Tierney, Braden T., Yingxuan Tan, Zhen Yang, Bing Shui, Michaela J. Walker, Benjamin M. Kent, Aleksandar D. Kostic, and Chirag J. Patel. 2022. "Systematically Assessing Microbiome–disease Associations Identifies Drivers of Inconsistency in Metagenomic Research." *PLOS Biology*. <https://doi.org/10.1371/journal.pbio.3001556>.
- Valles-Colomer, M., A. Blanco-Míguez, P. Manghi, F. Asnicar, L. Dubois, D. Golzato, F. Armanini, et al. 2023. "The Person-to-Person Transmission Landscape of the Gut and Oral Microbiomes." *Nature*, January.
- Veroniki, Areti Angeliki, Dan Jackson, Wolfgang Viechtbauer, Ralf Bender, Jack Bowden, Guido Knapp, Oliver Kuss, Julian P. T. Higgins, Dean Langan, and Georgia Salanti. 2016. "Methods to Estimate the between-Study Variance and Its Uncertainty in Meta-Analysis." *Research Synthesis Methods* 7 (1): 55–79.
- Waldron, Levi, Benjamin Haibe-Kains, Aedín C. Culhane, Markus Riester, Jie Ding, Xin Victoria Wang, Mahnaz Ahmadifar, et al. 2014. "Comparative Meta-Analysis of Prognostic Gene Signatures for Late-Stage Ovarian Cancer." *Journal of the National Cancer Institute* 106 (5). <https://doi.org/10.1093/jnci/dju049>.
- Wang, Dong D., Long H. Nguyen, Yanping Li, Yan Yan, Wenjie Ma, Ehud Rinott, Kerry L. Ivey, et al. 2021. "The Gut Microbiome Modulates the Protective Association between a Mediterranean Diet and Cardiometabolic Disease Risk." *Nature Medicine* 27 (2): 333–43.
- Wilmanski, Tomasz, Christian Diener, Noa Rappaport, Sushmita Patwardhan, Jack Wiedrick, Jodi Lapidus, John C. Earls, et al. 2021. "Gut Microbiome Pattern Reflects Healthy Ageing and Predicts Survival in Humans." *Nature Metabolism* 3 (2): 274–86.
- Wirbel, Jakob, Paul Theodor Pyl, Ece Kartal, Konrad Zych, Alireza Kashani, Alessio Milanese, Jonas S. Fleck, et al. 2019. "Meta-Analysis of Fecal Metagenomes Reveals Global Microbial Signatures That Are Specific for

Colorectal Cancer." *Nature Medicine* 25 (4): 679–89.

Zhang, Xiuying, Huanzi Zhong, Yufeng Li, Zhun Shi, Huahui Ren, Zhe Zhang, Xianghai Zhou, et al. 2021. "Sex- and Age-Related Trajectories of the Adult Human Gut Microbiota Shared across Populations of Different Ethnicities." *Nature Aging*. <https://doi.org/10.1038/s43587-020-00014-2>.

Zhang, Yuqing, Christoph Bernau, Giovanni Parmigiani, and Levi Waldron. 2018. "The Impact of Different Sources of Heterogeneity on Loss of Accuracy from Genomic Prediction Models." *Biostatistics*, September. <https://doi.org/10.1093/biostatistics/kxy044>.

Zhang, H. Zhong, Y. Li, Z. Shi, H. Ren, Z. Zhang, X. Zhou, et al. 2021. "Sex- and Age-Related Trajectories of the Adult Human Gut Microbiota Shared across Populations of Different Ethnicities." *Nature Aging*. <https://doi.org/10.1038/s43587-020-00014-2>.

Sincerely,

Levi Waldron, Ph.D.

Professor and Chair
Department of Epidemiology and Biostatistics
Institute for Implementation Science in Population Health
Graduate School of Public Health and Health Policy
City University of New York

email: levi.waldron@sph.cuny.edu
phone: (646) 364-9616
website: <https://www.waldronlab.io>

To the reviewers of “**Meta-analysis of 22,710 human metagenomes defines an index of oral to gut microbial introgression and associations with age, sex, BMI, and diseases**”,

We thank the reviewers for their time and feedback, and for their patience in this revision, which required identifying a new co-first author to handle revisions after the original first-author has moved to a new position. We respond to all points raised by the reviewers, below.

REVIEWER COMMENTS

Reviewer #1 (Remarks to the Author):

The authors have done an excellent job to address my concerns. I have no further questions.
We thank the reviewer.

Reviewer #2 (Remarks to the Author):

The revised version of the manuscript addresses nearly all my comments, and I thank the authors for their additional work. It would be valuable if the authors could address these remaining comments.

1. Regarding my first comment requesting that the authors define the inclusion and exclusion criteria for datasets in this study. The authors now include additional details, but they are still somewhat vague. For example, the authors note that sample size was an inclusion criterion – but what was the cut-off? What was the minimal sample size for inclusion? For the criteria listed in this section, specific numbers should be given where possible.

We thank the reviewer. To clarify our standpoint further, datasets are prioritized for inclusion in curatedMetagenomicData based on several factors: sample size, richness of metadata, contributions from users, assessment of the importance of datasets, and alignment with funded projects. Only certain topics, such as colorectal cancer and response to immunotherapy, have been added through systematic review and adherence to PRISMA guidelines as described in resulting meta-analysis publications. To perform a topic-specific systematic review using curatedMetagenomicData requires identification of studies, definition of inclusion and exclusion criteria, then manual curation, processing, and inclusion in the database. We are not able to similarly apply systematic inclusion across the entire literature of published metagenomic studies due to the time-intensiveness of manual curation. Topics of particular community interest, such as cancer therapies and non-westernized communities, are often addressed by small datasets; these have been included when possible, regardless of sample size. We also tried, when possible, to include specific requests from external users. In the revised version of the manuscript, we made this explicit in the Methods section, where we describe the eligibility criteria and the process of manual curation:

“Metadata was manually curated from original literature, supplementary information, and other sources. Manually curated metadata was checked against a controlled vocabulary using an automatic grammar-checker. When new metadata is discovered - for example, a new publication makes different information explicit, our team updates datasets, ensuring cMD 3 provides the most complete and up-to-date metadata. **The accessibility of per-sample metadata in original publications remains the main driver of dataset acquisition. Metadata heterogeneity and richness is judged based on the odds that the data will be employed in further studies. Datasets addressing high interest topics (i.e. cancer therapies and non-westernized communities), have been included regardless of their sample size, and users’ requests have been prioritized.**”

2. The authors have addressed most of my comments related to figure 1. Minor comment, the X-axis title in figure 1B is unclear, consider “Top 30 metadata present in...”.

We changed axes titles and removed a few unnecessary hyphens and underscores from x axis labels. The updated figure is the following:

Figure 1. curatedMetagemomicData (cMD) 3 is an open-source software package providing uniformly processed metagenomic data and manually curated metadata, available in R/Bioconductor and via the command line. a) [...] b) Barplot showing the top 30 most complete metadata attributes among the samples from individuals above 12 years (left) and below (right), where the red lines mark 500 and 250 samples, respectively. Blue lines mark the total number of samples in each category. Metadata are colored according to whether they are required (red), recommended (blue), or optional (orange).

3. Regarding my comment (#8) on the strong impact stemming primarily from two studies, the authors say that this is likely due to having ‘highly diseased’ and ‘particularly healthy’ individuals in these cohorts, but is there any evidence for this in the metadata? Or are they hypothesizing this to be the case? Please clarify in the text.

We apologize for stating this unsupported hypothesis, and have removed it from the discussion of Figure 2 which is an analysis of healthy subjects only. We have conducted another analysis, showing that the average differential abundance of the top 60 functional potential variables most strongly associated with sex (largely shown in **Supplementary figure 3**) is larger in these two datasets than in others. We now state two possible explanations: an actual greater difference in microbial functional

potential between the sexes in these two datasets, or batch effect. To distinguish between these two possibilities would likely require replication studies from those same populations.

The text now reads:

“The ability to predict sex based on the gut microbiome, in particular functional potential, was unusually high in two datasets (JieZ_2017 and QinJ_2012). These datasets show higher than average differential abundance of discriminant features between sexes (those listed in **Suppl. Fig. 3b**). This could be due to differences in diet or biology of these subjects compared to other studies, or to the presence of batch effects. To distinguish between these two possibilities would likely require replication studies from the same populations.”

The comparison of coefficients of association between KEGG orthologs and sex is presented in a new panel of **Supplementary Fig. 3**:

Supplementary Figure 3. Meta-analysis of 5,505 individuals metagenomes from 18 datasets (3,288 females, 2,216 males) reveals sex-associated microbial differences in the KEGG Orthologs potential of the healthy, adult, stool microbiome. a) [...] b) Two outlier studies (QinJ_2012, JieZ_2017) from the Random Forest analysis of Figure 1c show average stronger associations between sex and KEGG Orthologs microbial carriage compared to the overall meta-analysis, when considering the 60 strongest KEGG associations.

Reviewer #3 (Remarks to the Author):

The authors have made a commendable effort in compiling and meta-analyzing multiple datasets. However, while the execution has improved, it still offers only relatively few novel biological insights beyond what has already been shown in previous studies. A score calculated from 305 species identified from 857 oral metagenomes is far from straightforward.

Easy access to curated data is crucial, and regular updates are valuable. The current submission therefore adds incremental value over the authors' own prior work and that of others. For example, gmrepo 2, published recently, also analyzed the majority of existing microbiome datasets in a unified manner across disease indications, and it provides an online interface for exploration (PMID: 34788838). It is surprising that the authors have not cited this recent work, nor its earlier version. Proper referencing is essential for situating this study within the existing body of literature, and I recommend that the authors address this throughout the manuscript (see other, specific examples below).

We thank the reviewer. The work by Dai et al. (PMID: 34788838) was acknowledged in the introduction (now reference 26):

“cMD 3 provides a higher level of manual curation than alternatives (Dai et al. 2022; Kasmanas et al. 2020), allowing adjustment for some potential confounding factors in meta-analysis.”

We also added one extra sentence to highlight an added value provided by cMD:

“Additionally cMD3 is freely available via ExperimentHub and example vignettes are included in the resource, making quick usability a key advantage.”

The term “microbial introgression” remains a poor choice. The term introgression elsewhere means transfer of genetic material from one species into the gene pool of other species, not the detection of sets of DNA sequences in stool. Furthermore, they cite three studies in their rebuttal to this critique. The authors cite the same references in both their rebuttal and the revised manuscript, but none of these sources use the term “introgression.” As such, they do not support the use of this term in the context of the manuscript.

The authors acknowledge that the enrichment of oral bacteria during dysbiosis has been studied in previous work, but limit their literature references to outdated sources. They do not seem to be aware of Franklin et al, who (arguably more appropriately) termed the detection of oral cavity microbes in stool “oral stool microbiome coalescence” (PMID: 35433514). Importantly, Chen et al. (2024) in Nature Microbiology studied oral bacterial enrichment in stool samples as a marker for dysbiosis using mouse models and through the re-analysis of paired human oral and stool samples across diseases. Why do we need this again, and why do the authors insist on using a term that is signaling incorrect biology?

The reviewer correctly states that the literature we cite does not explicitly define the word “introgression”. However, they clearly describe the same phenomenon that we call “oral-to-gut introgression”. It is also true that a few publications refer to “oral to gut microbiome coalescence” (Franklin et al. 2022; Liao et al. 2024), but other works talk about “oral to gut transmission” (Schmidt et al. 2019), and others call it “translocation” (Kageyama et al. 2023; Khor et al. 2021). Introgression has also been used (“Altered Microbial Transcription in Long-Term Proton Pump Inhibitor Use: Findings From a United States Cohort Study” 2024).

Coalescence, in ecology, defines the coexistence of two ecological communities after one's invasion into the other ("Interchange of Entire Communities: Microbial Community Coalescence" 2015, "Toward an Integrative Framework for Microbial Community Coalescence" 2024, "Soil Microbes and Community Coalescence" 2016). The two terms coexist in that introgression refers to invasion, while coalescence refers to the consequences of the invasion (Castledine et al. 2020). We may thus consider changing the term to "coalescence", but what we are sure to observe is the invasion phase, not much about the coexistence of communities (or parts of them) in the gut. Also, we are not directly observing oral to gut transmission in this study design, only the presence of oral-typical species in the stool. Thus, we maintain that "introgression" is the most conservative description of the score we propose. While a universally accepted terminology is currently lacking, we anticipate that future statistical and genomic evidence will facilitate its development.

We believe this section contributes to the paper's main goals by:

- illustrating meta-analyses achievable with curatedMetagenomicData,
- providing a single reference for microbiome associations with multiple health-related conditions,
- presenting a simple score to quantify the relative abundance of oral-typical species in the gut based on meta-analysis and a much larger sample than previous human studies, and without the inherent limitations of a score developed in mice, and
- demonstrating suitable methods for assessing biological phenomena in microbiome research using public data.

Regarding my technical comments from the first round, the authors explain that they standardized the features of the CLR-transformed data. In other words, they added pseudo counts, log-transformed geometric mean-weighted relative abundances, and then z-scored those log-transformed values. By this point, all zeros in the sequencing data have been converted into non-zero figures that are difficult to interpret. We have seen from the tumor microbiome field that dealing with low biomass, such as oral DNA in stool, can be particularly problematic. There are inherent risks in performing extensive data manipulations early on in the analytical process, especially without thorough robustness testing. The authors should consider testing their key findings with analyses that remain closer to the raw data and are robust to these various transformations. This would be satisfied with simple distribution plots of untransformed data (e.g. raw / unmodified relative read counts) that clearly support their biologically most meaningful statistical findings.

We acknowledge the potential risks of complex data transformations. First, we want to note that we performed transformations only for differential abundance analysis, not for machine learning or for the oral introgression score, which was defined as the sum of the simple relative abundance of oral species (using output directly from MetaPhlAn), and alternatively as Shannon entropy calculated on these species. To assess the impact on regression coefficients of CLR transformation and pseudocount addition, we repeated sex, age, and BMI analyses using a simpler arcsin square-root variance-stabilizing transformation, which retains all zero values and does not change rank. Correlations

between meta-analysis coefficients using the two methods are shown in the following figure, which shows a high degree of concordance overall.

Rebuttal Figure 1 - Correlation plots between meta-analysis coefficients of centred log-ratio and arcsine square root-transformed species abundances - Sex, age, and BMI meta-analyses coefficients from two different meta-analysis models: X-axis species relative abundances are transformed using the CLR, a 10e-5 zero imputation and a filter on prevalence at 5%; the CLR transformation corrects data compositionality. On Y-axis species relative abundances are transformed by the arcsin square root of the abundance divided by 100; the arcsin square root transformation is variance-stabilizing and assumes binomial probability of species non-negativity.

In reply to my question “if diversity goes up with age, would therefore a negative association of many species with age be expected due to compositional effects?”, the authors argue that “since we compute linear models after having performed Centered Log-ratio Transformation, we do not expect these results to be driven by compositionality, even in the presence of dominant species.” I remain unconvinced by this argument. While CLR and related techniques help address compositionality in certain contexts, they are not a cure-all for the inherent limitations of sum-constrained data. Consequently, an increase in diversity, which requires a more even distribution when the number of features is fixed, will likely result in a higher chance of negative correlations between CLR-transformed abundances and age.

We apologize for our lack of clarity in understanding the reviewer's point. We agree that in a sum-constrained dataset, the total number of features correlating positively and negatively with a continuous parameter (e.g., age or BMI) is more likely to show an opposite pattern than alpha-diversity. Nevertheless, both alpha-diversity and the individual top-ranking correlations are of significant interest and warrant presentation independently. We thus modified the original paragraph:

“Shannon diversity was positively associated with age (meta-analysis correlation coefficient = .09, 95% CI [.06, .12], P = 1.6 x 10⁻¹⁰), in line with the observations reported by (de la Cuesta-Zuluaga et al. 2019; Zhang et al. 2021), and negatively associated with BMI (meta-analysis correlation = -.08, 95% CI [-.12, -.05], P = 7 x 10⁻⁵) as previously reported (Castañer and Schröder 2018). However, among the species and genera above 1% prevalence, we identified more negative than positive taxonomic associations with age (55 species and 19 genera negatively correlated versus 12 species and 5 genera positively correlated, FDR = 0.01, avg I² = 20%, none with statistically significant heterogeneity [Cochran’s Q_{gen} >

0.05, higher threshold used to increase sensitivity to potential heterogeneity]) (Fig. 3a, Supplementary Table 3), and more positive than negative associations with BMI (116 species and 30 genera positively correlated versus 34 species and 15 genera negatively correlated, FDR = 0.01, avg. I² = 18%, [Cochran's I^2 > 0.05]) (Fig. 3b, Supplementary Table 4). Taxa showing the strongest correlations with age were the negatively associated *Bacteroides* species "OM05 12" (meta-analysis correlation = -0.11, 95% CI [-0.16, -0.07], $Q < 0.0001$) and the positively correlated *Bifidobacterium dentium* [...]"

to:

"Shannon diversity was positively associated with age (meta-analysis correlation coefficient = 0.09, 95% CI [0.06, 0.12], $P = 1.6 \times 10^{-10}$), in line with the observations reported by (de la Cuesta-Zuluaga et al. 2019; Zhang et al. 2021), and negatively associated with BMI (meta-analysis correlation = -0.08, 95% CI [-0.12, -0.05], $P = 7 \times 10^{-5}$) as previously reported (Castañer and Schröder 2018). Sixty-seven species and 24 genera were significantly correlated with age (FDR = 0.01, Fig. 3a, Supplementary Table 3), while 150 species and 45 genera were significantly correlated with BMI (Fig. 3b, Supplementary Table 4). Taxa showing the strongest correlations with age were the negatively associated *Bacteroides* species "OM05 12" (meta-analysis correlation = -0.11, 95% CI [-0.16, -0.07], $Q < 0.0001$) and the positively correlated *Bifidobacterium dentium* [...]"

Overall, while the paper offers limited novel biological insights, the well-curated data adds value, and the study could be a useful resource for the community. I recommend publication in a technical journal focused on providing such resources.

We thank the reviewer for the consideration of our work.

Bibliography

- "Altered Microbial Transcription in Long-Term Proton Pump Inhibitor Use: Findings From a United States Cohort Study." 2024. *Gastroenterology* 167 (2): 405–8.e3.
- Castañer, Olga, and Helmut Schröder. 2018. "Response to: Comment on "The Gut Microbiome Profile in Obesity: A Systematic Review."" *International Journal of Endocrinology* 2018 (December):9109451.
- Castledine, Meaghan, Pawel Sierocinski, Daniel Padfield, and Angus Buckling. 2020. "Community Coalescence: An Eco-Evolutionary Perspective." *Philosophical Transactions of the Royal Society B*, May. <https://doi.org/10.1098/rstb.2019.0252>.
- Cuesta-Zuluaga, Jacobo de la, Scott T. Kelley, Yingfeng Chen, Juan S. Escobar, Noel T. Mueller, Ruth E. Ley, Daniel McDonald, et al. 2019. "Age- and Sex-Dependent Patterns of Gut Microbial Diversity in Human Adults." *mSystems* 4 (4). <https://doi.org/10.1128/mSystems.00261-19>.
- Dai, Die, Jiaying Zhu, Chuqing Sun, Min Li, Jinxin Liu, Sicheng Wu, Kang Ning, Li-Jie He, Xing-Ming Zhao, and Wei-Hua Chen. 2022. "GMrepo v2: A Curated Human Gut Microbiome Database with Special Focus on Disease Markers and Cross-Dataset Comparison." *Nucleic Acids Research* 50 (D1): D777–84.
- Franklin, Samantha, Samuel L. Aitken, Yushi Shi, Pranoti V. Sahasrabhojane, Sarah Robinson, Christine B. Peterson, Naval Daver, et al. 2022. "Oral and Stool Microbiome Coalescence and Its Association With Antibiotic Exposure in Acute Leukemia Patients." *Frontiers in Cellular and Infection Microbiology* 12 (March):848580.

- "Interchange of Entire Communities: Microbial Community Coalescence." 2015. *Trends in Ecology & Evolution* 30 (8): 470–76.
- Kageyama, S., S. Sakata, J. Ma, M. Asakawa, T. Takeshita, M. Furuta, T. Ninomiya, and Y. Yamashita. 2023. "High-Resolution Detection of Translocation of Oral Bacteria to the Gut." *Journal of Dental Research*. <https://doi.org/10.1177/00220345231160747>.
- Kasmanas, Jonas Coelho, Alexander Bartholomäus, Felipe Borim Corrêa, Tamara Tal, Nico Jehmlich, Gunda Herberth, Martin von Bergen, Peter F. Stadler, André Carlos Ponce de Leon Ferreira de Carvalho, and Ulisses Nunes da Rocha. 2020. "HumanMetagenomeDB: A Public Repository of Curated and Standardized Metadata for Human Metagenomes." *Nucleic Acids Research*, November. <https://doi.org/10.1093/nar/gkaa1031>.
- Khor, Brandon, Michael Snow, Elisa Herrman, Nicholas Ray, Kunal Mansukhani, Karan A. Patel, Nasser Said-Al-Naief, Tom Maier, and Curtis A. Machida. 2021. "Interconnections between the Oral and Gut Microbiomes: Reversal of Microbial Dysbiosis and the Balance between Systemic Health and Disease." *Microorganisms* 9 (3): 496.
- Liao, Chen, Thierry Rolling, Ana Djukovic, Teng Fei, Vishwas Mishra, Hongbin Liu, Chloe Lindberg, et al. 2024. "Oral Bacteria Relative Abundance in Faeces Increases due to Gut Microbiota Depletion and Is Linked with Patient Outcomes." *Nature Microbiology* 9 (6): 1555–65.
- Schmidt, Thomas S. B., Matthew R. Hayward, Luis P. Coelho, Simone S. Li, Paul I. Costea, Anita Y. Voigt, Jakob Wirbel, et al. 2019. "Extensive Transmission of Microbes along the Gastrointestinal Tract," February. <https://doi.org/10.7554/eLife.42693>.
- "Soil Microbes and Community Coalescence." 2016. *Pedobiologia* 59 (1-2): 37–40.
- "Toward an Integrative Framework for Microbial Community Coalescence." 2024. *Trends in Microbiology* 32 (3): 241–51.
- Zhang, H. Zhong, Y. Li, Z. Shi, H. Ren, Z. Zhang, X. Zhou, et al. 2021. "Sex- and Age-Related Trajectories of the Adult Human Gut Microbiota Shared across Populations of Different Ethnicities." *Nature Aging*. <https://doi.org/10.1038/s43587-020-00014-2>.

We have addressed the following final requests from reviewer 3:

- *However, while this provides some evidence of robustness, inspecting the correlation plot suggests a potential shift in the directionality of many taxonomic associations with age. It appears that there are now more positive correlations with age whereas the authors report more negative associations for CLR transformed data in the revised manuscript. The authors should clarify this, and consider reporting results from both analyses, with numbers of positive versus negative associations for both transformations. Fortunately, the strongest reported associations appear unaffected, which is reassuring. Adding this clarification would enhance the work by introducing appropriate caution regarding less stable associations implicitly, without requiring further analyses.*

We have added a “Sensitivity Analysis” subsection to the Results addressing this comment:

Effect of compositional data transformation

To assess the sensitivity of meta-analysis results to the choice of data transformation, we reanalyzed sex, age, and BMI using arcsine square-root transformed species relative abundances. The meta-analysis coefficients obtained from the two transformations were highly concordant (Spearman’s $r=0.9$ for sex, $r = 0.68$ for age, $r = 0.79$ for BMI; see Supplemental Figure 4). However, the proportion of significant positive and negative associations (FDR = 0.2) between species and these variables differed substantially depending on the transformation applied. With CLR transformation, the percentage of positive meta-analysis coefficients were 67% for sex (male), 21% for age, and 76% for BMI. In contrast, the arcsin square-root transformation yielded 31% for sex (male), 82% for age, and 34% for BMI. The results for the CLR transformation are reported in the main text, while tables of meta-analysis results under both transformations are available in Supplementary Files 2 (sex), 3 (age), and 4 (BMI).

And a comment at the end of the second paragraph of the Discussion:

We noted that although meta-analysis coefficients were strongly correlated between CLR compositional transformation and variance-stabilizing arcsin square-root transformation, the direction of many associations differed depending on the transformation applied. Interpretation of results varies between absolute and relative abundance, but this analysis does not favor one transformation over the other.

- *Lastly, regarding terminology, I understand where the authors are coming from. They observed an enrichment of reads from oral taxa in fecal DNA. This can arise from any number of dynamic processes, none of which can be elucidated by the current work. As such, coining a novel term with dynamic implication does not seem appropriate. The authors argue that: “what we are sure to observe is the invasion phase, not much about the coexistence of communities (or parts of them) in the gut.” Since the associations are from static sequencing data, and not derived from time series analyses or experiments, this is an overinterpretation. “Oral taxon enrichment score”, rather than any process related terms, such as coalescence or translocation, could be adequate for the present work.*

We have replaced the term “introgression” with “enrichment” everywhere in the manuscript.